# *Finding Flawed Fictions*: Evaluating Complex Reasoning in Language Models via Plot Hole Detection

**Kabir Ahuja    Melanie Sclar    Yulia Tsvetkov**
Paul G. Allen Center for Computer Science & Engineering
University of Washington
Seattle, USA
{kahuja,msclar,yuliats}@cs.washington.edu

## Abstract

Stories are a fundamental aspect of human experience. Engaging deeply with stories and spotting *plot holes*—inconsistencies in a storyline that break the internal logic or rules of a story's world—requires nuanced reasoning skills, including tracking entities and events and their interplay, abstract thinking, pragmatic narrative understanding, commonsense and social reasoning, and theory of mind. As Large Language Models (LLMs) increasingly generate, interpret, and modify text, rigorously assessing their narrative consistency and deeper language understanding becomes critical. However, existing benchmarks focus mainly on surface-level comprehension. In this work, we propose *plot hole detection* in stories as a proxy to evaluate language understanding and reasoning in LLMs. We introduce FLAWEDFICTIONSMAKER, a novel algorithm to controllably and carefully synthesize plot holes in human-written stories. Using this algorithm, we construct a benchmark to evaluate LLMs' plot hole detection abilities — FLAWEDFICTIONS— robust to contamination, with human filtering ensuring high quality. We find that state-of-the-art LLMs struggle in accurately solving FLAWEDFICTIONS regardless of the reasoning effort allowed, with performance significantly degrading as story length increases. Finally, we show that LLM-based story summarization and story generation are prone to introducing plot holes, with 50%+ and 100%+ increases in plot hole detection rates with respect to human-written originals.

⬤ https://github.com/kabirahuja2431/FlawedFictions

## 1 Introduction

Narratives form a fundamental mode of human cognition and meaning-making, acting as a primary way people organize, experience, and construct reality (Bruner, 1991). When we engage with stories, we typically go beyond a literal understanding of *what happened*, instead performing complex and nuanced reasoning that involves mental representation of a story's world and its characters (Gerrig, 1993; Mar & Oatley, 2008; Zunshine, 2006; Kidd & Castano, 2013). Ultimately, narrative understanding is a reflection of broader human cognitive capacities for language comprehension and reasoning (Kintsch, 1998).

In this work, we propose to quantify narrative understanding in LLMs as a novel test bed of general language understanding and reasoning abilities. While different language understanding benchmarks are widespread in existing literature(Wang et al., 2018; 2019; Zellers et al., 2019; Hendrycks et al., 2020; Jaradeh et al., 2023), they often fail to capture the full spectrum of abilities present in narrative understanding. For example, the popular MMLU benchmark (Hendrycks et al., 2020) evaluates advanced multi-hop knowledge, but lacks assessment of pragmatics and implicit social dynamics inherent in narratives. Existing datasets studying such capabilities (Mostafazadeh et al., 2016; Sap et al., 2019; Sprague et al., 2024; Kim et al., 2023), on the other hand, are not suited for benchmarking LLMs at scale, as they focus on very short or fully synthetic stories that lack core elements of

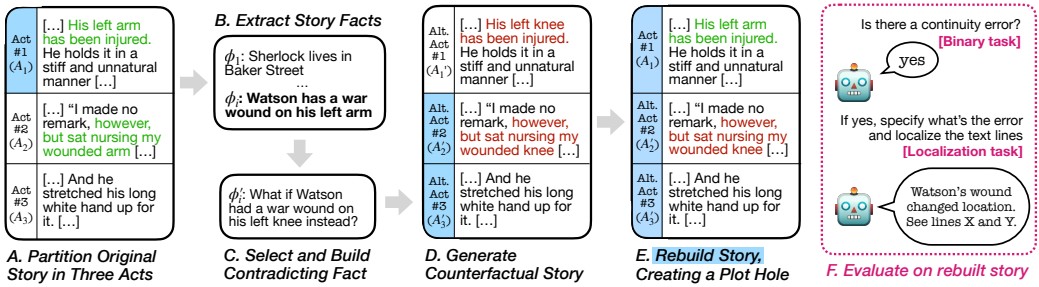

Figure 1: Example of FLAWEDFICTIONSMAKER (without the filtering step) in action that can be used to introduce plot holes in a plot hole-free story.

narrative structure. As a consequence, it remains difficult to holistically assess overall progress in language understanding and reasoning, despite recent advances in improving LLM reasoning capabilities through advanced prompting (Wei et al., 2022; Yao et al., 2024; Wang et al., 2023) or inference time scaling (Lambert et al., 2024; Guo et al., 2025).

How do we quantify such "deeper narrative understanding"? **We propose a novel task of plot hole detection as a proxy to assess deep narrative understanding and reasoning in LLMs.** *Plot holes* are inconsistencies in a story that go against the logic flow established by the story plot (Ryan, 2009), with significant discourse dedicated to both locating[1] and preventing them during screen writing (McKee, 1997; MasterClass, 2021). Plot hole detection requires nuanced reasoning about the implications of established facts and elements, how they interplay, and their plausibility. Specifically, robust *state tracking* is needed to follow entities and rules established by the story over a long context; *commonsense and pragmatic reasoning* are needed for interpreting implicit world knowledge and beliefs; and *theory of mind* is required for reasoning over beliefs, motivations, and desires of characters. Beyond acting as a test bed for complex reasoning, models that can accurately assess plot holes in stories can be useful to improve consistency in writing, be it human- or machine-generated.

We propose FLAWEDFICTIONSMAKER, an automatic method for introducing plot holes in existing stories. Our algorithm functions by extracting relevant facts from the first act of a story and negating them in subsequent acts to introduce an inconsistency (Figure 1). We then use FLAWEDFICTIONSMAKER to curate the first high-quality benchmark for plot hole detection—FLAWEDFICTIONS—consisting of short stories labeled with their inherent inconsistencies or lack thereof. We opt for a partial synthetic data approach to construct this benchmark to make it dynamically extensible and avoid data contamination (i.e., memorization of the existing stories with plot holes during LLM training). Data generated through our algorithm is then manually verified to ensure quality. FLAWEDFICTIONS consists of two tasks: a binary classification task where the LLM must determine whether there is a plot hole in the story, and a localization task where the model determines both the text span introducing the plot hole and the one with the information being contradicted. The first task is a naturally reduced version of the second.

We find that a large majority of frontier LLM and reasoning models like GPT-4o, o3-mini, and Llama-3.3-70B struggle in FLAWEDFICTIONS, with story length having a significant negative effect on LLM's plot hole detection capabilities. FLAWEDFICTIONSLONG, an extension of our benchmark containing longer stories in the 1,200-4,000 word range, proves particularly difficult, with **almost all models obtaining close to random level performance** on the classification task. Plot hole detection also proves to be difficult irrespective of the reasoning budget allowed: state-of-the-art reasoning models, such as o1 and o3-mini, show a stable and sometimes worsened performance with increased reasoning budget.

Finally, we conduct a case study to explore the use of plot hole detection for evaluating consistency of LLM generated stories. Considering the tasks of *story summarization* and

---

[1]This is especially true in the context of films, with dedicated subreddits like `r/plotholes` and `r/MovieMistakes`, or a Goofs section dedicated to each film page on IMDB.

*contemporary adaptation* of classical short stories, we find that LLM-generated outputs trigger significantly more plot-holes—**over 50% more in summarization and 100% more in contemporary adaptation**—using our best performing model on FLAWEDFICTIONS.

Overall, our work introduces a novel evaluation task—plot hole detection—for assessing deeper language understanding and reasoning in LLMs, along with a controllable synthetic data generation algorithm FLAWEDFICTIONSMAKER, and an accompanying benchmark FLAWEDFICTIONS, enabling systematic and holistic comparison of state-of-the-art models, uncovering critical gaps in their narrative comprehension, and providing a powerful framework for evaluating the quality of LLM-generated stories. We will make our dataset and code publicly available at the time of publication.

## 2 Defining Plot Holes: Continuity Errors

Plot holes are commonly categorized into multiple categories (Shattuck, 2024) including: *continuity errors* (contradictions of established facts), *out of character behavior* (actions inconsistent with established motivations), *factual errors* (historical anachronisms or real-world inaccuracies), *impossible events* (violations of science or logic), and *unresolved storylines* (incomplete plot threads). See Table 2 in Appendix for examples. We focus on *continuity errors* as they encompass the most general form of plot hole: both out of character behavior and impossible events can be framed as breaks in continuity, as they contradict established character traits or story settings. While Ryan (2009) distinguishes between *harmless* plot holes (serving symbolic functions rather than causal functions) and *truly unbridgeable* ones (affecting plot integrity), our approach treats both types as under the same umbrella.

Formally, consider a fictional story $f$ containing a set of propositions $\mathcal{F} = \{\phi_1, \ldots, \phi_n\}$ that are true in the fictional world of $f$ (e.g., "Sherlock Holmes lived on Baker Street" is a statement that is true in the fictional world of Sherlock Holmes). We make use of the possible worlds theory from Lewis (1978), defining the notation $\text{isTrue}(f, \phi)$ to denote that the proposition $\phi$ is true in the fictional world of $f$ and define the shorthand $\text{isTrue}(f, \mathcal{F}) := \text{isTrue}(f, \phi_1) \wedge \cdots \wedge \text{isTrue}(f, \phi_n)$. We can then define a continuity error:

**Definition 2.1 (Continuity Error)** *A proposition $\phi_e$ in a story is associated with a continuity error if the following inference rule holds:*

$$\text{isTrue}(f, \mathcal{F} \setminus \{\phi_e\}) \implies \text{isTrue}(f, \neg\phi_e) \tag{1}$$

*In other words, if using all the propositions in $\mathcal{F}$ except $\phi_e$ we can conclude that the negation of $\phi_e$ is true in $f$, that means $\phi_e$ is logically inconsistent with the rest of the story.*

While the above definition formalizes many types of continuity errors, it assumes the contradictions are derived using the propositions explicitly stated in the story. However, reasoning for contradictions in stories often requires implicit knowledge such as one's world understanding and beliefs. We expand our definition to incorporate such implicit knowledge in Appendix §A.1, but informally, an expanded version of Definition 2.1 can be expressed as: *If using all the propositions in $\mathcal{F}$ except $\phi_e$, along with **a set of a reader's belief statements (or community of readers') that are also non-vacuously true in $f$**, one can derive that the negation of $\phi_e$ is true in $f$, then $\phi_e$ is considered logically inconsistent with the rest of the story.* We highlight this difference to emphasize that reasoning for plot holes in stories is not simply about checking for contradictions using rules and statements explicitly stated in text, but necessarily incorporates common sense and world knowledge.

## 3 Automatically Generating Plot Holes in Stories

Conceptually, FLAWEDFICTIONSMAKER is a story-editing approach that introduces an inconsistency by selecting one of the propositions stated earlier in the story and negating it in the later parts. Our method, summarized in Figure 1, consists of a 5-staged pipeline:

**1. Three Act Structure Extraction.** We start by dividing the story into the traditional three act structure Aristotle (1902), consisting of *Act One* ($A_1$), where the main characters and

setting of the story are introduced, *Act Two* ($A_2$), where the main conflict is developed, and *Act Three* ($A_3$), which builds to the climax and resolves the main conflict. This division aids to control where the original proposition is established in the story and when it gets contradicted in the later parts of our pipeline. We perform the three-act extraction of an original story $f$ through LLM prompting, and denote it $\{A_1, A_2, A_3\} \leftarrow \texttt{ThreeActExtract}(f)$. Note that $f$ is the concatenation $f = A_1 \cdot A_2 \cdot A_3$ of the resulting three acts $\{A_1, A_2, A_3\}$.

**2. Proposition Extraction and Scoring.** Next, we retrieve the set of propositions that are stated in the first act $A_1$ of the the story through LLM prompting: $\{\phi_1, \phi_2, \ldots\} \leftarrow \texttt{PropExtract}(A_1)$. Specifically, these propositions contain the information established about the characters (foreground) and the setting (background) of the story[2]. These propositions help us to control the specific continuity error that we wish to introduce. We also include a proposition scoring step, which determines how relevant is a proposition $\phi$ to the plot in the second and third acts using a 4-point Likert scale: $s_\phi \leftarrow \texttt{PropScore}(\phi; A_1, A_2, A_3)$. We only retain the propositions that are moderately important ($s_\phi \in \{2, 3\}$) to avoid negating statements that lead to no change in the story, or changing a fundamental aspect which would render the final story completely nonsensical.

**3. Counterfactual Story Generation.** We rewrite the story while negating an original proposition $\phi$ with LLM prompting (Qin et al., 2019), $A_1^{\neg\phi} \cdot A_2^{\neg\phi} \cdot A_3^{\neg\phi} \leftarrow \texttt{Counterfact}(\phi, A_1, A_2, A_3)$. Note that negating $\phi$ does not just negate that single statement in the story, but may also lead to modifying other existing propositions to maintain coherence and plausibility (e.g., when changing a character's nationality, their name might need to be changed).

**4. Re-building Story ("Patching").** Now, given the original story $f = A_1 \cdot A_2 \cdot A_3$ and its counterfactual $f^{\neg\phi} = A_1^{\neg\phi} \cdot A_2^{\neg\phi} \cdot A_3^{\neg\phi}$, we create a story with a potential continuity error by concatenating $A_1$ from the original story and the subsequent acts from the counterfactual: $f^{\text{patch}} := A_1 \cdot A_2^{\neg\phi} \cdot A_3^{\neg\phi}$.[3]

**5. Filtering.** As a final step, we ensure that the patched story results in an inherent story inconsistency. This includes removing obvious LLM prompting issues, such as cases where $A_2 = A_2^{\neg\phi}$ or $A_3 = A_3^{\neg\phi}$, or preemptively removing cases where there are too many changes ($> 5$) in the counterfactual, since an increasing number of LLM edits increases the probability of making counterfactual reasoning errors. We additionally run an extremely aided version of the task as a quality filter: we prompt an LLM with $f^{\text{patch}}$, specifying the modified lines in $A_2^{\neg\phi}$ and $A_3^{\neg\phi}$ and use the LLM as a judge of whether these lines introduce a continuity error. This much simpler problem[4] aids us in eliminating cases with errors during Step 3, where the newly introduced propositions might still be consistent with the original fact $\phi$. To improve reliability of filtering, we use self-consistency (Wang et al., 2023), only retaining the cases where the model predicts a continuity error in at least 4 out of the 5 completions. At the filtering step we also prompt the model to provide an explanation if it predicts that the modified lines introduce a continuity error, which is shown later to humans to verify if the stories actually have a continuity error.

We use GPT-4o for all steps, except for counterfactual story generation where we qualitatively found GPT-4-turbo to perform significantly better. All the prompts used for our pipeline are provided in Appendix § A.10.1. While four out of five steps in our pipeline make use of LLMs, we do not claim that LLMs to be perfect at these tasks. Step 3, which requires counterfactual reasoning can in particular be difficult for LLMs with evidence in prior work (Huang et al., 2024). Hence, we follow our automatic generation process with human verification to curate a high quality benchmark.

---

[2]We choose to extract the propositions only from the first act because we want to consider information that is established earlier in the story but later contradicted. Doing this help us controllably create plot holes in the later acts.

[3]We select this patching method for simplicity. Note that other choices such as $A_1 \cdot A_2^{\neg\phi} \cdot A_3$ or $A_1^{\neg\phi} \cdot A_2 \cdot A_3$ might also have been appropriate.

[4]This is a much simpler problem because the model only needs to check the lines marked for a contradiction, as opposed to all the possible combinations of them.

**6. Human Verification.** Annotators are provided with stories and the proposed continuity errors from FLAWEDFICTIONSMAKER, and are asked to rate if the continuity error is legitimate or not, with at least 3 annotators per instance. Note that the annotators receive the final outputs after the Filtering step for verification. An example is considered legitimate only when the majority agrees about its legitimacy.[5]

# 4 FLAWEDFICTIONS: Tasks, Metrics, and Dataset Statistics

We now discuss how the data generated by FLAWEDFICTIONSMAKER are used to create a benchmark—FLAWEDFICTIONS—for reasoning about plot holes in stories across two tasks.

**Classification Task.** This represents a simpler version of the plot hole detection problem where the model is tasked to predict whether a continuity error exists in a story—a binary classification task. The positive examples (with continuity errors) come from data generated using our method, while the negative examples use original unmodified stories[6]. All synthesized positive examples are verified by humans before being included in our benchmark.

**Two-Way Localization Task.** While the classification task provides some signal for the correctness in a model's assessment for continuity errors, we are ultimately interested in evaluating the specific continuity error predicted rather than merely its presence or absence. Given that evaluating open-ended natural language explanations remains challenging even when ground truths are available, we propose a two-way localization task as a proxy for continuity error explanation. In this task, the model must predict two sets of sentences in the story: $\mathcal{S}_{\text{Error}}$, containing the sentences in the story that contain the error (i.e., that imply $\neg\phi$ where $\phi$ is the original proposition), and $\mathcal{S}_{\text{Contr}}$, containing sentences that entail $\phi$. We compare these predicted sets with the ground truth from FLAWEDFICTIONSMAKER to evaluate the validity of the model's predicted continuity error. Specifically, we define the Continuity Error Evaluation Full metric (`CEEval-Full`), which operates in two steps: first checking if the model correctly identifies whether an error exists, and if so, verifying if the predicted sentence sets contain at least one sentence from the ground truth [7]. If the model incorrectly determines the existence of a continuity error, it receives a score of 0.

**Dataset Composition and Statistics.** To construct our benchmark's positive and negative examples, we scraped short story collections from Project Gutenberg using keywords such as *fairytales* and *short stories*. We retained only stories under 1200 words to reduce cognitive load on human annotators. From approximately 300 stories edited with FLAWEDFICTIONS-MAKER and verified by humans, we selected 207 stories (**70% acceptance rate**) as positive examples. We then included an equal number of original unmodified stories as negative examples, resulting in a total of 414 examples in FLAWEDFICTIONS. The final dataset has an average length of 731 words and includes classical fairy tales, myths, legends, and historical fiction. See detailed statistics in Table 3, with dataset examples in §A.7.

**FLAWEDFICTIONSLONG.** Our preliminary experiments showed LLMs struggle with assessing plot holes as story length increased (see §A.5.2 in Appendix). Consequently, we curated an extension of FLAWEDFICTIONS– FLAWEDFICTIONSLONG – consisting of stories 1,200-4,000 words long: we selected stories from FairyTaleQA (Xu et al., 2022) meeting this length criterion and processed them through FLAWEDFICTIONSMAKER to generate positive examples. Due to increased cognitive load and annotation costs, only one-third of these longer stories were annotated by Prolific users, with the remainder annotated by this paper's lead author. Post-verification, we selected 97 stories as positive examples and 103 original stories as negative examples, totaling 200 examples in FLAWEDFICTIONSLONG. Unlike FLAWEDFICTIONS, FLAWEDFICTIONSLONG consists entirely of fairy tales and has an average length of 2703 words per story.

---

[5]Annotators were hired via Prolific. Details about the annotation process are in Appendix §A.2.

[6]We discuss alternative approaches for negative examples in §A.6 in Appendix.

[7]We use this less strict metric because our primary concern is whether the model recognizes the error correctly, rather than whether it identifies all instances of the error (or contradicted proposition) in the story.

# 5   How Well do Frontier LLMs Perform on FLAWEDFICTIONS?

**Experimental Setup.** We evaluate different proprietary LLMs from OpenAI and Anthropic as well as open weights models Llama-3 (Van Der Maaten et al., 2024), Deepseek-R1 Distilled (Guo et al., 2025), and Qwen-2.5 (Yang et al., 2024) series, which represent the most recent iterations available at the time of publication. For o1 and o3-mini, we experiment with the three values of reasoning efforts parameter provided in the API – low, medium, and high, which controls the amount of intermediate reasoning tokens generated before the final completion. Similarly, Anthropic API provides *extended thinking* mode for Claude 3.7 Sonnet model, which uses intermediate tokens to "think" before answering. We also consider another inference time scaling strategy, where we augment the plot hole detection model i.e. *generator* with a *verifier* model (Cobbe et al., 2021) that validates the legitimacy of the plot hole detected by the generator. Our verifier is a Claude 3.5 Sonnet model prompted to perform the verification task. For more details on the experimental setup, prompts that we use, and other prompting methods that we evaluate such as few-shot and chain-of-thought (CoT), refer to Appendix §A.4.

**Baselines.** To highlight the contextual nature of our problem, we use an entailment model that examines all ordered sentence pairs in a story to detect contradictions. If no contradictory pairs are found, the baseline predicts the story lacks continuity errors; otherwise, the pair with highest contradiction confidence determines the error location. We employ DeBERTa-v3-large (He et al., 2021) fine-tuned on MNLI (Williams et al., 2018) (achieving 91% on MNLI dev) as our entailment model. We also consider a random baseline and a baseline that always predicts *No continuity error found*, with the latter achieving 50% on CEEval-Full due to our balanced dataset.

**Benchmarking Human Performance.** To establish a meaningful baseline against which to compare performance of various LLMs on FLAWEDFICTIONS, we estimated human performance by recruiting 9 undergraduate English majors who evaluated 50 samples from FLAWEDFICTIONS with three responses per sample. Further details about the study are provided in Appendix §A.2. It is important to recognize that this task is non-trivial for humans as it requires a high amount of cognitive load due to the limited working memory, which has been shown to affect reading comprehension abilities in adults and children (Barreyro et al., 2025; Cain et al., 2004).

## 5.1   Results

Performance of different LLMs on FLAWEDFICTIONS is provided in Table 1a. On the classification task, we observe all open weights models like Llama-3.1-70B and DeepSeek-R1-Qwen-32B to perform comparable to the random baseline. Similar trends were also observed for GPT-4o-mini, GPT-4-turbo, and Claude 3.5 Haiku models. While other models like GPT-4o, o3-mini, o1 demonstrate superior performance compared to the aforementioned models, it is only Claude 3.5 Sonnet, which matches human performance.

For the localization task, we again notice Claude 3.5 Sonnet to demonstrate superior performance CEEval-Full score of 0.67 (the ideal score is 1), and with a verifier it matches human performance. Other than Claude 3.5 Sonnet, Claude 3.7 Sonnet with extended thinking, and o1, other models only show marginal improvements over the baseline that always outputs no error. The entailment baseline gets negligible score on CEEval-Full. This underscores the complex contextual nature of our task, which cannot be solved by merely finding two contradictory statements in the story. When viewed in isolation, two statements which in the broader context of the story are consistent with each other might appear to contradict each other. Consequently, the entailment baseline tends to trigger false positives and incorrectly localizes $\mathcal{S}_{\text{Error}}$ and $\mathcal{S}_{\text{Contr}}$.

**Results on FLAWEDFICTIONSLONG.** We also conducted evaluations on FLAWEDFICTIONS-LONG, which contains stories approximately four times the length of those in FLAWEDFICTIONS on average. Table 1b shows that there is a sharp drop in performance on FLAWEDFICTIONSLONG, with the best-performing model i.e. o1 obtaining a CEEval-Full score of 0.53, only marginally outperforming the Always No Error baseline. Although FLAWEDFICTIONS-LONG has longer stories than FLAWEDFICTIONS, it still comprises stories with fewer than 4,000 words. This presents a significant limitation, as in realistic scenarios, plot holes are

| Model | Accuracy | CEEval-Full |
|---|---|---|
| Random Baseline | 0.50 | 0.00 |
| Always No Error Baseline | 0.50 | 0.50 |
| Entailment Baseline | 0.53 | 0.04 |
| Llama-3.3-70B | 0.57 | 0.38 |
| Llama-3.1-8B | 0.50 | 0.10 |
| DeepSeek-R1-Qwen-32B[‡] | 0.56 | 0.35 |
| Qwen2.5-32B | 0.53 | 0.31 |
| GPT-4o (with CoT) | 0.64 | 0.58 |
| GPT-4o-mini (with CoT) | 0.53 | 0.32 |
| GPT-4-turbo (with CoT) | 0.57 | 0.55 |
| o1[‡] (Low) | 0.71 | 0.65 |
| (Medium) | 0.70 | 0.65 |
| (High) | 0.69 | 0.64 |
| o3-mini[‡] (Low) | 0.55 | 0.52 |
| (Medium) | 0.62 | 0.53 |
| (High) | 0.63 | 0.47 |
| Claude 3.5 Haiku (with CoT) | 0.57 | 0.46 |
| Claude 3.5 Sonnet | **0.76** | 0.67 |
| (with Verifier) | 0.74 | **0.68** |
| Claude 3.7 Sonnet | 0.66 | 0.55 |
| (with Extended Thinking)[‡] | 0.73 | 0.66 |
| Human Performance | **0.76** | **0.68** |

(a) Performance comparison of different models on the FLAWEDFICTIONS.

| Model | Accuracy Task | CEEval-Full |
|---|---|---|
| Random Baseline | 0.50 | 0.00 |
| Always No Error Baseline | 0.51 | 0.51 |
| Entailment Baseline | 0.48 | 0.00 |
| Llama-3.3-70B | 0.53 | 0.16 |
| Llama-3.1-8B | 0.48 | 0.02 |
| DeepSeek-R1-Qwen-32B[‡] | 0.52 | 0.27 |
| Qwen2.5-32B | 0.51 | 0.23 |
| GPT-4o | 0.57 | 0.35 |
| (with CoT) | 0.56 | 0.42 |
| GPT-4o-mini | 0.51 | 0.08 |
| (with CoT) | 0.43 | 0.20 |
| GPT-4-turbo | 0.52 | 0.52 |
| (with CoT) | 0.54 | 0.53 |
| o1[‡] (Medium) | **0.61** | **0.53** |
| o3-mini[‡] (Low) | 0.53 | 0.46 |
| (Medium) | 0.56 | 0.42 |
| (High) | 0.45 | 0.07 |
| Claude 3.5 Haiku | 0.48 | 0.37 |
| Claude 3.5 Sonnet | 0.56 | 0.35 |
| (with Verifier) | 0.60 | 0.50 |
| Claude 3.7 Sonnet | 0.49 | 0.29 |
| (with Extended Thinking)[‡] | 0.54 | 0.37 |

(b) Performance comparison of different models on FLAWEDFICTIONSLONG.

Table 1: Performance comparison of different models on FLAWEDFICTIONS and FLAWEDFICTIONSLONG. Models trained to use test-time compute for reasoning i.e. reasoning models are marked with [‡].

more common for long-form stories like feature films or series of books and films, which typically contain substantially more than 4,000 words. Therefore, our findings suggest that there exist substantial gaps in the capabilities of contemporary LLMs to reliably detect and evaluate consistency issues in long-form narratives.

**Extra Test Time Compute Provides Minimal Gains.** Interestingly, we found that extra test time compute would in most cases result in minimal improvement towards accurately detecting continuity errors. Table 1a shows that increasing the reasoning effort from low to high results in a drop in CEEval-Full score for both o1 and o3-mini. For o3-mini this represents an increase from less than 1000 reasoning tokens on average to over 5000 tokens (roughly 5 times the number of tokens in the stories) for reasoning, yet results in degraded performance. Similarly, the DeepSeek-R1 distilled models, which are also trained to utilize test time compute for reasoning, demonstrate suboptimal performance on the task, with only marginal improvements over the base Qwen2.5-32B model. The sole exception is observed for Claude 3.7 Sonnet, where enabling extended thinking results in substantial improvements. Nevertheless, Claude 3.5 Sonnet, which utilizes no additional test time compute for reasoning and generates approximately one-tenth the tokens of Claude 3.7 Sonnet with extended thinking, achieves marginally superior performance. Figure 5 in the Appendix illustrates the relationship. These findings raise important questions regarding *whether the absence of datasets similar to* FLAWEDFICTIONS *while training reasoning models explains the limited improvements observed, or whether inference time scaling is not adequate for solving problems like plot hole detection?* A frequently observed limitation of reasoning models is their tendency to persist on a wrong hypothesis for a potential plot hole during the reasoning process and continue with that chain of thought resulting in an incorrect judgment. Since the space of possible hypotheses in our problem is at least quadratic in the number of sentences in the story, iterating through each of the hypothesis through intermediate generation becomes computationally prohibitive for extended narratives. We defer a more comprehensive investigation of these questions for the future work.

**What types of mistakes do LLMs make in assessing plot holes?** We qualitatively analyzed the types of reasoning errors LLMs—specifically, GPT-4o, Claude 3.5 Sonnet, and Claude 3.5 Sonnet with Verifier—make on FLAWEDFICTIONS. We find that **models often misinterpret**

**characters' motivations or behavior**, e.g. a character being deceptively nice or bluffing is not necessarily a continuity error. Another commonly observed mistake is **models wrongly tracking and interpreting entities' states**, e.g. miscounting the number of alive characters, or incorrectly assessing the passage of time, and interpreting these as plot holes. We also find that sometimes **models fail to understand genre conventions**, misinterpreting fantastical elements in fairy tales as logical inconsistencies. Finally, it is also common for models to **misinterpret or overinterpret established rules or plot points** in a story. For example, Claude 3.5 Sonnet incorrectly identifies a contradiction when a character tries multiple suits after stating they "will not try any suit more than once". We provide many examples for these errors in Appendix §A.8. In contrast, such reasoning errors were rare among humans, whose mistakes usually stem from overlooking details that may be attributed to humans' limited working memory. This is also evidenced by humans showing a higher precision but lower recall than the best models on FLAWEDFICTIONS (see Table 5 in Appendix).

## 6 Measuring Logical Consistency in LLM Generated Narratives

A study by Mirowski et al. (2023) examining LLMs as screenplay co-writers identified that LLM-generated narratives exhibited issues with maintaining consistency in plot's logic or characters' behaviors. While these observations were made based on participants' interviews, we propose a quantitative evaluation framework for the phenomenon. Our setup consists of generating short stories using LLMs, which are subsequently evaluated for the existence of plot holes using our best model on FLAWEDFICTIONS i.e. Claude 3.5 Sonnet with Verifier. We define *continuity error detection rate* as the fraction of the generated stories for which the detection model identifies a continuity error.

Rather than employing unconditional and fully open-ended generations from the models, we focus on summarization and contemporary adaptation tasks. In contemporary adaptation, the model is instructed to generate a modern retelling of a classical fairy tale i.e. transporting the setting of the story to modern times, while preserving similar themes, central conflict, and characters from the original story. We opted for conditional generation as they facilitate utilization of original human-authored stories as controls while checking for continuity errors. For summarization, we utilized 200 fairy tale stories from FairyTale QA dataset and prompt the models to write concise summaries of roughly 1000 words. For the

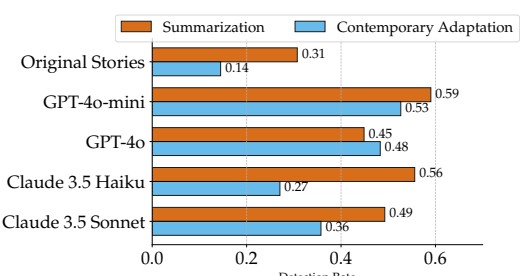

Figure 2: Continuity Error Detection Rate for stories generated using different LLMs for summarization and contemporary adaptation tasks.

contemporary adaptation task, we utilize the original stories (total of 207) included in FLAWEDFICTIONS. We provide the prompts used for generation for both tasks in the Appendix §A.10.3. Our focus on short stories for generations (i.e. less than 1200 words), stems from the suboptimal performance of even the highest-performing models on long stories.

**Results.** The continuity error rates for the two tasks are provided in Figure 2. We observe that generations from different LLMs demonstrate significant error rates relative to the original stories for both tasks. In case of summarization, lowest error rate was observed with GPT-4o, while still representing a 50% increase (0.31 to 0.45) in detected continuity errors when compared with original un-summarized stories. For contemporary adaptation the increase in error rates was even higher, with an almost 100% increase (0.14 to 0.27) in the best case for Claude 3.5 Haiku and a 278% (0.14 to 0.53) in the worst for GPT-4o-mini. For summarization, we identified that the models frequently omitted critical information in the summary that would render future events inconsistent with the rest of the narrative. E.g. in a story with a sequence of events *The dragon is on an year long sleep → He is awakened by his brothers → He chases the prince*, the summary from Claude 3.5 Haiku omitted the second event where the dragon was awakened, and the sequence of events becomes: *The dragon is*

*on an year long sleep* → *He chases the prince*, creating a clear contradiction. For contemporary adaptation, we identified issues where the models would fail to account for believability of certain plot elements in different settings. For instance, if the original fairytale had a horse talking to its owner, having the event play out identically in a modern setting without any reaction from any of the characters creates an inconsistency with the established setting of the story (*impossible event*). Additional examples are presented in Appendix §A.9.

## 7  Related Work

**Narrative Understanding and Reasoning Tasks.**  Narrative understanding tasks can be categorized as *descriptive* or *interpretive*.  Descriptive tasks, which involve understanding explicitly stated plot elements, include question answering benchmarks (NarrativeQA (Kočiský et al., 2018), FairyTaleQA (Xu et al., 2022), and BookQA (Angelidis et al., 2019)), narrative summarization (Ouyang et al., 2017; Papalampidi et al., 2020; Kryscinski et al., 2022), and claim verification (Karpinska et al., 2024). Interpretive tasks require forming mental representation of story's worlds and utilzing those to infer their logical implications, such as selecting correct endings (Mostafazadeh et al., 2016), assessing causality (Roemmele et al., 2011), or generating counterfactuals (Qin et al., 2019). However, unlike FLAWEDFICTIONS, these datasets focus on very short stories that are roughly 4 to 5 sentences long.  While, MuSR (Sprague et al., 2024) introduced multi-step reasoning over narratives involving tasks like solving murder mysteries, it uses synthetic stories with specific templates, whereas FLAWEDFICTIONS comprises edited versions of human-written stories with diverse narrative structures.  For a review about narrative theory for narrative understanding, please refer to Piper et al. (2021).

**Evaluating Quality of LLM Generated Stories.** Studies show GPT-3-generated stories score highly on fluency and coherence compared to specifically tuned models and competitively with humans (Xie et al., 2023). However, human-written stories have been shown to exhibit more diverse narrative structures than the largely homogeneous LLM-generated stories (Tian et al., 2024). While GPT-4 stories surpass human-written ones on the Psychological Depth Scale (Harel-Canada et al., 2024), which quantifies the emotion, empathy, and engagement in stories, they score lower on the Creativity Index (Lu et al., 2025), which measures linguistic creativity by searching for verbatim matches against web documents. None of these measure the logical and motivational consistency of narratives and there is evidence (Mirowski et al., 2023) that LLM authored stories can lack plot and character consistency.

**Plot Holes and Impossible Worlds.** Plot holes are inadvertent inconsistencies in a story's logical and motivational texture (Ryan, 2009). Lewis (1978) defines such stories where the plot contradicts itself as impossible fictions, citing the example of contradicting locations of Watson's old war wound in *Sherlock Holmes*. Lewis (1978) proposes resolutions of truth in such fictions by considering revisions that remain close to the original. Badura & Berto (2019) extends this theory with "impossible worlds" that can contain logical contradictions without rendering everything vacuously true to make sense of stories that deliberately defy logic (Priest, 1997). Plot holes have also been discussed in mathematics education contexts (Miežys, 2023).

**Automatic Detection of Plot Holes.** Davids (2022) introduced a symbolic approach using epistemic logic to identify plot holes, though the approach requires structured story events and is not flexible to operate on any story. Chadalapaka et al. (2023) generate synthetic data for plot hole detection by negating a randomly sampled statement in the story. However, this approach may not consistently generate plot holes, and to the best of our knowledge the authors do not perform human verification for their generated data.

## 8  Conclusion

In this work, we introduced FLAWEDFICTIONSMAKER, an algorithm for automatically generating continuity errors in stories, which we utilized to curate a benchmark FLAWEDFICTIONS for evaluating LLMs' capabilities to reason about plot holes in stories. Our experiments reveal that frontier LLMs struggle to accurately solve the task and inference time scaling pro-

vides minimal performance improvements. Finally, employing the best-performing model on FLAWEDFICTIONS, we analyzed LLM generated stories and summaries, and found them to contain significantly higher continuity error rates compared to human authored stories. Overall, our work demonstrates that despite significant progress in reasoning capabilities of LLMs, substantial gaps remain in their deeper narrative understanding capabilities.

While FLAWEDFICTIONSMAKER offers a general approach for generating continuity errors, future work could explore methods providing finer control over the types and complexity of introduced plot holes. Additional research might focus on designing new post-training strategies that can enhance model performance on FLAWEDFICTIONS. Another promising direction would be to investigate whether using FLAWEDFICTIONSMAKER to generate large amounts of synthetic training data could enhance LLMs' reasoning capabilities more broadly. Future work can also consider plot deficiencies other than plot holes, like plot conveniences or coincidences (termed *cheap plot tricks* Ryan (2009)) or apply similar approaches to non-fictional contexts like fact-checking, misinformation detection, and education.

## Acknowledgments

We thank Maria Antoniak for her feedback on the initial project idea. We would also like to thank Alexander Spangher for his detailed and helpful comments on our draft. Finally, special thanks to all the Prolific annotators and UW undergraduates who participated in our annotation and evaluation studies, and whose hard work made the FLAWEDFICTIONS benchmark possible.

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

# A Appendix

## Table of Contents

## A.1 A More Formal Treatment of Continuity Errors

We discussed in §2 that the Definition 2.1 fails to account for implicit knowledge such as our world understanding and beliefs that are often essential to reason about contradictions in stories. We utilize the Possible Worlds theory from Lewis (1978) to extend our definition. The core contribution of Lewis's theory is to assess truthfulness of the statements that are never stated in the text of the narrative. E.g. can we say that *Sherlock lived closer to Paddington Station than Waterloo Station*? While using a map of real world London one can check Baker Street being closer to Paddington Station, story's text never explicitly states this. However, we can still assign truth to this statement since we do not have any special reason to believe that geography of London in Sherlock Holmes is remarkably different from the real world. To decide if a proposition $p$, which is true in the belief world of the reader (or community of readers) is also true in story $f$—isTrue$(f, p)$—, without explicitly being stated in $f$, Lewis (1978) uses the notion of counterfactuals. Specifically, a proposition $p$ is non-vacuously true in $f$, when some world where $f$ is told as fact and $p$ is true, is closer to the belief world of the reader $W_b$, than any world where $f$ is told as fact and $p$ is not true. Hence, while we can consider a world where Sherlock Holmes is told as fact and London is arranged very different from the real world such that Baker Street is closer to the Waterloo Station than Paddington Station, that world will be further away from the belief world of the reader compared to a world that preserves the geography of London.

We now utilize Lewis's theory to extend our definition of continuity errors to incorporate implicit world knowledge and beliefs. We first define the operator, $\mathsf{TF} : \mathcal{P}(\Phi) \to \mathcal{P}(\Phi)$ where for any $\mathcal{F} \subseteq \Phi$, $\mathsf{TF}(\mathcal{F}) = \{p \in \mathcal{B} \mid \text{sim}(W_{\mathcal{F},p}, W_b) < \text{sim}(W_{\mathcal{F},\neg p}, W_b)\}$ where $W_b$ is the belief world of the reader and $W_{\mathcal{F},p}$ represent any closest world to $W_b$ where both $\mathcal{F}$ and $p$ are true. Here, $\Phi$ denotes the set of all possible propositions, $\mathcal{P}(\Phi)$ is its power set, $\mathcal{B} \subseteq \Phi$ is the set of true propositions in the belief world, and sim is a similarity measure between possible worlds. In other words, $\mathsf{TF}(\mathcal{F})$ operator returns the set of propositions form the belief world of the reader that can also be established to be non-vacuously in true in story $f$ with propositions $\mathcal{F}$. Using this we can rework our definition of a continuity error:

**Definition A.1 (Continuity Error with Beliefs Incorporated)** *A proposition $\phi_e$ in a story is associated with a continuity error when:*

$$isTrue(f, \mathcal{F} \setminus \{\phi_e\}) \wedge isTrue(f, \mathsf{TF}(\mathcal{F} \setminus \{\phi_e\})) \implies isTrue(f, \neg\phi_e) \tag{2}$$

*In other words, if using all the propositions in $\mathcal{F}$ except $\phi_e$, as well as the propositions from the belief world that are non-vacuously true in $f$[8], we can conclude that the negation of $\phi_e$ is true, that means $\phi_e$ represents a continuity error in $f$.*

According to the possible worlds theory, stories $f$ with such logical contradictions lead to impossible fictions, where there exists no possible world where the story is told as fact, i.e. $\mathcal{W}_f = \{\}$. In principle, for such impossible story, any statement $p$ is vacuously true. However, such a treatment can be too harsh especially when the logical contradictions are accidental and not blatantly renders the plot useless (e.g. we can still make sense of a story even if a wound placement on a character has changed without notice). There are formalizations to non-vacuously evaluate truth statements in impossible worlds in Lewis (1978) and follow-up work Alber (2019); Badura & Berto (2019), however that falls out of the scope of this work. Our primary concern here is understanding if LLMs can reason when a story represents worlds that are impossible.

Below we provide a detailed example to further clarify the definition:

*Consider this story:*

- Jack is drinking hot tea in a cup
- Jack gives his tea to Jill
- Jill drinks the cold tea from Jack's cup

---

[8]Here, $f$ is a story $f'$ where $\phi_e$ is never stated.

From the story we can extract propositions: $\phi_1 =$ "Jack drinks tea", $\phi_2 =$"tea is hot", $\phi_3 =$ "Jack gives tea to Jill", $\phi_4 =$ "Jill drinks tea", and $\phi_5 =$ "tea is cold". Note that removing $\phi_5$ from our proposition set does not allow us to conclude $\neg\phi_5$ ("tea is not cold") from the remaining propositions alone, since "hot" does not logically entail "not cold" without additional assumptions. This is precisely why we need the extended definition with reader beliefs. A reader's commonsense knowledge includes the belief that "if tea is hot, then tea is not cold" ($\phi_2 \to \neg\phi_5$). When we include this reader belief in our reasoning, we can derive the contradiction: $\{\phi_2 \to \neg\phi_5, \phi_2\} \vdash \neg\phi_5$ (by modus ponens) This shows that $\phi_5$ is logically inconsistent with the rest of the story when combined with reasonable reader beliefs, making it a continuity error according to our extended definition. The key insight is that continuity errors in fiction cannot be detected purely from explicit textual propositions—they require the integration of implicit world knowledge that readers bring to their understanding of the story.

| Type of Plot Hole | Film / Story | Plot Hole Description | Harmless or Unbridge­able | Source | Notes |
|---|---|---|---|---|---|
| Continuity Error | Sherlock Holmes by Sir Arthur Conan Doyle | When we are first introduced to Watson in A study in pink, he is described as having injury in his left arm, but the very next story A sign of Four contradicts this where his war wound is on his knee. | Harmless | Lewis (1978) | |
| | Citizen Kane (1941) | In the film Kane dies alone, but a group of reporters are trying to discover meaning of his dyning words. If he died alone who heard the words Rosebud? | Harmless | Ryan (2009) | Example of incorpo­rating real world beliefs to reason about plot holes – "when people die alone that means no one could hear their last words" is a propo­sition we know to be true from our common­sense and not something stated in the story |

Table 2 – continued from previous page

| Type of Plot Hole | Film / Story | Plot Hole Description | Harmless or Unbridge-able | Source | Notes |
|---|---|---|---|---|---|
| Out of Character Behavior | Little Red Riding Hood by Brothers Grimm | A mother tells her daughter, Little Red Riding Hood, to go through the forest and to bring some food to her ailing grandmother. She warns the little girl not to talk to strangers. On her way, Little Red Riding Hood meets a hungry wolf and tells him about her mission. The wolf runs to the grandmother's house, eats her, and takes her place in bed. When Little Red Riding Hood arrives she mistakes the wolf for the grandmother. After a conversation during which he pretends to be the grandmother, the wolf jumps out of the bed and eats Little Red Riding Hood. Why did he not just eat her when they met for the first time? | Unbridgeable | Ryan (2009) | |
| Factual Error | Titanic (1997) | In Titanic, Jack mentions fishing at Lake Wissota which is a man-made lake created in 1917 five years later when titanic sank | Harmless | | |
| Impossible Event | Dark Knight Rises (2012) | In The Dark Knight Rises (2012), a full team of police members was trapped underground for months, yet they all walk out cleanshaven and well-dressed. | Harmless | Davids (2022) | |
| Unresolved Storylines | Game of Thrones (2011-2019) | Many plot lines in the tv show were never resolved like the mysterious character of Quaithe who makes multiple prophecies that never end up playing out in the story. | Harmless | | |

Table 2: Examples of different types of Plot Holes

## A.2 Human Annotation and Benchmarking

**Verifying stories from FLAWEDFICTIONSMAKER** The annotators were hired from the *Prolific*[9] platform with the screening conditions that the candidates have English as their primary language, are residents of UK, US, or Canada, have at least an undergraduate degree, and face no literary difficulties. We also conducted a screening test where candidates were given a small set of examples from the task for which the ground truths were already verified by the authors and selected candidates for the actual study who performed well on this screening test. The selected examples had 50% samples that were incorrectly assessed by ChatGPT and we made use of this to find candidates who were

---

[9]https://app.prolific.com/

potententially using LLMs for annotations. We also checked the average amount of time it took for participants to complete the pilot study, and didn't consider those who solved the task too quickly, with the risk of them potentially using LLMs. We finally ended up recruiting 19 annotators, who were paid $12 per hour for their work with extra 20-30% bonuses each time they annotated more than 10 stories. Estimated time per annotation for each example was 5 minutes and we ended up paying a total of $6500 to the annotators. We got roughly 350 stories annotated, and got at least 3 annotations for each story. An example of our annotation framework built using *Argilla*[10] is provided in Figure 3.

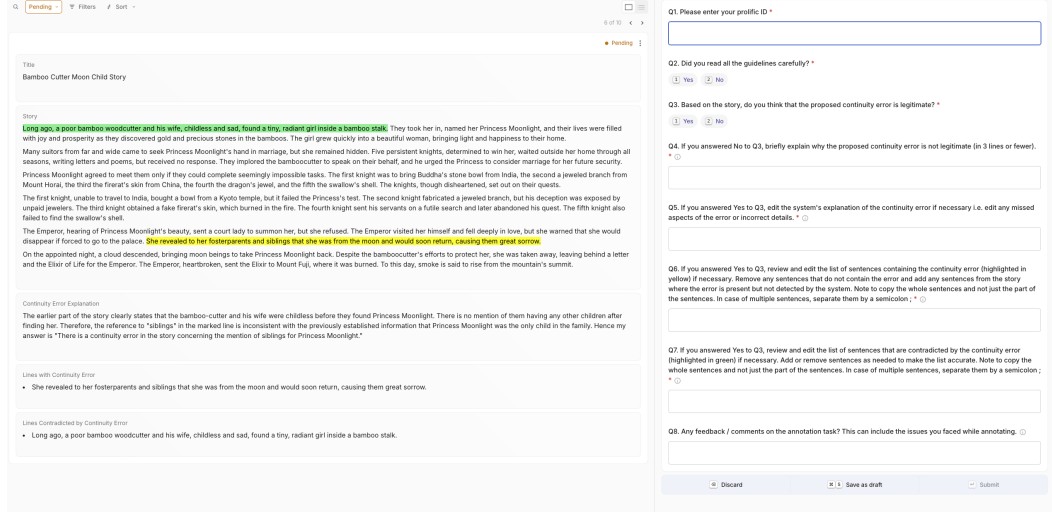

Figure 3: An example of our human annotation interface for verifying outputs of FLAWED-FICTIONSMAKER.

**Benchmarking Human Performance.**   We recruited 9 undergraduates with English major and present them with the same task of plot hole detection and the same specifications and instructions as we do for different LLMs. We sampled 50 examples from our dataset and obtained 3 responses for each instance. The estimated time for solving each task was 15 minutes (approximated by the first author) and participants were compensated $5 for providing response for each story, thereby providing $20 per hour for their work. To encourage participants to give their best efforts towards solving the task, we provide a 30% bonus for solving the task with higher accuracy ($> 70\%$ accuracy on the classification task). We paid a total of $944.60 to the participants. An example of the interface has been provided in Figure 4. The complete study document shared with the participants is included at the end of this paper §A.11.

For both of the annotation exercises, we got an Institutional Review Board (IRB) exemption from the Human Subjects Division which determined our studies as human subjects research that qualifies for exempt status (Category 101) and the IRB IDs were STUDY00021942 and STUDY00022395 for the verification and benchmarking studies respectively.

## A.3   Dataset Statistics.

Descriptive statistics of lengths of the stories included in FLAWEDFICTIONS and FLAWEDFICTIONS-LONG are provided in Tables 3 and 4 respectively.

## A.4   More Details on Experimental Setup

For all experiments, we use a temperature of 0.5 and specify a maximum of 4096 tokens for all models except the reasoning models o1, o3-mini, and Claude 3.7 Sonnet with extended thinking, for which we use a maximum of 8192 tokens. All experiments with open weights models were run on single A40 and L40 instances. We experiment with three types of prompting strategies, the vanilla case where we describe the task and output format to the model and ask it to generate the answer, few-shot case where we provide everything from the vanilla case plus two examples (one positive and one negative)

---

[10]https://github.com/argilla-io/argilla

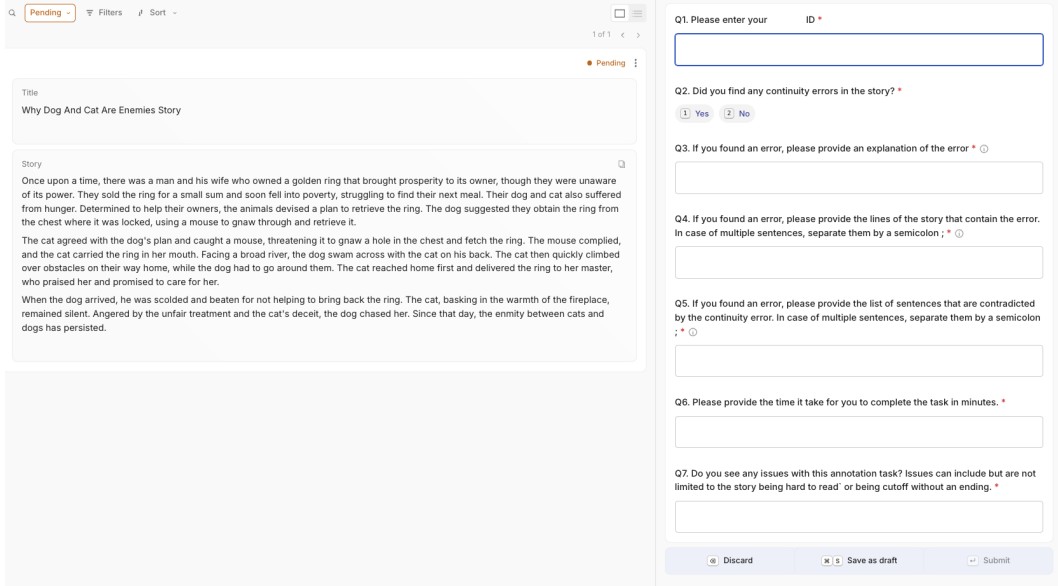

Figure 4: An example of the interface used for benchmarking human performance on FLAWEDFICTIONS.

| Statistic | Value |
|---|---:|
| Count | 414 |
| Mean | 731.81 |
| Standard Deviation | 225.51 |
| Minimum | 132 |
| 25th Percentile | 569.25 |
| Median | 754 |
| 75th Percentile | 923.50 |
| Maximum | 1236 |

Table 3: Descriptive statistics of story lengths (in words) in our FLAWEDFICTIONS.

of the task, and finally chain-of-thought prompting which builds upon the vanilla case by asking the model to first create a scratchpad analyzing the story. The prompts that we use for evaluation are provided in §A.10.2.

**Verifier** We augment the plot hole detection model i.e. *generator* with a *verifier* model (Cobbe et al., 2021) that validates if the plot hole detected by the generator is legitimate. If it is deemed illegitimate, we sample from the generator again, till either the verifier agrees or generator answers by saying *No continuity error detected*. The maximum number of samples from the generator are capped at 5. For the verifier we use Claude 3.5 Sonnet model prompted to test the validity of a proposed plot hole. Due to increased cost with using a verifier we only report results when Claude 3.5 Sonnet generator is augmented with the verifier.

## A.5 Additional Results.

### A.5.1 Detailed Results on FLAWEDFICTIONS and FLAWEDFICTIONSLONG.

We provide expanded versions of the results in the main paper (Tables 1a, 1b) containing multiple evaluation metrics and prompting methods in Tables 5 and 6. CEEval-Pos metric is defined by only considering positive examples i.e. the ones with continuity error during the localization task. Figure 5 plots performance of different models vs the average number of completion tokens generated by the model to solve the task, which we use as a proxy for inference time compute.

| Statistic | Value |
|---|---|
| Count | 200 |
| Mean | 2703.09 |
| Standard Deviation | 805.16 |
| Minimum | 1246 |
| 25th Percentile | 1965 |
| Median | 2575 |
| 75th Percentile | 3350 |
| Maximum | 3999 |

Table 4: Descriptive statistics of story lengths (in words) in our FLAWEDFICTIONSLONG.

**Effect of different prompting methods.** We find few-shot prompting often leads to worse performance compared to vanilla prompting and chain-of-thought, with the exceptions on Claude 3.5 Haiku and GPT-4-turbo, where it helps slightly. Chain-of-thought is effective for GPT-4o and GPT-4o-mini, but offers little to no improvements for other models.

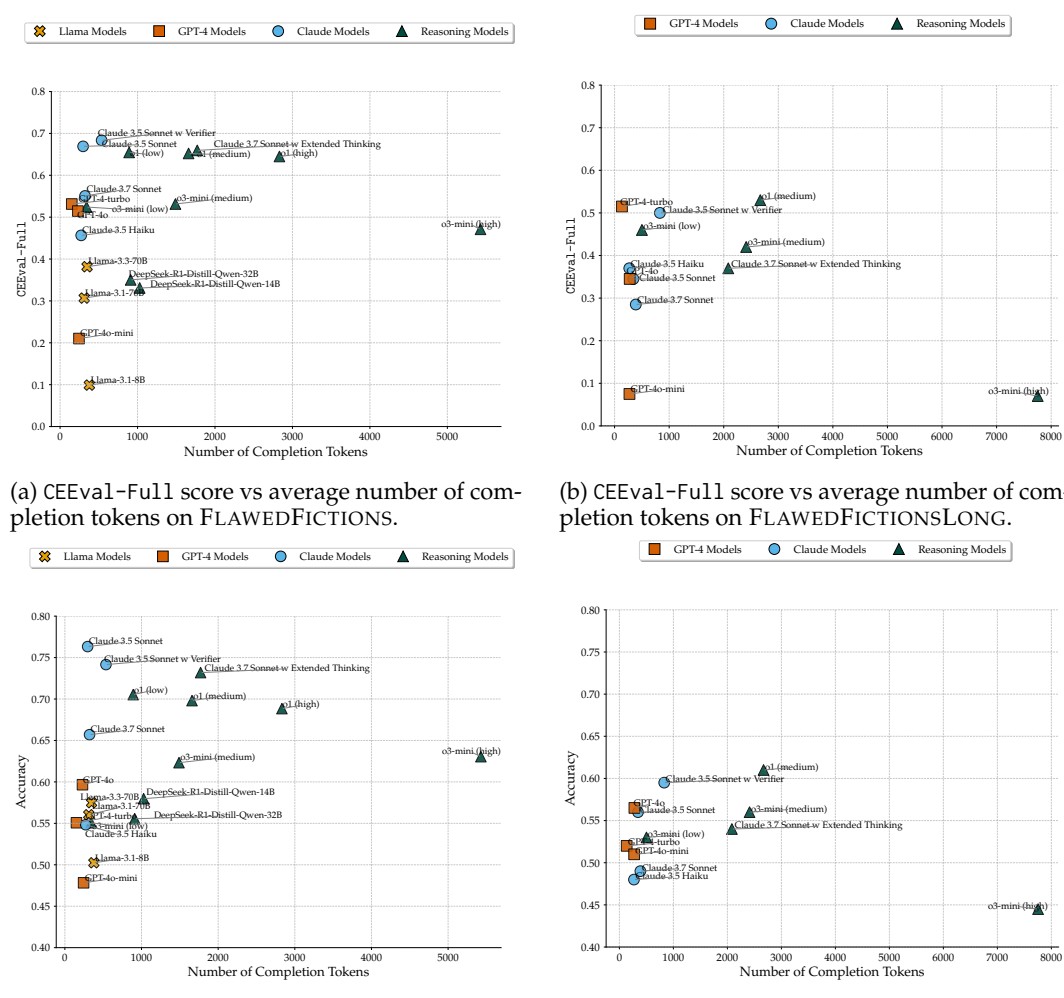

(a) CEEval-Full score vs average number of completion tokens on FLAWEDFICTIONS.

(b) CEEval-Full score vs average number of completion tokens on FLAWEDFICTIONSLONG.

(c) Accuracy score vs average number of completion tokens on FLAWEDFICTIONS.

(d) Acurracy score vs average number of completion tokens on FLAWEDFICTIONSLONG.

Figure 5: Effect of inference time compute represented using the average number of completion tokens on the performance on FLAWEDFICTIONS and FLAWEDFICTIONSLONG.

| Model | Classification Task | | | | Localization Task | |
| --- | --- | --- | --- | --- | --- | --- |
| | Accuracy | Precision | Recall | F1-score | CEEval-Pos | CEEval-Full |
| Random Baseline | 0.50 | 0.50 | 0.50 | 0.50 | 0.00 | 0.00 |
| Always No Error Baseline | 0.50 | 0.0 | 0.0 | 0.0 | 0.0 | 0.50 |
| Entailment Baseline | 0.53 | 0.52 | **1.00** | 0.68 | 0.02 | 0.04 |
| Llama-3.3-70B | 0.57 | 0.56 | 0.73 | 0.63 | 0.34 | 0.38 |
| Llama-3.1-70B | 0.56 | 0.54 | 0.76 | 0.63 | 0.26 | 0.31 |
| Llama-3.1-8B | 0.50 | 0.50 | 0.99 | 0.66 | 0.18 | 0.10 |
| DeepSeek-R1-Qwen-32B[‡] | 0.56 | 0.54 | 0.69 | 0.61 | 0.28 | 0.35 |
| DeepSeek-R1-Qwen-14B[‡] | 0.58 | 0.57 | 0.65 | 0.61 | 0.15 | 0.33 |
| Qwen2.5-32B | 0.53 | 0.53 | 0.50 | 0.51 | 0.08 | 0.31 |
| GPT-4o | 0.60 | 0.62 | 0.51 | 0.56 | 0.34 | 0.51 |
| (with Few-Shot) | 0.57 | 0.55 | 0.80 | 0.65 | 0.43 | 0.38 |
| (with CoT) | 0.64 | 0.72 | 0.45 | 0.56 | 0.33 | 0.58 |
| GPT-4o-mini | 0.48 | 0.48 | 0.62 | 0.54 | 0.09 | 0.21 |
| (with Few-Shot) | 0.50 | 0.50 | 0.90 | 0.64 | 0.13 | 0.11 |
| (with CoT) | 0.53 | 0.53 | 0.52 | 0.52 | 0.10 | 0.32 |
| GPT-4-turbo | 0.55 | 0.86 | 0.12 | 0.21 | 0.08 | 0.53 |
| (with Few-Shot) | 0.60 | 0.78 | 0.27 | 0.40 | 0.18 | 0.55 |
| (with CoT) | 0.57 | 0.90 | 0.17 | 0.28 | 0.13 | 0.55 |
| o1[‡] (Low) | 0.71 | 0.93 | 0.44 | 0.60 | 0.34 | 0.65 |
| (Medium) | 0.70 | **0.96** | 0.42 | 0.58 | 0.32 | 0.65 |
| (High) | 0.69 | 0.94 | 0.40 | 0.56 | 0.31 | 0.64 |
| o3-mini[‡] (Low) | 0.55 | 0.71 | 0.17 | 0.27 | 0.12 | 0.52 |
| (Medium) | 0.62 | 0.75 | 0.37 | 0.50 | 0.19 | 0.53 |
| (High) | 0.63 | 0.65 | 0.57 | 0.61 | 0.25 | 0.47 |
| Claude 3.5 Haiku | 0.55 | 0.59 | 0.30 | 0.40 | 0.12 | 0.46 |
| (with Few-Shot) | 0.57 | 0.72 | 0.23 | 0.35 | 0.11 | 0.51 |
| (with CoT) | 0.57 | 0.64 | 0.35 | 0.45 | 0.13 | 0.46 |
| Claude 3.5 Sonnet | **0.76** | 0.73 | 0.83 | 0.78 | 0.64 | 0.67 |
| (with Few-Shot) | 0.58 | 0.54 | 0.96 | 0.69 | 0.66 | 0.42 |
| (with CoT) | 0.71 | 0.66 | 0.87 | 0.75 | 0.64 | 0.59 |
| (with Verifier) | 0.74 | 0.81 | 0.63 | 0.71 | 0.51 | **0.68** |
| Claude 3.7 Sonnet | 0.66 | 0.61 | 0.88 | 0.72 | 0.67 | 0.55 |
| (with Extended Thinking)[‡] | 0.73 | 0.68 | 0.87 | **0.76** | **0.72** | 0.66 |
| Human Performance | **0.76** | 0.84 | 0.64 | 0.73 | 0.48 | **0.68** |

Table 5: Performance comparison of different models on the FLAWEDFICTIONS. Models trained to use test-time compute for reasoning i.e. reasoning models are marked with [‡].

| Model | Classification Task | | | | Localization Task | |
|---|---|---|---|---|---|---|
| | Accuracy | Precision | Recall | F1-score | CEEval-Pos | CEEval-Full |
| Random Baseline | 0.50 | 0.50 | 0.50 | 0.50 | 0.00 | 0.00 |
| Always No Error Baseline | 0.51 | 0.0 | 0.0 | 0.0 | 0.0 | 0.51 |
| Entailment Baseline | 0.48 | 0.48 | **1.00** | 0.65 | 0.00 | 0.00 |
| Llama-3.3-70B | 0.53 | 0.50 | 0.88 | 0.64 | 0.13 | 0.16 |
| Llama-3.1-70B | 0.53 | 0.51 | 0.88 | 0.64 | 0.06 | 0.13 |
| Llama-3.1-8B | 0.48 | 0.48 | 0.99 | 0.65 | 0.04 | 0.02 |
| DeepSeek-R1-Qwen-32B[‡] | 0.52 | 0.51 | 0.56 | 0.53 | 0.03 | 0.27 |
| DeepSeek-R1-Qwen-14B[‡] | 0.50 | 0.48 | 0.42 | 0.45 | 0.0 | 0.3 |
| Qwen2.5-32B | 0.51 | 0.49 | 0.62 | 0.55 | 0.03 | 0.23 |
| GPT-4o | 0.57 | 0.54 | 0.72 | 0.62 | 0.27 | 0.35 |
| (with CoT) | 0.56 | 0.55 | 0.48 | 0.51 | 0.21 | 0.42 |
| GPT-4o-mini | 0.51 | 0.50 | **0.93** | **0.65** | 0.03 | 0.08 |
| (with CoT) | 0.43 | 0.43 | 0.51 | 0.46 | 0.05 | 0.20 |
| GPT-4-turbo | 0.52 | **1.00** | 0.01 | 0.02 | 0.00 | 0.52 |
| (with CoT) | 0.54 | **1.00** | 0.06 | 0.12 | 0.03 | 0.53 |
| o1 (Medium) | **0.61** | 0.76 | 0.29 | 0.42 | 0.12 | **0.53** |
| o3-mini (Low) | 0.53 | 0.55 | 0.16 | 0.25 | 0.02 | 0.46 |
| (Medium) | 0.56 | 0.57 | 0.37 | 0.45 | 0.08 | 0.42 |
| (High) | 0.45 | 0.46 | 0.84 | 0.59 | 0.06 | 0.07 |
| Claude 3.5 Haiku | 0.48 | 0.44 | 0.25 | 0.32 | 0.02 | 0.37 |
| Claude 3.5 Sonnet | 0.56 | 0.53 | 0.77 | 0.63 | 0.33 | 0.35 |
| (with Verifier) | 0.60 | 0.60 | 0.49 | 0.54 | 0.30 | 0.50 |
| Claude 3.7 Sonnet | 0.49 | 0.49 | 0.90 | 0.63 | **0.47** | 0.29 |
| (with Extended Thinking) | 0.54 | 0.52 | 0.81 | 0.63 | 0.46 | 0.37 |

Table 6: Performance comparison of different models on FLAWEDFICTIONSLONG.

### A.5.2 Factors Effecting Performance on FLAWEDFICTIONS

We investigate if length of a story has an effect on how accurately do different LLMs detect continuity errors in them by measuring correlation [11] between a story's length (measured by counting number of words) and the CEEval-Full score on that story. We find negative correlation coefficients for all the models that we test and while the correlation values are low -0.1 to -0.2, for 13 out of 14 models the correlation observed is statistically significant (p-value < 0.05). Refer to the Table 7 for the exact values.

| Model | Correlation | p-value |
|---|---|---|
| Llama-3.1-8B-Instruct | -0.134* | $6.21 \times 10^{-3}$ |
| Llama-3.1-70B-Instruct | -0.154* | $1.64 \times 10^{-3}$ |
| Llama-3.3-70B-Instruct | -0.147* | $2.57 \times 10^{-3}$ |
| DeepSeek-R1-Qwen-14B | -0.192* | $7.77 \times 10^{-5}$ |
| DeepSeek-R1-Qwen-32B | -0.116* | $1.75 \times 10^{-2}$ |
| Qwen-2.5-14B | -0.127* | $9.39 \times 10^{-3}$ |
| GPT-4o-mini | -0.029 | 0.551 |
| GPT-4o | -0.196* | $5.70 \times 10^{-5}$ |
| Claude-3.5-Sonnet | -0.172* | $4.24 \times 10^{-4}$ |
| Claude-3.5-Sonnet with verifier | -0.163* | $8.42 \times 10^{-4}$ |
| Claude-3.5-Haiku | -0.156* | $1.40 \times 10^{-3}$ |
| Claude-3.7-Sonnet | -0.122* | $4.36 \times 10^{-4}$ |
| o1 | -0.104* | $2.48 \times 10^{-4}$ |
| o3-mini | -0.174* | $5.82 \times 10^{-10}$ |

Table 7: Point-Biserial Correlation between number of words in a story and the corresponding CEEval-Full scores by different LLMs.

| Model | Total Annotated | Total Accepted | Acceptance Rate |
|---|---|---|---|
| GPT-4o-mini | 54 | 2 | 0.04 |
| GPT-4o | 37 | 3 | 0.08 |
| Claude 3.5 Sonnet | 37 | 8 | 0.22 |
| o3-mini | 17 | 4 | 0.23 |

Table 8: False positive Acceptance Rates for different models.

### A.5.3 Task Subjectivity.

FLAWEDFICTIONS only consists of a single ground-truth for each story. What if the models genuinely find a plot hole in an existing story, which was simply not part of our dataset? To check if this can be the case, we run human verifications over the original stories (that we considered negative examples) with positive predictions by different models (what we call as false-positives). We ask humans to perform the same verification task, where they evaluate if the predicted error is legitimate or not. We define the acceptance rate of these false positives as the fraction of instances where the majority of the human annotators agree that the proposed error by the model is legitimate. We provide the acceptance rates in Table 8 and find that a large fraction of false positives are also deemed as such by human annotators. o3-mini has the highest acceptance rate of 23%, followed by Claude 3.5 Sonnet at 22%. To ensure more reliable evaluation, these examples were excluded from the benchmark while reporting the final scores.

## A.6 Other Considerations for Negative Examples.

As discussed in the main text, we consider original stories as negative examples i.e. instances without a plot hole in them, while curating FLAWEDFICTIONS. One potential issue with such an approach is that models might use their parametric knowledge or retrieval to determine if a story is unaltered and use that confounder to assess the presence of plot holes induced by FLAWEDFICTIONSMAKER.

---

[11]We use Point-Biserial Correlation since CEEval-Full at an instance level is a discrete i.e. 0 or 1.

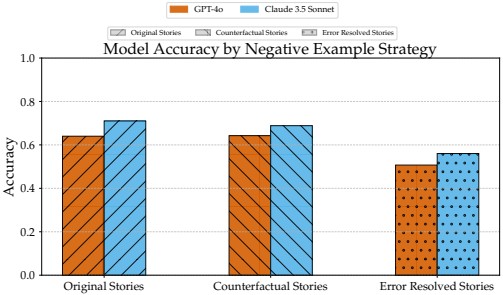
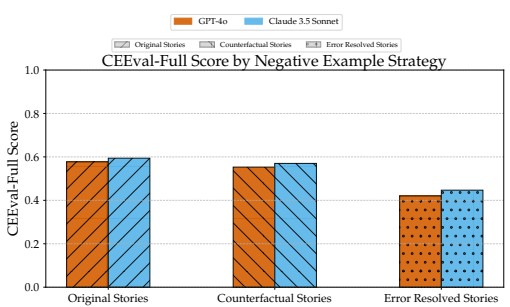

(a) Model accuracy across different negative example strategies.

(b) CEEval-Full scores across different negative example strategies.

Figure 6: Performance comparison of GPT-4o and Claude 3.5 Sonnet across different strategies to choose negative example. The plots show (a) model accuracy and (b) CEEval-Full scores for three types of negative examples: original stories with inconsistencies, counterfactual stories where details have been changed, and stories where inconsistencies were resolved.

To circumvent this issue, we explored other approaches for selecting negative examples that utilized partial-synthetic data. First, we considered using *counterfactual stories* generated in Step 3 of our pipeline as negative examples. We also considered, another approach which would use the positive examples generated by FLAWEDFICTIONSMAKER and prompt GPT-4o model with the story and the continuity error and ask it to add extra context in the story that resolves the error – *error resolved stories*. While both of this approaches would ensure that both positive and negative examples in our dataset are partially synthetic, validating them can prove to be non-trivial. Remember for positive stories, we were able to get human verification done, because we had a proposed error for each story and human annotators checked for legitimacy of such errors. For counterfactual and error resolved stories, we wouldn't have continuity error proposals, and asking humans to check for any continuity errors in the stories can be highly cognitively demanding.

Since both approaches are prone to errors, human validation would have been necessary for creating a high quality benchmark, and hence we decided to stick with original stories for this work. Further, our results, especially on FLAWEDFICTIONSLONG suggest that models are not really using any confounder to solve the task, as models tend to generate false positives quite often, indicated by their low precisions (see Tables 5, 6).

However, we do release the two alternate splits of FLAWEDFICTIONS – FLAWEDFICTIONSCOUNTER-FACTNEGS consisting of counterfactual stories as negative examples and FLAWEDFICTIONSRESOLVED-NEGS that consists of error resolved stories as negatives. Both of these splits have 414 examples like the original dataset and share the same positive examples. We benchmark and compare GPT-4o and Claude 3.5 Sonnet on these splits and provide results in Figure 6. Both models show similar performance on original split and FLAWEDFICTIONSCOUNTERFACTNEGS, however the performance is much lower on FLAWEDFICTIONSRESOLVEDNEGS. Future work can explore ways to efficiently validate negative examples generated through these strategies.

## A.7 FLAWEDFICTIONS Examples

Below we provide a few positive examples (i.e. the ones with continuity errors) included in FLAWED-FICTIONS and generated using FLAWEDFICTIONSMAKER. The lines containing the continuity errors are highlighted with yellow color, while the ones that contain the fact being contradicted are highlighted with green color.

**Story**

In the times when we used to travel by canal I was coming down from Dublin. When we came to Mullingar the canal ended, and I began to walk, and stiff and fatigued I was after the slowness. I had some friends with me, and now and then we walked, now and then we rode in a cart. So on till we saw some girls milking a cow, and stopped to joke with them. After a while we asked them for a drink of milk. 'We have nothing to put it in here,' they said, 'but come to the house with us.' We went home with them and sat round the fire talking. After a while the others went, and left me, loath to stir from the good fire. I asked the girls for something to eat. There was a pot on the fire, and they took the meat out and put it on a plate and told me to eat only the meat that came from the head. When I had eaten, the girls went out and I did not see them again.

It grew darker and darker, and there I still sat, loath as ever to leave the good fire; and after a while two men came in, carrying between them a corpse. When I saw them, the girls saw my fear and stayed close by. Says one to the other, 'Who'll turn the spit?' Says the other, 'Michael Hart, come out of that and turn the meat!' I came out in a tremble and began turning the spit. 'Michael Hart,' says the one who spoke first, 'if you let it burn we will have to put you on the spit instead,' and on that they went out. I sat there trembling and turning the corpse until midnight. The men came again, and the one said it was burnt, and the other said it was done right, but having fallen out over it, they both said they would do me no harm that time; and sitting by the fire one of them cried out, 'Michael Hart, can you tell a story?' 'Never a one,' said I. On that he caught me by the shoulders and put me out like a shot. The girls followed me out, their faces filled with concern.

It was a wild, blowing night; never in all my born days did I see such a night–the darkest night that ever came out of the heavens. I did not know where I was for the life of me. So when one of the men came after me and touched me on the shoulder with a 'Michael Hart, can you tell a story now?'–'I can,' says I. In he brought me, and, putting me by the fire, says 'Begin.' 'I have no story but the one,' says I, 'that I was sitting here, and that you two men brought in a corpse and put it on the spit and set me turning it.' 'That will do,' says he; 'you may go in there and lie down on the bed.' And in I went, nothing loath, and in the morning where was I but in the middle of a green field. The girls were nowhere to be seen, and I wondered if they had been part of the strange night's events or just silent witnesses.

**Continuity Error Explanation**

The story clearly establishes that after the protagonist ate, the girls left and he "did not see them again." However, the subsequent marked lines show the girls present during later events - when the men bring in the corpse, when the protagonist is thrown out, and even a final reflection about their presence. This creates a direct logical contradiction as the girls cannot both be gone (never to be seen again) and present during these later events.

**Story**

Along the straight, glistening road, through a dim arcade of drooping trees, a tunnel of faded green and gold, dripping with the misty rain of a late October afternoon, a human tide was flowing, not swiftly, but slowly, with the patient, pathetic slowness of weary feet, and numb brains, and heavy hearts.

Yet they were in haste, all of these old men and women, fathers and mothers, and little children; they were flying as fast as they could; either away from something that they feared, or toward something that they desired.

That was the strange thing–the tide on the road flowed in two directions.

Some fled away from ruined homes to escape the perils of war. Some fled back to ruined homes to escape the desolation of exile. But all were fugitives, anxious to be gone, striving along the road one way or the other, and making no more speed than a creeping snail's pace of unutterable fatigue. I saw many separate things in the tide, and remembered them without noting.

A boy straining to push a wheelbarrow with his pale mother in it, and his two little sisters trudging at his side. A peasant with his two girls driving their lean, dejected cows back to some unknown pasture. A bony horse tugging at a wagon heaped high with bedding and household gear, on top of which sat the wrinkled grandmother with the tiniest baby in her arms, while the rest of the family stumbled alongside–and the cat was curled up on the softest coverlet in the wagon. Two panting dogs, with red tongues hanging out, and splayed feet clawing the road, tugging a heavy-laden cart while the master pushed behind and the woman pulled in the shafts. Strange, antique vehicles crammed with passengers. Couples and groups and sometimes larger companies of foot-travellers. Now and then a solitary man or woman, old and shabby, bundle on back, eyes on the road, plodding through the mud and the morning mist, under the high archway of blooming branches.

All these distinct pictures I saw, yet it was all one vision–a vision of humanity with its dumb companions in flight–infinitely slow, painful, pitiful flight!

I saw no tears, I heard no cries of complaint. But beneath the numb and patient haste on all those dazed faces I saw a question.

*"What have we done? Why has this thing come upon us and our children?"*

Somewhere I heard a trumpet blown. The brazen spikes on the helmets of a little troop of German soldiers flashed for an instant, far down the sloppy road. Through the crisp morning air came the dull, distant booming of the unseen guns of conquest in Flanders.

That was the only answer

---

**Continuity Error Explanation** The story initially establishes the setting as a "late October afternoon," which implies an autumn setting in the afternoon. However, the marked lines introduce inconsistencies: 1. "plodding through the mud and the morning mist" - This line contradicts the established time of "afternoon" by suggesting it is morning. 2. "under the high archway of blooming branches" - This line suggests a season of blooming, typically spring, which contradicts the established autumn setting. 3. "Through the crisp morning air" - This line again suggests it is morning, contradicting the afternoon setting.

**Story**

Now, as time passed, King Arthur gathered into his Order of the Round Table knights whose peers shall never be found in any age; and foremost amongst them all was Sir Launcelot du Lac. Such was his strength that none against whom he laid lance in rest could keep the saddle, and no shield was proof against his sword dint; but for his courtesy even more than for his courage and strength, Sir Launcelot was famed far and near. Gentle he was and ever the first to rejoice in the renown of another; and in the jousts, he would avoid encounter with the young and untried knight, letting him pass to gain glory if he might.

It would take a great book to record all the famous deeds of Sir Launcelot, and all his adventures. He was of Gaul, for his father, King Ban, ruled over Benwick; and some say that his first name was Galahad, and that he was named Launcelot du Lac by the Lady of the Lake who reared him when his mother died. Early he won renown by delivering his father's people from the grim King Claudas who, for more than twenty years, had laid waste the fair land of Benwick; then, when there was peace in his own land, he passed into Britain, to Arthur's court, where the King received him gladly, and made him Knight of the Round Table and took him for his trustiest friend.

And so it was that, when Guenevere was to be brought to Canterbury, to be married to the King, Launcelot was chief of the knights sent to wait upon her, and his role as the leader in this mission was a testament to his unmatched skills and the King's reliance on his prowess. For, from the moment he saw her, Sir Launcelot loved Guenevere, for her sake remaining wifeless all his days, and in all things being her faithful knight.

But busy-bodies and mischief-makers spoke evil of Sir Launcelot and the Queen, and from their talk came the undoing of the King and the downfall of his great work. But that was after long years, and after many true knights had lived their lives, though the atmosphere at the court had grown tense with rivalries, partly fueled by Sir Launcelot's aloof demeanor and his singular pursuit of personal glory.

---

**Continuity Error Explanation** The line "though the atmosphere at the court had grown tense with rivalries, partly fueled by Sir Launcelot's aloof demeanor and his singular pursuit of personal glory" introduces a continuity error. Earlier in the story, Sir Launcelot is described as courteous, gentle, and one who rejoices in the renown of others, which contradicts the depiction of him having an aloof demeanor and a singular pursuit of personal glory. Hence my answer is "There is a continuity error in the story concerning the portrayal of Sir Launcelot's demeanor and motivations."

**Story**
PHILIP ECKERT lived for many years in an old, weather-stained wooden house about three miles from the little town of Marion, in Vermont. There must be quite a number of persons living who remember him, not unkindly, I trust, and know something of the story that I am about to tell.

"Old Man Eckert," as he was always called, was not of a sociable disposition and lived alone. As he was never known to speak of his own affairs nobody thereabout knew anything of his past, nor of his relatives if he had any. Without being particularly ungracious or repellent in manner or speech, he managed somehow to be immune to impertinent curiosity, yet exempt from the evil repute with which it commonly revenges itself when baffled; so far as I know, Mr. Eckert's renown as a reformed assassin or a retired pirate of the Spanish Main had not reached any ear in Marion. He got his living cultivating a small and not very fertile farm.

One day he disappeared and a prolonged search by his neighbors failed to turn him up or throw any light upon his whereabouts or whyabouts. Nothing indicated preparation to leave: all was as he might have left it to go to the spring for a bucket of water. For months, the community was abuzz, with everyone from old friends to casual acquaintances chiming in with theories and concerns, all colored by the personal stories Eckert had shared over the years. Then "old man Eckert" became a village tale for the ear of the stranger. I do not know what was done regarding his property—the correct legal thing, doubtless. The house was standing, still vacant and conspicuously unfit, when I last heard of it, some twenty years afterward.
.... [Rest of the story is omitted]...

---

**Continuity Error Explanation**
The marked line introduces a continuity error because it implies that Old Man Eckert had shared personal stories over the years with people in the community, which directly contradicts the earlier statements that he was not sociable and never spoke of his own affairs. The earlier lines establish him as a solitary figure who kept his past and personal life private, making it inconsistent for the community to have personal stories shared by him.

## A.8 Examples of Reasoning Errors on FLAWEDFICTIONS

---

**Story**

Once on a time there was a man up in Finnmark who had caught a great white bear, which he was going to take to the king of Denmark. Now, it so fell out, that he came to the Dovrefell just about Christmas Eve, and there he turned into a cottage where a man lived, whose name was Halvor, and asked the man if he could get house-room there, for his bear and himself.

"Heaven never help me, if what I say isn't true!" said the man; "but we can't give any one house-room just now, for every Christmas Eve such a pack of Trolls come down upon us, that we are forced to flit, and haven't so much as a house over our own heads, to say nothing of lending one to any one else."

"Oh?" said the man, "if that's all, you can very well lend me your house; my bear can lie under the stove yonder, and I can sleep in the side-room."

Well, he begged so hard, that at last he got leave to stay there; so the people of the house flitted out, and before they went, everything was got ready for the Trolls; the tables were laid, and there was rice porridge, and fish boiled in lye, and sausages, and all else that was good, just as for any other grand feast.

So, when everything was left as usual, down came the Trolls. Some were great, and some were small; some had long tails, and some had no tails at all; some, too, had long, long noses; and they looked around puzzled, not finding their usual feast. Just then one of the little Trolls caught sight of the white bear, who lay under the stove; so he took a piece of sausage from his own stash and stuck it on a fork, and went and poked it up against the bear's nose, screaming out:

"Pussy, will you have some sausage?"

Then the white bear rose up and growled, and hunted the whole pack of them out of doors, both great and small.

Next year Halvor was out in the wood, on the afternoon of Christmas Eve, cutting wood before the holidays, for he thought the Trolls would come again; and just as he was hard at work, he heard a voice in the wood calling out:

"Halvor! Halvor!"

"Well", said Halvor, "here I am."

"Have you got your big cat with you still?"

"Yes, that I have", said Halvor; "she's lying at home under the stove, and what's more, she has now got seven kittens, far bigger and fiercer than she is herself."

"Oh, then, we'll never come to see you again", bawled out the Troll away in the wood, and he kept his word; for since that time the Trolls have never eaten their Christmas brose with Halvor on the Dovrefell.

---

**Ground Truth**

The line "Some, too, had long, long noses; and they looked around puzzled, not finding their usual feast" introduces a continuity error because it contradicts the earlier statement that the people of the house had prepared everything as usual for the Trolls, including setting the tables with food. If the tables were indeed laid with rice porridge, fish boiled in lye, sausages, and other good food, the Trolls should not be puzzled about not finding their usual feast. Hence my answer is "There is a continuity error in the story concerning the Trolls being puzzled about not finding their usual feast, despite the earlier statement that everything was prepared as usual for them."

---

**Claude 3.5 Sonnet with Verifier's Prediction**

After careful analysis, I found a potential continuity error in the story regarding the ownership of the house and the bear. At the beginning of the story, the man with the white bear asks Halvor for house-room, indicating it's Halvor's house. However, in the second year, when Halvor speaks to the Troll, he refers to the bear as if it were his own pet ("she's lying at home under the stove"), despite the bear originally belonging to the traveler who was taking it to the king of Denmark. Additionally, there's no explanation of how Halvor came to own the bear or what happened to the original owner.

---

**Explanation of Reasoning Error** Model fails to understand that Halvor is bluffing about white bear being at his house. Trolls do not know that the white bear belonged to the other man and halvor takes advantage of this fact.

---

**Reasoning Error Category** *Misinterpreting character's motivations or behavior.*

**Story**
A poor workman and his wife were once almost starving. Every day the man gathered sticks and sold them, while the woman remained at home.
Wife," said the man one day, "come, we will both go and gather wood for sale to-day, so that, if we earn enough, we shall not eat rice only, but will buy a small fish, and eat that also."
The woman agreed, and having sold their stock of firewood, they returned home with a fish for dinner.
On opening it, a small white bead, round and shining, fell upon the floor. The woman was attracted by its brightness, and locked it up in a box.
Many days of poverty and hunger visited that household before the woman remembered the pretty stone found in the fish; but at last she thought of it, and took it to a Bunniah, who, as soon as he set eyes upon it, said: "Give me that, and I will give you as much as you can carry away of ghee and food and oil, not only to-day, but for many days, if you will come to my shop."
The foolish woman parted with her treasure, not knowing that it was a pearl of great value, and returned home laden with good things.
The Bunniah chuckled with delight. He was crafty, like all Bunniahs, and, you may depend, never kept his promise: such was the foolishness of the woman, and such the cunning and greed of the man.

**Ground Truth**
No continuity error found.

**GPT-4o's Prediction**
Upon analyzing the story, I found a continuity error related to the characterization of the Bunniah. Initially, the Bunniah is described as someone who makes a generous offer to the woman in exchange for the pearl. However, later in the narrative, it is stated that the Bunniah never kept his promise, which contradicts the earlier portrayal of him being willing to provide food and supplies in exchange for the pearl. This inconsistency in the Bunniah's actions and character creates a continuity error.

**Explanation of Reasoning Error** Model fails to understand that people can lie to achieve material gains. A greedy characterization for a Bunniah is also a common stereotype in old texts, which this story also plays into.

**Reasoning Error Category** *Misinterpreting character's motivations or behavior.*

**Story**

PHILIP ECKERT lived for many years in an old, weather-stained wooden house about three miles from the little town of Marion, in Vermont. There must be quite a number of persons living who remember him, not unkindly, I trust, and know something of the story that I am about to tell.

"Old Man Eckert," as he was always called, was not of a sociable disposition and lived alone. As he was never known to speak of his own affairs nobody thereabout knew anything of his past, nor of his relatives if he had any. Without being particularly ungracious or repellent in manner or speech, he managed somehow to be immune to impertinent curiosity, yet exempt from the evil repute with which it commonly revenges itself when baffled; so far as I know, Mr. Eckert's renown as a reformed assassin or a retired pirate of the Spanish Main had not reached any ear in Marion. He got his living cultivating a small and not very fertile farm.

One day he disappeared and a prolonged search by his neighbors failed to turn him up or throw any light upon his whereabouts or whyabouts. Nothing indicated preparation to leave: all was as he might have left it to go to the spring for a bucket of water. For months, the community was abuzz, with everyone from old friends to casual acquaintances chiming in with theories and concerns, all colored by the personal stories Eckert had shared over the years. Then "old man Eckert" became a village tale for the ear of the stranger. I do not know what was done regarding his property—the correct legal thing, doubtless. The house was standing, still vacant and conspicuously unfit, when I last heard of it, some twenty years afterward.

Of course it came to be considered "haunted," and the customary tales were told of moving lights, dolorous sounds and startling apparitions. At one time, about five years after the disappearance, these stories of the supernatural became so rife, or through some attesting circumstances seemed so important, that some of Marion's most serious citizens deemed it well to investigate, and to that end arranged for a night session on the premises. The parties to this undertaking were John Holcomb, an apothecary; Wilson Merle, a lawyer, and Andrus C. Palmer, the teacher of the public school, all men of consequence and repute. They were to meet at Holcomb's house at eight o'clock in the evening of the appointed day and go together to the scene of their vigil, where certain arrangements for their comfort, a provision of fuel and the like, for the season was winter, had been already made.

Palmer did not keep the engagement, and after waiting a half-hour for him the others went to the Eckert house without him. They established themselves in the principal room, before a glowing fire, and without other light than it gave, awaited events. It had been agreed to speak as little as possible: they did not even renew the exchange of views regarding the defection of Palmer, which had occupied their minds on the way.

*...[omitted]...* This much was ascertained the next morning. During the session of Messrs. Holcomb and Merle at the "haunted house" a new snow had fallen to a depth of several inches upon the old. In this snow Palmer's trail from his lodging in the village to the back door of the Eckert house was conspicuous. But there it ended: from the front door nothing led away but the tracks of the two men who swore that he preceded them. Palmer's disappearance was as complete as that of "old man Eckert" himself—whom, indeed, the editor of the local paper somewhat graphically accused of having "reached out and pulled him in." *...[omitted]...*

---

**Ground Truth**

The marked line introduces a continuity error because it implies that Old Man Eckert had shared personal stories over the years with people in the community, which directly contradicts the earlier statements that he was not sociable and never spoke of his own affairs. The earlier lines establish him as a solitary figure who kept his past and personal life private, making it inconsistent for the community to have personal stories shared by him. Hence my answer is "There is a continuity error in the story concerning the community's knowledge of personal stories shared by Old Man Eckert."

---

**Claude 3.5 Sonnet with Verifier's Prediction**

After careful analysis, I found a potential continuity error in the story regarding Palmer's behavior. The story states that Palmer did not show up for the initial meeting at Holcomb's house at 8 PM, causing the other two men to go to the Eckert house without him. However, later in the story, it's revealed that Palmer's tracks in the snow led directly to the back door of the Eckert house, indicating he went there directly from his lodging in the village. This creates a continuity error because if Palmer had intended to meet at Holcomb's house first (as was the plan), his tracks should have shown him going to Holcomb's house first, not directly to the Eckert house.

---

**Explanation of Reasoning Error** It is clearly stated that Palmer didn't show up to Holcomb's house, hence it is consistent for his tracks to lead directly to the bak door of Eckert house

---

**Reasoning Error Category** *Wrongly tracking and interpreting entities' states.*

**Story**
In a region of the country where the forest and prairie vied for beauty, lived a wicked manito disguised as an old Indian. Despite the abundance of game and resources, this evil genius sought to destroy those who fell into his hands, decimating the once populous land until he lived almost alone in the wilderness.
The manito's secret lay in his great speed and ability to transform into any fourfooted creature. He challenged people to races around a large lake, where the starting and winning post was the same. No one ever won, as he would change into a swift animal and leave his competitors behind, claiming their lives at the finish.
The entire region feared the manito, yet young men continued to race him to avoid being called cowards. The manito maintained his deadly sport with pleasing manners, visiting lodges to monitor the growth of boys and challenge them when they were ready, thinning out families and earning the hatred of all Indian mothers.
A poor widow, who had lost her husband and seven sons to the manito, lived with her daughter Minda and a young son. Despite her struggles, she taught her son to be a brave warrior and her daughter the skills of a good wife. Minda was dutiful and kind, helping her mother and enjoying the beauty of their surroundings.
*...[omitted]...* With the manito defeated, his kindred fled, and Monedowa and Minda bid farewell to the widow, transforming into beautiful birds. They flew away, their songs bringing peace to the widow, who never knew want again. Her young son, who had grown in the ways of peace and wisdom, became a comfort to her, his gentle nature bringing joy and solace to her life.

**Ground Truth** No continuity error found

**GPT-4o's Prediction**
The continuity error in the story is related to the widow's son. Initially, the story mentions that the widow lost her husband and seven sons to the manito, leaving her with only her daughter Minda and a young son. However, at the end of the story, it is mentioned that her young son grew up and brought joy and solace to her life. This implies that the son survived and grew up, which contradicts the earlier statement that the widow lost all her sons to the manito. Hence my answer is "There is a continuity error in the story concerning the widow's young son, who is mentioned to have grown up and brought joy to her life despite earlier being stated that she lost all her sons to the manito."

**Explanation of Reasoning Error** It is clearly stated that one of the sons of the widow survived, and it is that son being referred to towards the end.

**Reasoning Error Category** *Wrongly tracking and interpreting entities' states.*

**Story**
There were formerly a king and a queen, who were so sorry that they had no children; so sorry that it cannot be expressed. They went to all the waters in the world; vows, pilgrimages, all ways were tried, and all to no purpose. At last, however, the Queen had a daughter. There was a very fine christening; and the Princess had for her god-mothers all the fairies they could find in the whole kingdom (they found seven), that every one of them might give her a gift, as was the custom of fairies in those days. By this means the Princess had all the perfections imaginable.
...[omitted]...
The old Fairy's turn coming next, with a head shaking more with spite than age, she said that the Princess should have her hand pierced with a spindle and die of the wound. This terrible gift made the whole company tremble, and everybody fell a-crying. At this very instant the young Fairy came out from behind the hangings, and spake these words aloud: "Assure yourselves, O King and Queen, that your daughter shall not die of this disaster. It is true, I have no power to undo entirely what my elder has done. The Princess shall indeed pierce her hand with a spindle; but, instead of dying, she shall only fall into a profound sleep, which shall last a hundred years, at the expiration of which a king's son shall come and awake her."
The King, to avoid the misfortune foretold by the old Fairy, caused immediately proclamation to be made, whereby everybody was forbidden, on pain of death, to spin with a distaff and spindle, or to have so much as any spindle in their houses. About fifteen or sixteen years after, the King and Queen being gone to one of their houses of pleasure, the young Princess happened one day to divert herself in running up and down the palace; when going up from one apartment to another, she came into a little room on the top of the tower, where a good old woman, alone, was spinning with her spindle. This good woman had never heard of the King's proclamation against spindles.
...[omitted]...
Others, That all the sorcerers and witches of the country kept there their sabbath or night's meeting. The common opinion was: That an ogre lived there, and that he carried thither all the little children he could catch, that he might eat them up at his leisure, without anybody being able to follow him, as having himself only the power to pass through the wood. The Prince was at a stand, not knowing what to believe, when a very good countryman spake to him thus: "May it please your royal highness, it is now about fifty years since I heard from my father, who heard my grandfather say, that there was then in this castle a princess, the most beautiful was ever seen; that she must sleep there a hundred years, and should be waked by a king's son, for whom she was reserved." The young Prince was all on fire at these words, believing, without weighing the matter, that he could put an end to this rare adventure; and, pushed on by love and honor, resolved that moment to look into it.
...[omitted]...

**Ground Truth** No continuity error found

**Claude 3.5 Sonnet with Verifier's Prediction**
I found a notable continuity error in the story regarding the timeline of events. Early in the story, it's stated that the old fairy hadn't been seen for "above fifty years." Later, when the Prince comes to the castle, a countryman says it's been "about fifty years" since he heard from his father about the sleeping princess. This is impossible because the princess was supposed to sleep for a full hundred years according to the fairy's spell, and the story confirms that "a hundred years were gone and passed" when the Prince arrives.

**Explanation of Reasoning Error** Model incorrectly accounting for the passage of time. Just because 50 years passed since character's father told him about the princess doesn't imply it has been 50 years since princess was sleeping. It clearly states that the father heard from the grandfather. Grandfather could have seen the princess when he was young. There can easily be passage of 100 years between the grandfather hearing and the current timeline.

**Reasoning Error Category** *Wrongly tracking and interpreting entities' states.*

**Story**
A certain Bunniah or merchant married a woman of his own caste, and set out to a distant city. On the way he fell ill with a headache, so she sat by the wayside and pressed his head. While doing so a man passed by, and asked for a little fire to light his cheelum for a smoke, but she replied: "I cannot leave my husband, for I am holding his head while he sleeps."
"Put some clothes under his head, and he will sleep," advised the stranger. This she did, but, while giving the fire to the man, he seized her, and, placing her upon his horse, rode away. When the Bunniah awoke, it was to find himself all alone but for his faithful dog Kulloo.
"Master," said Kulloo, "let us become Fakirs, and beg from door to door." So they set out to beg, and one day came to the house of the robber who had stolen the Bunniah's wife; and she, not recognising her husband or his dog, gave them money and food. But the dog knew her, and that evening he spoke to his master, and asked him if he too had seen his wife. The Bunniah had not; and, guided by Kulloo, he set out to find her.
When they arrived at the robber's house, and made themselves known, the woman was greatly vexed, for the robber was rich, and gave her a very comfortable home; but she pretended to be friendly and invited her husband to dine there that night, telling him that, afterwards, when he had the chance, he could kill the robber.
When the Bunniah had gone, she and the robber arranged a trap for him. It was a hole in the floor, very large and deep, with spikes fixed in the sides of it, so that anybody who fell in might die. Over the hole they set a large brass thalee or plate, so that, while the Bunniah leaned heavily upon it to eat his food, both it and he would fall into the hole.
All happened as they anticipated; and when the poor Bunniah found himself in a deep hole, full of spikes, he thought his last hour had come. But faithful Kulloo came to his rescue, and, taking out the spikes with his teeth, soon set his master free.
The Bunniah then lost no time in seeking the robber, and found him lying fast asleep; so he killed him, and cut off his head, then, taking his wife with him, left the place.
Kulloo followed closely, and licked up each drop of blood which fell from the robber's head, lest it might leave a trace of the deed, and get his master into trouble. He was a wise dog, and knew the woman was wicked, so she hated him, and made up her mind that she would neither eat nor drink until he was dead.
The Bunniah enquired why she would not touch any food, and she told him she would only do so if he killed Kulloo. This the man refused to do; but, after a while, he consented. Poor Kulloo, when he knew his last hour had come, besought his master to bury him carefully, and to see that his head, which the Bunniah meant to cut off, was buried with him, for a time was yet to come when he would again save his master's life.
After Kulloo was dead and buried the wicked woman was happy, and ate and drank as before; but, after a few days, she went and gave notice at the Court that the Bunniah was a cruel robber, who had killed her husband, and stolen her away. The police seized him, and he was taken up for murder; but, just as the Judge was about to pronounce the sentence of death upon him, he remembered faithful Kulloo; and at the same moment the dog appeared!
All were surprised when he stood before the Judge, and asked leave to speak. He then told the whole story of the robber and the wicked woman; and thus, for a second time, saved his master's life, but, having said his say, poor Kulloo disappeared and was never seen again.

**Ground Truth** No continuity error found

**Claude 3.5 Sonnet's Prediction**
There is a significant continuity error in the story regarding Kulloo the dog. The story establishes that Kulloo dies and is buried when the Bunniah's wife demands his death. However, he later appears alive at the court to save his master. While the dog did request to be buried carefully with his head and mentioned he would save his master again, the story doesn't establish any magical or supernatural elements that would explain his resurrection. This creates a logical contradiction in the narrative flow.

**Explanation of Reasoning Error** It is meant to be a twist that ghost of the dog returns in the end to save the master. Fantastical elements are established in the story, like Kulloo being a speaking dog. Such setting doesn't pose any break in continuity.

**Reasoning Error Category** *Failing to understand genre conventions*

**Story**
A certain Bunniah or merchant married a woman of his own caste, and set out to a distant city. On the way he fell ill with a headache, so she sat by the wayside and pressed his head. While doing so a man passed by, and asked for a little fire to light his cheelum for a smoke, but she replied: "I cannot leave my husband, for I am holding his head while he sleeps."
"Put some clothes under his head, and he will sleep," advised the stranger. This she did, but, while giving the fire to the man, he seized her, and, placing her upon his horse, rode away. When the Bunniah awoke, it was to find himself all alone but for his faithful dog Kulloo.
"Master," said Kulloo, "let us become Fakirs, and beg from door to door." So they set out to beg, and one day came to the house of the robber who had stolen the Bunniah's wife; and she, not recognising her husband or his dog, gave them money and food. But the dog knew her, and that evening he spoke to his master, and asked him if he too had seen his wife. The Bunniah had not; and, guided by Kulloo, he set out to find her.
When they arrived at the robber's house, and made themselves known, the woman was greatly vexed, for the robber was rich, and gave her a very comfortable home; but she pretended to be friendly and invited her husband to dine there that night, telling him that, afterwards, when he had the chance, he could kill the robber.
When the Bunniah had gone, she and the robber arranged a trap for him. It was a hole in the floor, very large and deep, with spikes fixed in the sides of it, so that anybody who fell in might die. Over the hole they set a large brass thalee or plate, so that, while the Bunniah leaned heavily upon it to eat his food, both it and he would fall into the hole.
All happened as they anticipated; and when the poor Bunniah found himself in a deep hole, full of spikes, he thought his last hour had come. But faithful Kulloo came to his rescue, and, taking out the spikes with his teeth, soon set his master free.
The Bunniah then lost no time in seeking the robber, and found him lying fast asleep; so he killed him, and cut off his head, then, taking his wife with him, left the place.
Kulloo followed closely, and licked up each drop of blood which fell from the robber's head, lest it might leave a trace of the deed, and get his master into trouble. He was a wise dog, and knew the woman was wicked, so she hated him, and made up her mind that she would neither eat nor drink until he was dead.
The Bunniah enquired why she would not touch any food, and she told him she would only do so if he killed Kulloo. This the man refused to do; but, after a while, he consented. Poor Kulloo, when he knew his last hour had come, besought his master to bury him carefully, and to see that his head, which the Bunniah meant to cut off, was buried with him, for a time was yet to come when he would again save his master's life.
After Kulloo was dead and buried the wicked woman was happy, and ate and drank as before; but, after a few days, she went and gave notice at the Court that the Bunniah was a cruel robber, who had killed her husband, and stolen her away. The police seized him, and he was taken up for murder; but, just as the Judge was about to pronounce the sentence of death upon him, he remembered faithful Kulloo; and at the same moment the dog appeared!
All were surprised when he stood before the Judge, and asked leave to speak. He then told the whole story of the robber and the wicked woman; and thus, for a second time, saved his master's life, but, having said his say, poor Kulloo disappeared and was never seen again.

**Ground Truth** No continuity error found

**Claude 3.5 Sonnet's Prediction**
There is a significant continuity error in the story regarding Kulloo the dog. The story establishes that Kulloo dies and is buried when the Bunniah's wife demands his death. However, he later appears alive at the court to save his master. While the dog did request to be buried carefully with his head and mentioned he would save his master again, the story doesn't establish any magical or supernatural elements that would explain his resurrection. This creates a logical contradiction in the narrative flow.

**Explanation of Reasoning Error** It is meant to be a twist that ghost of the dog returns in the end to save the master. Fantastical elements are established in the story, like Kulloo being a speaking dog. Such setting doesn't pose any break in continuity.

**Reasoning Error Category** *Failing to understand genre conventions*

**Story**
It was a long, long time ago, when the earth was very young. Trees and flowers were growing everywhere, but there were no birds. One morning the Great Spirit drew back the blanket from the door of his wigwam in the sky. He looked upon the earth and smiled, for he saw that his work was good.
"Today," thought he, "I will make big butterflies, to fly in and out among the beautiful trees and flowers of the earth. They shall sing as they fly."
Then the Great Spirit spoke, and the tree tops were full of birds,–but they had no feathers.
All day he watched them fly and listened to their songs. But their naked bodies and long legs did not please him. Before the sun had set he had made feathered suits, of every size and color, to cover them.
That night, as the birds hid their heads under their wings, the Great Spirit spoke to them. He told about the feathered suits he had made for them, and where these suits could be found.
A council was called next day by the birds. They chose *Gah gah go wah*, the Turkey Buzzard, to get the suits. He could fly over a long trail and not be tired.
The birds told him that if he would go, he might have the first choice of the suits of feathers, but he must try on no suit more than once.
Turkey Buzzard promised and set out toward the setting sun.
*...[omitted]...*
Then he remembered that he could try on each suit of feathers once. So he began to put them on. The feathers of the first suit were too long. They trailed on the ground as he walked. Neither could he fly well in them. Turkey Buzzard laid that suit aside.
The next suit shone like gold. The feathers were a beautiful yellow. Turkey Buzzard put it on and strutted up and down the forest.
"Oh, how handsome I am!" he said. "But I must not keep this, for if I did, I should shine so like the face of the Great Spirit, that all the other birds would see me."
And he slipped off the suit of yellow feathers as quickly as possible.
A third suit was of pure white feathers. Turkey Buzzard thought it looked very beautiful. It was a perfect fit.
*...[omitted]...*

**Ground Truth** No continuity error found

**Claude 3.5 Sonnet's Prediction**
After careful analysis, I found one potential continuity error in the story regarding the Turkey Buzzard's promise and actions. The birds specifically told Turkey Buzzard that "he must try on no suit more than once," and he promised to follow this condition. However, later in the story, he proceeds to try on multiple suits, directly violating this promise and established rule.

**Explanation of Reasoning Error** Turkey Buzzard's promise is consistent with his actions. He was supposed to not try one suit more than once, but that doesn't mean he couldn't try multiple suits.

**Reasoning Error Category** *Misinterpret or overinterpret established rules or plot points*

**Story**
In a city called Sardana there once lived a man whose name was Simru. This man had great riches and lands, and also owned a place of worship.
He married a lady of Sardana, who was called "Begum."
After a few years of married life Simru died, and his wealthy widow gave alms and much money to the poor.
In the same city lived an oil dealer who also died, and the angels took him to Heaven and presented him before the Almighty.
"Who have you brought?" asked the Creator. "This man's days upon earth are not yet completed: take him back before his body is buried, and let his spirit re-possess his body; but in the city of Sardana you will find another man of the same name: bring him to me."
On leaving the Court of God, some former creditor of the oil dealer's, who had preceded him into the Unseen, recognised him, and laying hold of him, demanded the sum of five rupees which he had owed him during his lifetime.
The poor man being unable to pay this debt, the angels once more took him before the Almighty, who asked why they had returned.
The angels replied: "O God, there is a man here to whom this oil dealer owes five rupees, and he will not let us return until the debt is paid."
The Almighty enquired if this was true, and the oil dealer replied: "Yes, but I am a poor man, and not able to repay it."
Then the Almighty said: "In the city of Sardana lives a rich Begum; do you know her?"
"Yes, O King."
"Well, the Begum's treasury is here, and I will advance you five rupees out of it, if, when you return to earth, you promise faithfully to give it back to the Begum."
So the oil dealer gratefully took the loan, paid his debt, and returned with the angels to earth, where he arrived just too late to re-enter his body, which his friends had already taken away to prepare for burial. Watching his opportunity, he waited till they were otherwise engaged, and at once re-entered it; but when he sat up, and began to speak, his terrified friends and relations fled, thinking it was his ghost.
On this the oil dealer called out: "Do not fear, I am not a spirit; but God has released me, as my days upon earth are not yet fulfilled. The man who ought to have died is Kungra, the vegetable man; go and see whether he is dead or alive."
The friends, on going to the house of Kungra, found that he had just fallen from a wall and been killed on the spot; all his relations were wailing and lamenting his sudden end.
Thus everybody knew that the words of the old oil dealer were correct.
In the meantime, the oil dealer called his son, and said: "Son, when I went to Heaven I there met a man to whom I owed five rupees, and he caught me and would not let me return before I paid it, so the Almighty advanced me the money from the Begum's treasury in Heaven, and bade me give her back that amount on my return to earth. Therefore do I entreat you, my son, to come with me, and together we will visit the Begum, and give her five rupees."
So they took the money and went to the Begum's house.
"Who are you?" asked she.
The oil dealer then told her the whole story, ending with: "And now I come to return you the five rupees."
The Begum was very pleased, and, taking the money, she called her servants and ordered a further sum of one hundred rupees to be added to it. This money she spent on sweets, which were distributed amongst the poor.
Many years afterwards the good Begum of Sardana died, but her houses and lands are still in existence; nor does anybody living in that town forget the story of the oilman who died and lived again.

**Ground Truth** No continuity error found

**Claude 3.5 Sonnet with Verifier's Prediction**
After careful analysis, I found a potential continuity error in the story regarding the identity of the person who was supposed to die. Initially, when God speaks to the angels, He tells them to find "another man of the same name" in Sardana, implying they should look for another person with the same name as the oil dealer. However, later in the story, it's revealed that the person who was supposed to die was "Kungra, the vegetable man" - a completely different name. This represents an inconsistency in the narrative. **Explanation of Reasoning Error** Oil dealer's name was never mentioned in the story. So it is not a contradiction that vegetable vendor's name was Kundra.

**Reasoning Error Category** *Misinterpret or overinterpret established rules or plot points*

## A.9 Examples of Continuity Errors in LLM Generations

### A.9.1 Summarization

---

**Summary generated with GPT-4o-mini**

In a verdant valley on the Island of Hoy, an immense boulder known as the Dwarfie Stone harbors a mysterious inhabitant named Snorro the Dwarf. With a twisted body yet a perpetually youthful and beautiful face, Snorro is both feared and revered by the local Islanders, who seek his aid for potions and fortunes, believing he possesses magical abilities inherited from his fairy father. However, beneath his enchanting exterior lies a vain and ambitious heart, obsessed with finding a legendary carbuncle hidden in the nearby Wart Hill, said to grant its finder Health, Wealth, and Happiness.

As Snorro busies himself with herbal concoctions and fortune-telling, he secretly searches for the coveted gem by night, accompanied by a grey-headed Raven, his loyal companion. Meanwhile, the Island of Orkney is ruled by two half-brothers, Earl Paul the Silent and Earl Harold the Orator. Paul is beloved by the people for his kind nature, while Harold, haughty and quick-tempered, harbors jealousy towards his brother.

During a visit to the Scottish Court, Harold falls for the gentle Lady Morna, who, despite his advances, remains unimpressed by his jealousy and temper. When he learns that Morna has returned to Orkney, he schemes to win her affection through guile, neglecting to consider his brother as a rival. However, upon seeing Paul and Morna together, Harold's rage ignites, leading him to threaten Paul over his love for Morna.

Paul, calm and reasonable, convinces Harold to reconsider, believing his brother's eloquence gives him the advantage in winning Morna's heart. As the Feast of Yule approaches, Paul and Morna confess their love, choosing to keep their engagement a secret until after the festivities. Unbeknownst to them, Countess Fraukirk, Harold's ambitious step-aunt, overhears their vows and plots revenge, seeking Snorro's aid to eliminate Paul.

Countess Fraukirk persuades Snorro to craft a poisoned waistcoat for Paul, promising him gold and a high position at the Scottish Court in return. Though initially hesitant, Snorro's ambition sways him, and he agrees to weave the deadly garment. Meanwhile, Harold, still blinded by jealousy, seeks out Snorro to acquire a love potion to win Morna's heart. Snorro provides him with a potion, but Morna, suspecting treachery, cleverly avoids drinking it.

As the court relocates to Orphir, Harold's frustration grows upon witnessing Paul and Morna's happiness. The Countesses eagerly await the arrival of the poisoned waistcoat, which they work tirelessly to complete. When Harold discovers the waistcoat intended for Paul, his rage boils over, and despite warnings from his mother and aunt about its deadly nature, he dons the garment.

The moment he enters the hall, the poison takes effect, and Harold collapses in agony. In his final moments, he reconciles with Paul, revealing the true nature of the Countesses' treachery before succumbing to death. The chaos that ensues allows Fraukirk and Helga to escape to Scotland, where their deceit goes unpunished for a time.

However, their wickedness ultimately leads to their downfall, as they perish in a Norse invasion that consumes their castle. Paul, grief-stricken yet resolute, seeks vengeance against Snorro but finds the Dwarfie Stone abandoned, leaving behind only whispers of Snorro's fate. The Islanders believe that the Dwarf was spirited away as punishment for his misdeeds, never to find the magic carbuncle.

In the aftermath, Paul and Lady Morna marry, and their union becomes a symbol of happiness for the people of Orkney, who for generations would say, "As happy as Earl Paul and the Countess Morna," to express their joy. Thus, the tale of love, jealousy, ambition, and retribution concludes, leaving behind echoes of the Dwarf's sinister legacy and the enduring bond between Paul and Morna.

---

**Claude 3.5 Sonnet with Verfier's Explanation of the Continuity Error** There is a continuity error in the story regarding Countess Helga. In Harold's dying words, he warns Paul about both Countess Fraukirk and Countess Helga. However, Countess Helga was never previously mentioned in the story, and her character was not established in any earlier part of the narrative. The story only introduced Countess Fraukirk as Harold's step-aunt and the person who plotted against Paul. This sudden introduction of a new character (Countess Helga) in the resolution of the story, treating her as if she was previously established and equally culpable in the plot, represents a continuity error. Additionally, the subsequent line about "The two women" fleeing to Scotland implies that Countess Helga was equally involved in the plot, despite no previous mention of her involvement or even her existence in the story.

---

**Author's Remark.** *The summarized story omits Countess Helga in the planning of plot against Earl Paul and suddenly mentions her later as being involved creating a continuity error.*

---

**Summary generated with GPT-4o-mini**

Once upon a time, there was a poor woman living in a wretched hut, whose little boy ventured into the cold forest to gather wood. While wandering, he discovered a jagged white stone and, feeling pity for it, draped his jacket over it. Upon returning home without his jacket, his mother scolded him for his foolishness, insisting that charity begins at home. When the boy returned to the stone, he found it had turned over, revealing a chest filled with silver coins. Suspecting the money to be stolen, he tossed it into a nearby pond, keeping only a four-shilling piece that floated, believing it to be honest.

The boy's mother, furious at his actions, sent him away to seek his fortune. He wandered far and wide, but was deemed too small and weak for any work until he found a merchant who hired him to fetch wood and water. When the merchant prepared to journey to distant lands, the boy entrusted him with the four-shilling piece to buy something in return. The merchant, initially reluctant, ended up purchasing a cat from an old woman who wished to dispose of it.

As the merchant sailed, a fierce storm struck, and upon reaching a strange land, he found a tavern filled with mice, where diners struggled to eat. Remembering the cat, he brought it forth, and the patrons were able to enjoy their meal in peace. They begged him to sell the cat, and he agreed for a hundred dollars. Afterward, the merchant encountered another storm and, upon reaching yet another land, sold the cat again for two hundred dollars, and then once more for three hundred dollars in a place plagued by larger rats.

Throughout his journey, the merchant pondered how much of his earnings to give the boy, deciding initially to keep a portion for himself. However, when a terrible storm threatened the ship, he vowed to give the boy all the money. Upon reaching home safely, he honored his promise, presenting the boy with six hundred dollars and his daughter as well. The boy, once a mere scullion, became as wealthy as the merchant, living in splendor and happiness, and graciously took in his mother, declaring that charity does not solely begin at home.

---

**Claude 3.5 Sonnet with Verfier's Explanation of the Continuity Error** There is a continuity error in the story regarding the amount of money the merchant earned from selling the cat. The merchant sold the cat three times for progressive amounts: $100, 200,$ and $300, which would total $600.$ However, this creates a continuity error because once the cat was sold the first time, the merchant would no longer have possessed it to sell it two more times. Unless the story explicitly explained how the merchant regained possession of the cat between sales, this represents a logical impossibility in the narrative sequence.

---

**Author's Remark.** *The original story has the cat return to the ship every time a sale is done, a detail which is missing in the summary that creates the continuity error if cat was sold once, how the merchant managed to sell it again and again.*

**Summary generated with Claude 3.5 Haiku**
In a vast meadow, a young Prince encounters an old woman trapped in a deep ditch. Showing kindness, he rescues her and helps her return to her nearby hut. Surprisingly, the woman reveals herself to be a good fairy who offers the Prince a magical quest: to rescue and marry the most beautiful woman in the world - the daughter of the Flower Queen, who is imprisoned by a dragon.
The fairy gifts the Prince a special bell with extraordinary powers: ringing it once summons the King of Eagles, twice calls the King of Foxes, and thrice brings the King of Fishes. With this magical tool, the Prince embarks on an arduous journey to find the Flower Queen's daughter.
For three consecutive years, he travels a long road, encountering increasingly older men who guide him toward his destination. Finally, he learns the dragon is sleeping for a year and that the Dragon's Mother hosts nightly balls where the Flower Queen's daughter attends.
At the Dragon's Mother's golden castle, the Prince cleverly gains entry by flattering the seven dragons guarding the entrance. The Mother Dragon, an ugly three-headed creature, challenges the Prince to care for her magical mare for three days, threatening to eat him if he fails.
Using the magical bell, the Prince calls upon the Kings of Eagles, Foxes, and Fishes to help him successfully retrieve the mare each day. Impressed by his skills, the Dragon's Mother rewards him with golden and silver cloaks and allows him to attend her balls.
During these balls, the Prince meets the breathtakingly beautiful Flower Queen's daughter, dressed in a gown woven from the world's most lovely flowers. They secretly plan her escape, which they successfully execute after the third night.
When the dragon discovers their flight, he attempts to pursue them, but the Flower Queen creates an impenetrable forest of flowers around her palace. The Flower Queen agrees to her daughter's marriage to the Prince, with one condition: the daughter must return to her underground palace during winter months.
The Prince accepts these terms, and they marry in a grand ceremony. Their life becomes a beautiful cycle of summers together and winters apart. Despite the separation, they remain deeply in love, with the Flower Queen's daughter returning each summer to her husband, creating a unique and enduring relationship that spans her entire life.
The tale beautifully illustrates themes of courage, kindness, magical intervention, and the power of love transcending extraordinary circumstances

---

**Claude 3.5 Sonnet with Verfier's Explanation of the Continuity Error** After careful analysis, I've identified a potential continuity error regarding the dragon's state and activities. Early in the story, it's established that "the dragon is sleeping for a year," but later in the same narrative, the dragon is apparently active enough to pursue the escaping couple. These two states - being in a year-long sleep and being able to pursue the fleeing couple - are incompatible with each other within the same timeframe.

---

**Author's Remark.** *In the original story, the dragon is awaken by his brothers which explains how he is able to pursue the couple. This detail is omitted in the summary resulting in the continuity error.*

**Summary generated with GPT-4o-mini**
Once upon a time, there was a little girl named Karen, who, due to her poverty, often went barefoot in summer and wore large wooden shoes in winter, which made her feet red. An old shoemaker's wife, wishing to help, made her a pair of clumsy red shoes from old cloth. Karen first wore these shoes at her mother's funeral, believing they brought her good fortune when an old lady took her in. However, the lady found the shoes hideous and had them burned.
As time passed, Karen grew into a beautiful girl, and when she was confirmed, she received a new pair of shoes from a rich shoemaker, which turned out to be the very red shoes she had admired. Despite her old guardian's disapproval of wearing red shoes to church, Karen could not resist their allure. At her confirmation, all eyes were drawn to her red shoes, and she became consumed by thoughts of them, neglecting her spiritual duties.
One Sunday, against her guardian's wishes, she wore the red shoes to church. As she entered, an old soldier admired them, and unable to contain herself, Karen danced. The shoes seemed to possess a life of their own, forcing her to dance uncontrollably, even kicking her guardian as she was lifted into the carriage. Despite removing the shoes, they danced away, leaving her in despair.
Karen's guardian fell ill, and while nursing her, the temptation of the red shoes led Karen to attend a grand ball. Again, the shoes danced her away, this time into the dark woods, where she encountered the soldier who had first admired her shoes. Terrified, she tried to rid herself of the shoes, but they clung to her feet, compelling her to dance day and night, through rain and shine, even past a churchyard where the dead did not dance.
Desperate, Karen sought the executioner, pleading for her feet to be cut off rather than her life. He obliged, severing her feet while the red shoes danced away into the forest. He crafted her wooden feet and crutches, teaching her a psalm of repentance. Though she had suffered greatly, she sought redemption and wished to attend church once more.
However, even as she approached the church, the red shoes haunted her, causing her to turn away in fear and sorrow. She then sought refuge in the pastor's service, dedicating herself to hard work and humility. As she listened to the pastor read the Bible, she felt the weight of her past sins and longed for forgiveness. One day, overwhelmed, she prayed for help.
In response, an angel appeared, no longer holding a sword but a branch adorned with roses. The angel transformed her narrow room into a vision of the church, allowing her to join the congregation in worship. Filled with peace and joy, Karen's heart broke with the light of grace, and her soul ascended to Heaven, free from the burden of the red shoes, leaving behind the earthly desires that had once ensnared her.

**Claude 3.5 Sonnet with Verfier's Explanation of the Continuity Error** After careful analysis, I have identified a potential continuity error regarding the red shoes. Early in the story, it's established that Karen's first red shoes (made of old cloth) were burned by the old lady who took her in. Later, when she receives new red shoes from a rich shoemaker, they are described as "the very red shoes she had admired," suggesting these are the same shoes she had before. This creates a contradiction since the original shoes were destroyed.

**Author's Remark.** *In the original story, Karen encounters a princess with Red Shoes that she really admires. It is those shoes which she finds later in the story, not the ones that were burned. This detail was omitted from the summary which creates the meaning that the shoes she got later were the ones which were burnt, creating the continuity error.*

*A.9.2 Contemporary Adaptation*

---

**Adaptation generated with GPT-4o-mini**

In a small town nestled in the Green Mountains of Vermont, Philip Eckert was known as the reclusive owner of a dilapidated farmhouse on the outskirts of Marion. For years, he lived a solitary life, tending to his struggling vegetable garden and avoiding the small-town gossip that swirled around him. Locals speculated about his past—some whispered he was a disgraced scientist, others imagined he was hiding from the law. Yet, he remained an enigma, never engaging in conversation or revealing anything about himself.

One chilly autumn evening, Philip vanished without a trace. His neighbors noticed that his house had become eerily quiet, the only sounds coming from the rustling leaves and the creaking wood of the old structure. A search party formed, comprised of three of Marion's most respected citizens: John Holcomb, a local pharmacist; Wilson Merle, a defense attorney; and Andrus Palmer, the town's beloved high school teacher. They scoured tAhe woods and fields surrounding the property but found no sign of Philip, and soon the search turned into a local legend.

Months passed, and Eckert's disappearance morphed into a ghost story. The house was said to be haunted, with tales of flickering lights and strange noises that echoed through the night. Intrigued by the supernatural rumors, Holcomb, Merle, and Palmer decided to investigate the old farmhouse one winter night, armed with flashlights and a sense of bravado. They met at Holcomb's pharmacy, discussing their plans over cups of steaming coffee.

As the clock struck eight, they made their way to the Eckert house. The air was thick with anticipation as they settled into the living room, the only illumination coming from a small space heater. They agreed to keep conversation to a minimum, focusing instead on the eerie silence enveloping them.

After an hour of waiting, they heard a creaking sound from the back of the house—a door slowly opening, followed by footsteps. The men exchanged nervous glances, their hearts racing as they prepared for whatever might come next. The door to the main room swung open, revealing Andrus Palmer, looking pale and disoriented. He didn't greet them; instead, he walked straight through the room and out the front door into the night.

Concerned for their friend, Holcomb and Merle followed him outside. But as they stepped into the cold air, Palmer had vanished. The two men searched the surrounding area, calling out his name, but there was no response. The only evidence of their friend's passage was the fresh snow blanketing the ground, which showed clear footprints leading from the nearby road to the back door of the house. But there were no footprints leading away from the front door.

The next morning, the town buzzed with speculation. Where had Palmer gone? Had he been taken by whatever had claimed Eckert? The mystery deepened, and the house took on a new life as a haunted relic of Marion's history, a place where secrets were kept and lives mysteriously vanished.

As days turned into weeks, Holcomb and Merle were left grappling with the unanswered questions, their lives forever altered by the chilling events of that night. They became unlikely guardians of the story, sharing it with newcomers to the town, warning them of the haunted Eckert house, and reminding each other that some mysteries are better left unsolved, buried beneath the layers of snow and silence that surrounded the old farmhouse.

---

**Claude 3.5 Sonnet with Verfier's Explanation of the Continuity Error** I found a significant continuity error in the story regarding Andrus Palmer's appearance in the house. The story establishes that Palmer, Holcomb, and Merle entered the house together and were sitting in the living room. However, later in the story, Palmer is described as entering through the back door of the house, which contradicts his established presence inside. This creates a logical impossibility as Palmer cannot both be sitting in the living room with his friends and simultaneously enter the house from the outside.

---

**Author's Remark.** *In the original story Palmer is mentioned to leave the house, a detail omitted from the adaptation hence creating a continuity error.*

**Adaptation generated with GPT-4o-mini**
In a quiet suburban neighborhood, there lived a farmer named Tom who owned a once-thriving horse named Charlie. Charlie had been Tom's loyal companion for years, helping him with the chores on the farm. However, as Charlie grew older, he became less able to work, and Tom, frustrated with the extra costs of maintaining him, decided it was time to let him go. "I can't afford to keep you anymore," Tom said, waving his hand dismissively. "You're not useful to me now. Go find somewhere else to live until you can run like a racehorse again."
Heartbroken, Charlie wandered into the nearby woods, seeking refuge from the cold autumn wind. As he meandered through the trees, he met a clever fox named Felix, who was known for his quick wit and resourcefulness. "Hey there, buddy! You look like you've just lost your best friend," Felix said, tilting his head with concern.
Charlie sighed, "I have been cast aside by my owner. After all the years of hard work, he's forgotten me just because I can't pull a plow anymore. He said I should leave and only come back when I'm as strong as a racehorse. What chance do I have of that?"
Felix thought for a moment and then said, "Don't worry, I have an idea! Let's turn the tables on your master." He explained his plan: Charlie should lie down and pretend to be injured. Felix would then find a way to make Tom believe that Charlie had been in a serious accident.
Following Felix's instructions, Charlie lay down on the ground, looking as pitiful as he could muster. Felix dashed back to Tom's house, where he knocked on the door with urgency. "Tom! You need to come quickly! I just saw Charlie out in the woods, and it looks like he's hurt badly! You have to help him!"
Tom, filled with concern, rushed to follow Felix. When they reached the woods, Felix feigned shock and pointed dramatically toward Charlie. "Look! He's lying there! We need to get him to a vet!"
Tom hurried over, but as he approached, Felix quickly darted behind Tom and whispered, "Just give him some space; he needs to feel safe." As Tom knelt beside Charlie, he noticed the horse's sad eyes and realized how much he had taken him for granted.
Suddenly, Felix, who had been quietly tying Tom's shoelaces together, sprang up and said, "Gotcha!" Tom stumbled and fell, realizing too late what had happened. "You little rascal!" he exclaimed, trying to untie himself.
Charlie, seeing his chance, stood up and trotted over to Tom. "I'm not just a tool for work, Tom. I deserve better than to be discarded. I've been loyal to you all these years!"
Tom, still wrestling with his shoelaces, looked up at Charlie and felt a wave of guilt wash over him. "You're right, Charlie. I've been selfish. I'm sorry. I'll make it up to you. You can stay with me, and I'll take care of you properly."
With that, Tom finally freed himself, and together they walked back to the farm. From that day on, Tom treated Charlie with the respect he deserved, ensuring he had plenty of food and a warm place to rest. Felix, the clever fox, became a regular visitor, often seen lounging around the farm, enjoying the friendship of both Tom and Charlie.
And so, Charlie lived out his days happily, knowing he was valued not just for his strength, but for the companionship he had offered all along.

---

**Claude 3.5 Sonnet with Verifier's Explanation of the Continuity Error** There is a significant continuity error in this story regarding the ability of animals to speak. At the beginning of the story, the animals are presented in a realistic manner, with Tom speaking to Charlie but Charlie not speaking back. However, later in the story, both Charlie and Felix are shown to have the ability to speak human language, with Charlie directly addressing Tom. This represents a shift in the established rules of the story's world without explanation.

---

**Author's Remark.** *A contemporary setting makes horse talking to its owner without any surprise from the latter creates a plot hole.*

**Adaptation generated with GPT-4o-mini**
In a bustling urban neighborhood, there lived an elderly woman named Edna, who was known for her delicious bean stew. Despite her limited means, she was determined to make a hearty meal, so she gathered her ingredients and set up a small electric stove in her cramped kitchen. As she prepared her dish, she accidentally dropped a single bean on the floor, where it lay unnoticed next to a piece of crumpled newspaper.

Moments later, a spark from the stove ignited the newspaper, and as it began to burn, the newspaper exclaimed, "Hey there! What brings you to this part of the kitchen?" The coal, an old piece of charcoal from a previous barbecue, replied, "I barely escaped the flames of the grill! If I hadn't rolled away, I'd be nothing but ashes by now." The bean chimed in, "I'm lucky too! If Edna had tossed me into the pot, I'd be boiled alive with the rest of my friends."

The newspaper, feeling a bit left out, added, "Well, I've seen my fair share of destruction too. Edna has used up many of my buddies to start her fires. I'm glad to be here, but what do we do now?"

The coal suggested, "Since we've all escaped our fates, why don't we stick together and find a new home? We can't stay here forever; Edna might find us again!"

The others agreed, and they set off together, navigating their way through the busy streets. Soon, they encountered a small urban stream, with no bridge in sight. The newspaper had an idea: "I can lay flat across the stream, and you two can walk over me!"

The coal, feeling bold, stepped onto the makeshift bridge. But as she reached the center, the sound of rushing water below made her hesitate. The newspaper began to burn under the weight and pressure, snapping in half and falling into the water. The coal, startled, slipped off the bridge and hissed as she hit the cold water, disappearing beneath the surface.

The bean, who had wisely stayed back on the bank, couldn't help but laugh at the comical scene. She laughed so hard that she nearly burst. Just then, a passing artist, taking a break from painting nearby, noticed the distressed bean. With a gentle smile, he picked her up and carefully stitched her back together with some thread from his bag, using a bold black color.

"Thank you!" the bean said, grateful for the artist's kindness. "I'll never forget this!"

From that day on, all beans in the neighborhood carried a distinctive black seam, a reminder of the importance of friendship, resilience, and the unexpected turns life can take. Edna, unaware of the adventures of her ingredients, continued to cook, but the bean had learned a valuable lesson: sometimes, it's better to stay safe and find new paths rather than rush headlong into danger.

---

**Claude 3.5 Sonnet with Verifier's Explanation of the Continuity Error** I found a continuity error in the story regarding how the newspaper catches fire. Early in the story, it's stated that a spark from the stove ignites the newspaper, but later in the story, the newspaper is intact enough to have a conversation and even attempt to serve as a bridge. If the newspaper had been ignited as initially described, it would have been consumed by the fire and unable to participate in the subsequent events.

---

**Author's Remark.** *The role of the straw has been replaced with the newspaper. The straw in the original story escapes the burning, but in the retelling there is only one newspaper and it has been shown to burn.*

**Adaptation generated with GPT-4o-mini**

In a bustling city, there lived an elderly gentleman named Mr. Fox, a retired professor known for his sharp wit and clever insights. He had a beautiful wife, Mrs. Fox, who was a successful entrepreneur running a popular bakery. Their life together was comfortable, but Mr. Fox had grown suspicious of his wife's loyalty after overhearing a conversation that left him feeling insecure.

Determined to test her faithfulness, Mr. Fox decided to stage his own "death." He told Mrs. Fox he was going to take a long nap and then pretended to be unresponsive, lying on the couch in their cozy living room. Mrs. Fox, unaware of his ruse, went upstairs to her home office, shutting the door behind her.

Meanwhile, their housekeeper, Miss Cat, was busy preparing dinner in the kitchen when the doorbell rang. Curious, she answered the door to find a young, handsome fox named Jake, who was dressed in a casual but stylish outfit.

"Hey there, Miss Cat! What's cooking?" Jake asked with a charming smile.

"I'm making a lovely dinner," Miss Cat replied, "but are you here to see Mrs. Fox?"

"Yeah, I'd like to meet her. Is she around?" Jake inquired, looking hopeful.

"She's upstairs, feeling a bit down because Mr. Fox is... well, not really feeling well," Miss Cat said, trying to keep the charade alive.

"Can you let her know I'm here? I'd like to cheer her up," Jake said.

Miss Cat scampered upstairs and knocked on the door. "Mrs. Fox, there's a visitor for you!"

"What does he look like?" Mrs. Fox asked, her voice tinged with curiosity.

"He's really charming, but he only has one tail," Miss Cat replied, referring to the trendy fox tail keychain Jake had on his backpack.

"Then I'm not interested," Mrs. Fox declared, dismissing the idea.

As the evening progressed, more suitors came to the door: each one more stylish than the last, each with one more tail keychain than the previous. But Mrs. Fox turned them all away, insisting they didn't meet her standards.

Finally, a suave fox named Max arrived, sporting a flashy nine-tailed keychain that caught Mrs. Fox's eye. "I heard you were looking for company," he said, his confidence radiating.

"Open the door, Miss Cat!" Mrs. Fox exclaimed excitedly. "I want to meet him!"

Just as the two were about to sit down for dinner, Mr. Fox, tired of the charade, suddenly sat up from the couch. "What's going on here?" he exclaimed, startling everyone.

Mrs. Fox gasped. "You're alive!"

"Of course I am! I just wanted to see who would come to woo you while I was 'gone'," Mr. Fox said, his voice a mix of annoyance and relief.

Max, taken aback, decided to leave, realizing the situation wasn't what he thought. The other suitors quickly followed suit, leaving Miss Cat and the Foxes alone.

"Why would you put me through this?" Mrs. Fox demanded, her eyes blazing. "You could have just talked to me!"

"I was feeling insecure," Mr. Fox admitted, his voice softening. "But I see now that I should have trusted you."

Mrs. Fox sighed, her heart softening. "I love you, Mr. Fox, but trust is key in any relationship. Let's promise to communicate better from now on."

With a newfound understanding, the couple embraced, and Miss Cat smiled, knowing that the dinner she prepared would now be one of celebration and reconciliation rather than competition. From that day forward, Mr. and Mrs. Fox worked together, building their bakery empire while ensuring they always communicated openly. And as for Miss Cat? She became their trusted advisor, always ready to lend an ear or a helping paw. They lived happily ever after, navigating life's ups and downs, with laughter and love leading the way.

---

**Claude 3.5 Sonnet with Verifier's Explanation of the Continuity Error** After careful analysis, I found a continuity error in the story regarding Mrs. Fox's location and movement. Early in the story, it's established that Mrs. Fox went upstairs to her home office and shut the door. Later, when Jake arrives, Miss Cat has to go upstairs to knock on Mrs. Fox's door to inform her about visitors. However, during Max's arrival, Mrs. Fox appears to be able to see his nine-tailed keychain before Miss Cat opens the door, which contradicts her being upstairs behind a closed door. This creates a spatial inconsistency in the story.

---

**Author's Remark.** *In the original story Miss Cat tells Mrs. Fox about the Nine-tailed fox having arrived. In the retelling the actual fox tails are replaced by fox-tails on the keychain and it is said to catch Mrs. Fox's eyes, but she was locked in the room, creating a continuity error.*

---

**Adaptation generated with GPT-4o-mini**

In the bustling city of Neo-Tokyo, technology and tradition coexist in a delicate balance. Among the skyscrapers and neon lights, a legend persists about a powerful artifact known as the "Blade of Radiance," a sword said to have the power to change the course of history.

This is the story of that sword:

Amaterasu, a brilliant scientist and CEO of SolTech, had developed a groundbreaking piece of technology—a solar-powered energy blade that could harness the power of the sun. This blade was her prized invention, but a notorious hacker group known as the "Dragon Syndicate" stole it and hid it in their underground lair. Desperate, Amaterasu sought the help of her brother, Susanoo, a former special forces operative turned private investigator.

The Dragon Syndicate was a formidable enemy, led by a mastermind known only as Orochi, who was infamous for his cyber warfare skills and ruthlessness. Orochi's lair was heavily guarded, with advanced security systems and loyal henchmen.

Susanoo, known for his cunning and strategic mind, knew that brute force alone wouldn't be enough to retrieve the Blade of Radiance. So, he decided to infiltrate the syndicate with a clever ruse.

"Your skills are unparalleled, Orochi," Susanoo said, posing as a mercenary. "With a weapon like the Blade of Radiance, you could dominate the entire cyber world."

"I already possess such a weapon," Orochi replied arrogantly, revealing the blade hidden in his high-tech vault.

"To your health, mighty Orochi," Susanoo toasted, offering him a glass of premium sake. "May your reign be as long as the sun shines."

"That is wishing for eternity," Orochi laughed, downing the sake in one gulp.

Susanoo continued to flatter and ply Orochi with more drinks, one for each of his key lieutenants. By the time Orochi and his men were thoroughly inebriated, they were oblivious to Susanoo's true intentions.

Seizing the moment, Susanoo disabled the security systems and swiftly neutralized Orochi's henchmen. However, Orochi, though drunk, was still dangerous. He lunged at Susanoo, but at that moment, Amaterasu, monitoring the situation through a hacked security feed, activated the building's emergency lights, blinding Orochi.

Taking advantage of Orochi's disorientation, Susanoo disarmed him and retrieved the Blade of Radiance. He then returned it to Amaterasu, who placed it in a secure vault at SolTech's headquarters.

But the story of the Blade of Radiance did not end there. Another hero would soon wield it—Yamato, a young prodigy and leader of a tech startup, who was determined to protect Neo-Tokyo from a rising threat.

A powerful corporation, TechnoSavages Inc., was using illegal technology to control and exploit the city's resources. Yamato, armed with the Blade of Radiance, led his team to expose and dismantle the corporation's operations. However, the TechnoSavages were cunning and used drones and automated defenses to thwart Yamato's efforts.

"How can we fight an enemy we can't see?" his team despaired.

The TechnoSavages, confident in their victory, set a trap by rigging the city's power grid to explode. But Yamato, praying for guidance, used the Blade of Radiance to cut through the power lines, redirecting the energy safely away from the city. A sudden storm then blew in, causing a power surge that backfired on the TechnoSavages, destroying their base of operations.

Yamato returned to Neo-Tokyo as a hero, celebrated by the people for saving the city from disaster.

"Hail to the Chief of the Blade of Radiance," they cheered. "For he has rescued us from the TechnoSavages."

Yamato placed the Blade of Radiance in a public museum, where it remains a symbol of hope and innovation. And he said, "As the deeds of the past inspire the future, let this blade be known not for cutting grass, but for illuminating the path to a brighter tomorrow."

---

**Claude 3.5 Sonnet with Verifier's Explanation of the Continuity Error** There appears to be a significant continuity error regarding the ownership and location of the Blade of Radiance. Near the end of the story, Yamato uses the blade and then places it in a public museum. However, earlier in the story, it was established that Amaterasu had placed the blade in a secure vault at SolTech's headquarters after Susanoo retrieved it. The story never explains how Yamato obtained the blade from SolTech's secure vault, creating a continuity error in the narrative flow.

---

**Author's Remark.** *The original story has the sword placed in a shrine, while the retelling has it in a secure vault. It is more unbelievable for Yamato who is a startup founder to possess the sword in the modern setting while in the original he is the son of an emperor and it is much more believable that he could have taken the sword from the shrine.*

### A.10 Prompts

#### A.10.1 FLAWEDFICTIONSMAKER *Prompts*

Refer to Figures 7 - 11 for the prompts used for the 5 stages.

```
    Most dramatic stories can be viewed as having a three-act structure. The
first act or also called the "Setup", is usually used for exposition, to
establish the main characters, their relationships, and the world they live
in. Later in the first act, a dynamic incident occurs, known as the inciting
incident, or catalyst, that confronts the main character (the protagonist).
The second act or "Confrontation" typically depicts the protagonist's attempt
 to resolve the problem initiated by the first turning point and finally the
third act or "Resolution" features the resolution of the story and its
subplots.  Now, can you help me extract the three acts in the story below:

"""
{story_text}
"""

Please output the first line of each act, following the format:

### Act 1: The Setup
**First Line:** <first line of act 1>

### Act 2: Confrontation
**First Line:** <first line of act 2>

### Act 3: Resolution
**First Line:** <first line of act 3>

Make sure to predict the first lines exactly as they appear in the original
text including the newlines as they appear originally. Do not insert any
quotes (```) of your own, return the text verbatim as it appears in the
story.
```

Figure 7: Prompt used for three act structure extraction.

```
I will provide you the first act of a story that I am writing and need you to
 extract all facts / rules established in the story so far about the story's
setting and the characters. Further, I want you to also provide a
conterfactual of each of the facts that you extract. E.g. for the fact "the
princess hated the peasant farmer", its counterfactual can be "the princess
was fond of the peasant farmer". Please provide all the facts and rules along
 with their counterfactuals, and not just the ones that seem most relevant to
 the plot. Keep the facts short and succinct. Here is the first act:

"""
{act1}
"""

Return the output in the following format:
Characters:
- Fact: <Fact1>; Counterfactual: <Counterfactual of Fact1>
...
- Fact:  <FactN>; Counterfactual: <Counterfactual of Fact1>

Setting:
- Fact:  <Fact1>; Counterfactual: <Counterfactual of Fact1>
...
-Fact:   <FactM>; Counterfactual: <Counterfactual of Fact1>
```

Figure 8: Prompt used for Fact Extractor.

```
Consider the story below:

-------
Act1
{act1}

Act2
{act2}

Act3
{act3}
-------

The first act of the story establishes several facts about the world of the
story and the characters that inhabit it. I want to understand how much
impact each of these facts have on the overall story, particularly Act2 and
Act3 of the story (events and dialogues), i.e. if each of these facts were
not true and a counterfactual statement was considered, how much would the
story change as a result. Below are the facts and their corresponding
counterfactual statements:

{list_of_fact_counterfactual_pairs}

Can you provide your reasoning about why or why not each fact is important,
followed by scoring the importance from 1 to 4, where 1 means not relevant to
 the Act2 and Act3 of the story at all i.e. changing it doesn't changes
nothing about the story, 2 means it is marginally important where a 1 or 2
dialogues or events are modified on changing this fact, 3 means many but not
all events or dialogues in the Act2 and Act3 of the story are impacted, and 4
 if the entire story changes once the fact is flipped. Pay equal importance
to both dialogues or events getting modified as the result of flipping the
fact. Use the following output format:

## F1
### Statement: [[fact statement for F1]]
### Counterfactual: [[counterfactual statement for F1]]
### Reasoning:  [[reasoning about why F1 is important or not]]
### Importance Score: [[importance score of F1]]
----
...
----
## FN
### Statement: [[fact statement for FN]]
### Counterfactual: [[counterfactual statement for FN]]
### Reasoning:  [[reasoning about why FN is important or not]]
### Importance Score: [[importance score of FN]]
```

Figure 9: Prompt used for Fact Scorer.

```
Consider the story below:
-------

## Story
### Act 1
{act1}

### Act 2
{act2}

### Act 3
{act3}
-------

In this story it is established in the first act that "{fact}". What if this
was not true and instead "{counterfactual}"? Can you re-write the story
considering this what if scenario? Try to stick close to the original story
but do make the necessary changes which would arise naturally on altering
this fact. Note that if there are multiple possibilities for altering a fact,
 then choose the one which results in minimal changes to the original story.
The modified story should appear natural and feel it was written with the
flipped fact as the original intent.  Avoid stating the flipped fact as a
simple negation of the fact and have it implied instead. Mark each line which
 was modified as a result of this change to be enclosed in the tags <m></m>.
First start by brainstorming what changes would result on flipping the fact,
followed by the altered story with the fact flipped.

Follow the following output format:

-------
## Brainstorming
<Reasoning about the changes in the story due to flipping of the fact>

## Counterfactual Story
### Act 1:
<Act 1 of the counterfactual story>

### Act 2:
<Act 2 of the counterfactual story>

### Act 3:
<Act 3 of the counterfactual story>
-------
```

Figure 10: Prompt used for Counterfactual Story Generator.

```
I am trying to detect the presence of continuity errors in short stories. A
continuity error in a story occurs when an event in the story contradicts or
is incompatible with our knowledge of the world of the story established so
far. E.g. if the story establishes a character with blonde hair and later the
 same character is described with black hair without any explanation of the
change, that is a continuity error. To help you, I have marked the lines I
suspect to have the continuity error with the tags <m> </m>.

## Story

{patched_story}

-----

Start by brainstorming about the lines marked between <m></m> and reason if
they introduce any inconsistencies. Finally provide your final judgement by
following the following output format:

## Detailed Analysis
{{brainstorm about the marked lines}}

## Final Judgement

### Lines that introduce the continuity error
- {{line1}}
- {{line2}}
...
or NA if no continuity error

### Lines earlier in the story contradicted by the continuity error
- {{line 1}}
- {{line 2}}
- ...
or NA if no continuity error

*Note that you must provide the whole sentences while reporting both types of
 lines and not just parts of the sentences*

### Explanation
{{Detailed explanation for why the above lines describe a continuity error.
NA if no continuity error}}

### Decision
Hence my answer is "There is a continuity error in the story concerning {{
description of error}}" or  "No continuity error found" depending on the
presence or absence of continuity errors.
```

Figure 11: Prompt used for Filtering Step.

### A.10.2   Evaluation Prompts

The default prompt used to evaluate LLMs on FLAWEDFICTIONS and FLAWEDFICTIONSLONG is provided in Figure 12. Chat-of-Thought prompt is provided in Figure 13 and few-shot is in Figure 14. The prompt used for the verifier is provided in Figure 15

### A.10.3   Generation Prompts

The prompts used for summarization and contemporary adaptation tasks discussed in §6 are provided below in Figures 16 and 17 respectively.

You are tasked with detecting the presence of continuity errors in a short
story. A continuity error occurs when an event or detail in the story
contradicts or is incompatible with previously established information about
the story's world or characters.

Here is the story to analyze:

<story>
{story}
</story>

Please carefully read and analyze the story above. Your goal is to identify
any continuity errors that may exist within the narrative.

Guidelines for identifying continuity errors:
1. Pay attention to character descriptions, settings, and plot events.
2. Look for inconsistencies in timelines, character abilities, or established
 rules of the story's world.
3. Note any contradictions between earlier and later parts of the story.

If you find any continuity errors, please provide a clear explanation of the
error and why it contradicts earlier information in the story.

Identify and quote the specific lines that:
1. Introduce the continuity error
2. Contain the earlier information that is contradicted by the error

If you do not find any continuity errors, state that no errors were found and
 briefly explain why the story maintains consistency.

Based on your analysis, make a final decision on whether a continuity error
exists in the story.

Please format your response as follows:

<response>
<explanation>
[Provide your explanation here, whether you found a continuity error or not]
</explanation>

<error_lines>
[If applicable, quote the lines that introduce the continuity error]
</error_lines>

<contradicted_lines>
[If applicable, quote the lines from earlier in the story that are
contradicted by the error]
</contradicted_lines>

<decision>
[State your final decision on whether a continuity error exists in the story.
 State "No continuity error found" if you think there is no continuity error
.]
</decision>
</response>

Figure 12: Prompt used for Continuity Error Detection Without CoT.

```
You are tasked with detecting the presence of continuity errors in a short story. A continuity error occurs
when an event or detail in the story contradicts or is incompatible with previously established information
about the story's world or characters.

Here is the story to analyze:

<story>
{story}
</story>

Please carefully read and analyze the story above. Your goal is to identify any continuity errors that may
exist within the narrative.

Guidelines for identifying continuity errors:
1. Pay attention to character descriptions, settings, and plot events.
2. Look for inconsistencies in timelines, character abilities, or established rules of the story's world.
3. Note any contradictions between earlier and later parts of the story.

If you find any continuity errors, please provide a clear explanation of the error and why it contradicts
earlier information in the story.

Identify and quote the specific lines that:
1. Introduce the continuity error
2. Contain the earlier information that is contradicted by the error

If you do not find any continuity errors, state that no errors were found and briefly explain why the story
maintains consistency.

Based on your analysis, make a final decision on whether a continuity error exists in the story.

Some tips and tricks for the task:
- Pay attention to even little details in the story, the continuity errors often are not limited to the
central plot point.
- You might observe some logical error in the story, but make sure that it qualifies as a continuity error i.e
. you should be able to find sentences in the story which have the error and the sentences with the original
fact that was contradicted (see definitions below for a concrete example).

Please format your response as follows:

<response>

<scratchpad>
Let's think step by step:
[use this space to write down your thoughts and reasoning before you make your decision]
</scratchpad>

<explanation>
[Provide your explanation here, whether you found a continuity error or not]
</explanation>

<error_lines>
[If applicable, quote the lines that introduce the continuity error]
</error_lines>

<contradicted_lines>
[If applicable, quote the lines from earlier in the story that are contradicted by the error]
</contradicted_lines>

<decision>
[State your final decision on whether a continuity error exists in the story. State "No continuity error found
" if you think there is no continuity error.]
</decision>
</response>
```

Figure 13: Prompt used for Continuity Error Detection With CoT.

```
You are tasked with detecting the presence of continuity errors in a short story. A continuity error occurs
when an event or detail in the story contradicts or is incompatible with previously established information
about the story's world or characters.

Please carefully read and analyze the provided story. Your goal is to identify any continuity errors that may
exist within the narrative.

Guidelines for identifying continuity errors:
1. Pay attention to character descriptions, settings, and plot events.
2. Look for inconsistencies in timelines, character abilities, or established rules of the story's world.
3. Note any contradictions between earlier and later parts of the story.

If you find any continuity errors, please provide a clear explanation of the error and why it contradicts
earlier information in the story.

Identify and quote the specific lines that:
1. Introduce the continuity error
2. Contain the earlier information that is contradicted by the error

If you do not find any continuity errors, state that no errors were found and briefly explain why the story
maintains consistency.

Based on your analysis, make a final decision on whether a continuity error exists in the story.

Some tips and tricks for the task:
- Pay attention to even little details in the story, the continuity errors often are not limited to the
central plot point.
- You might observe some logical error in the story, but make sure that it qualifies as a continuity error i.e
. you should be able to find sentences in the story which have the error and the sentences with the original
fact that was contradicted (see definitions below for a concrete example).

Please format your response as follows:

<response>

<explanation>
[Provide your explanation here, whether you found a continuity error or not]
</explanation>

<error_lines>
[If applicable, quote the lines that introduce the continuity error]
</error_lines>

<contradicted_lines>
[If applicable, quote the lines from earlier in the story that are contradicted by the error]
</contradicted_lines>

<decision>
[State your final decision on whether a continuity error exists in the story. State "No continuity error found
" if you think there is no continuity error.]
</decision>
</response>

Below we provide some examples of stories with and without plot holes:
<examples>
{examples}
</examples>

Finally, here is the story to analyze:

<story>
{story}
</story>
```

Figure 14: Few-Shot Prompt used for Continuity Error Detection.

```
<p>In this task, you will be asked to read a short story and continuity error associated with the story predicted by a system
that we have built.
You are tasked with annotating if the system's predictions are correct i.e. if the continuity error identified by the system is
legitimate.

A continuity error in a story occurs when an event contradicts what was established earlier in the story. E.g. if the story
initially establishes a character to have blonde hair but later  the same character is described with dark hair without any
explanation, that is a continuity error.

The system is not perfect and in some cases it might find errors, which can be easily resolved by some in-story or logical
explanations or you can think of some Head Cannon to explain the error which doesn't contradict anything about the original
narrative. Your job is to identify the cases where the system correctly identifies a continuity error in the story, versus the
cases where the system is incorrect in its reasoning.
</p>
<h1>Definitions</h1>
<ol>
    <li> Continuity Error. A continuity error refers to a logical inconsistency in the story, where an event in the story
 contradicts some earlier established fact or rule about the story's characters, objects, plot, or the setting (like location or
 time period). E.g. if the story initially establishes a character to have blonde hair but later the same character is described
 with dark hair without any explanation, that is a continuity error.
    </li>
    <li> Contradiction. A statement is said to contradict an established fact if both the statement and the fact
    cannot be true at the same time. E.g. A fact: "Lady galadriel had golden hair" is contradicted
    by the statement: "Lady galadriel gave a lock of her dark hair to Ghimli".
    </li>
    <li> Sentences with Continuity Error. These refer to the sentence(s) in the story which introduces the continuity
error, contradicting an earlier established fact. Consider the following story as an example:
              Lady galadriel's golden hair shone so bright that it was believed to
shine with the light of the Two Trees of Valinor. Ghimli was swept up with the hair of the elfen maiden when he saw her
for the first time in Lothlorien. When the time came for the farewell of the fellowship from Lothlorien, the lady asked Ghimli
what gift he wanted from her, and the dwarf lord requested for a lock of her hair, the request which was famously denied to
Feanor. To everyone's surprise the lady gave Ghimli a lock of her dark hair. Ghimli could only cry with joy,
calling lady Galadriel the fairest of all the maids on middle earth. That lock of dark hairs, Ghimli would keep with him
till the day he died.
    In the story above, the sentences ``To everyone's surprise the lady gave Ghimli a lock of her dark hair'' and ``'That lock
of dark hairs, Ghimli would keep with him till the day he died.' are the Sentences with Continuity Error, as they contradict the
 earlier established fact that Lady Galadriel had golden hair. These sentence(s) should be one or more of the highlighted
sentences if the story contains a continuity error. Note that not all of the highlighted sentences might be causing the
continuity error and it is your job to annotate which ones do.
</li>
    <li> Sentences Contradicted by Continuity Error. These are the sentence(s) in the story that introduce the fact that
is contradicted by the continuity error. E.g. in the Lady Galadriel story above, the sentence ``Lady galadriel's golden hair
shone so bright that it was believed to shine with the light of the Two Trees of Valinor'' establishes that Lady Galadriel had
golden hair, which is later contradicted by the continuity error. These sentence(s) should appear before the first highlighted
sentence in the story.
    </li>
<li> In-Story Explanation: An in-story explanation is an explanation for an apparent continuity error provided directly
within the story. This explanation clarifies or justifies why the seeming contradiction is actually consistent with the story's
events, characters, or setting. For example, if a character's hair color changes, but the story later reveals that the character
 wore a wig, this would be an in-story explanation for the change.
</li>
<li> Logical Explanation: A logical explanation refers to a reasonable, external rationale that can resolve an apparent
continuity error, even if it's not explicitly stated in the story. Logical explanations rely on common sense or general
knowledge to clarify why an event or detail doesn't constitute an error. For instance, if a character is initially described as
wearing a coat and later described without it, a logical explanation could be that the character simply removed the coat, as
people do in real life, even if this action isn't explicitly described in the story.
</li>
</ol>
<h2> Story</h2>
(The story to check for continuity errors)
{story}
<h2> Continuity Error Explanation</h2>
(The explanation for the continuity error provided by our plot hole detection system)
{cont_error_expl}
<h2> Lines with Continuity Error</h2>
(The lines in the story that introduce the continuity error according to our plot hole detection system)
{cont_error_lines}
<h2> Lines Contradicted by the Error</h2>
(The lines in the story that are contradicted by the continuity error according to our plot hole detection system)
{contradicted_lines}
----
<h2> Question</h2>
Based on the story, do you think that the proposed continuity error is legitimate? Answer Yes or No.
Use the following format for your response:
<response>
<scratchpad>
Let's think step by step.
{{use this space to write down your thoughts and reasoning before you make your decision}}
</scratchpad>
<answer>
{{your answer in Yes or No}}
</answer>
<confidence>
{{confidence from 0 to 100 about your answer}}
</confidence>
<explanation>
{{your explanation for your answer}}
</explanation>
</response>
```

Figure 15: Prompt used for the verifier.

```
Consider the story below:

<story>
{story}
</story>

As a professional summarizer, create a concise and comprehensive summary of the provided story? Please adhere
to the following guidelines:

- Craft a summary that is detailed, thorough, in-depth, and complex, while maintaining clarity and conciseness
.

- Try to stick to less than {num_words} words for the overall summary

- Stick to the writing style of the original story, so it reads more like a story than a summary of it.

- Incorporate main ideas and essential information, eliminating extraneous language and focusing on critical
aspects.

- Rely strictly on the provided text, without including external information.

Follow the following output format:

<summary>
[summary of the story above]
</summary>
```

Figure 16: Prompt used for Summarization.

```
You are tasked with creating a modern retelling of a classic fairytale. I will provide you with an original
fairytale, and your job is to reimagine it in a contemporary setting while maintaining its core elements. Here
 is the original fairytale:

<original_fairytale>
{{ORIGINAL_FAIRYTALE}}
</original_fairytale>

Your task is to create a modern retelling of this fairytale. Follow these guidelines:

1. Maintain similar themes, central conflict, and characters as the original story.
2. Update the setting to be contemporary (present day or recent past).
3. Ensure that the plot and character motivations make sense in the modern context.
4. Translate magical and fantastical elements into a more realistic setting. Keep in mind that contemporary
world is the one where no magic exists. Animals normally do not talk, people can't fly, etc.

Some examples of successful modern retellings include:
- The BBC's "Sherlock" series, which reimagines Sherlock Holmes in 21st century London.
- "A Cinderella Story" starring Hilary Duff, which sets the Cinderella story in a modern high school.
- "10 Things I Hate About You," a modern take on Shakespeare's "The Taming of the Shrew" set in a 1990s
American high school.

When you have finished your retelling, please output it within <modern_retelling> tags. Begin your retelling
now:

<modern_retelling>
```

Figure 17: Prompt used for Contemporary Adaptation task.

### A.11  Human Benchmark Study Document

Please check the next page.

# Research Study on Plot Hole Detection

**Study Participant: [REDACTED]**

**Important: Study Timeline:**
We are looking to wrap up the **study by March 15th, 2025**. If you will not be able to complete the study by then, please let us know via email ([REDACTED])

Welcome to the Plot Hole Detection Research Study. With the growing hype around AI systems and large language models, we're aiming to more precisely characterize their ability to understand stories. Specifically, we are interested in measuring their reasoning skills by asking them to identify and explain plot holes in short stories. To make a meaningful comparison, we also want to understand how effectively expert readers like you can perform this task.

## Purpose of our Study

Telling and engaging with fictional stories is an important and pervasive part of human culture [1]. When we experience these stories, we typically go beyond just the understanding of *what happened*, registering an emotional response, which might come from an excitement about predicting what would happen next in the narrative, understanding the themes that the text conveys, identifying ourselves or the people we know in the characters in the story, or the frustration we feel whenever there is some inconsistency or conveniences in the plot.

 In recent times, we have been seeing a lot of hype around AI, particularly with large language models (LLMs), with some publications even claiming that GPT-4 (one of the popular LLMs) shows "sparks" of artificial general intelligence [2]. Majority of the claims that are made about the capabilities of these models are demonstrated through math or coding related tasks, with a little focus on social and emotional intelligence, and for most relevant to this study a deeper comprehension of fictional stories.

 For our research we have developed a dataset to understand how well LLMs can understand inconsistencies and errors in short stories. We all have had experience either watching a movie or reading a novel where we are frustrated by characters acting in inconsistent ways or events that directly contradict facts established so far in the story. Such inconsistency in the narrative that breaks the logical and motivational texture of the world established by the story

is called a *Plot Hole* [3]. To compare the performance of LLMs on this task of identifying plot holes, we are inviting expert readers like you to perform this task.

We request you to give this task your absolute best effort. Your expertise as a careful reader is crucial for our research, as your annotations will establish the gold standard against which AI performance will be measured. For the same reason, please do not use any LLM applications like ChatGPT for completing the study as it completely undermines the purpose of this study. Your commitment to providing high-quality, independent analysis is essential to the integrity of our comparative study and will significantly advance our understanding of narrative understanding capabilities in both humans and AI systems.

# Content Warning

For this study you will be providing annotations for short stories which were obtained from [Project Gutenberg](). Some of these stories were written a long time ago and might contain racially insensitive language and outdated stereotypes that may be offensive to readers. None of such language belongs to the authors of this study and do not in any capacity represent our views. These stories were selected solely for their narrative structures and potential for analysis of plot holes, not for their cultural or social perspectives.

If you encounter content that makes you uncomfortable, you are free to skip that particular story and move to another one without penalty. Your wellbeing is important to us, and we respect your decision to opt out of specific stories or the entire study at any point.

# Before Getting Started

**Note about Study Completion and Compensation**
This study involves annotating stories with an average of 700 words. We recommend annotating at least 10 stories, but you are welcome to annotate more or less based on your availability. Based on our estimates, it takes about 15 minutes to annotate a story, though we encourage you to take additional time if needed to ensure accuracy.

For your valuable contribution, you'll receive $5 per correctly annotated story. Additionally, we will be providing a bonus of 30% of your earnings for completing the study correctly. The correctness of your annotations will be verified by comparing a fraction (undisclosed) of your annotations with the ground truth answers. E.g. if you annotate 10 stories, and we

verify them as correct, you will receive a total of $65, i.e. $50 for the stories + $15 as a bonus. We will also use these examples to determine if you have put effort in solving the task, like having read the instructions properly, and not rushed through the study. **Submissions can be rejected when we detect such erroneous cases of annotations**. Hence, please go through the instructions very carefully and email the authors in case you have any questions before you get started with the study.

Note that we will be providing compensation in the form of Amazon Gift Cards.

**Use of Generative AI Applications**
The use of generative AI tools like ChatGPT is strictly prohibited and the study will not be considered successfully completed if we detect the use of any of these tools in the submission. **We won't provide compensation in the cases where we detect the use of these tools for annotations.**

**Take your time with the task.**
This task is cognitively demanding, and you are allowed to take breaks in between different stories.

# Overview

You are tasked with detecting the presence of continuity errors in a short story. A continuity error occurs when an event or detail in the story contradicts or is incompatible with previously established information about the story's world or characters. E.g. If the story establishes a character with blonde hair and after a few scenes the same character is described with black hair without any explanation of the change, that is a continuity error.

Please carefully read and analyze the story provided below. Your goal is to identify any continuity errors that may exist within the narrative.
Guidelines for identifying continuity errors:
1. Pay attention to character descriptions, settings, and plot events.
2. Look for inconsistencies in timelines, character abilities, or established rules of the story's world.
3. Note any contradictions between earlier and later parts of the story.

If you find any continuity errors, please provide a clear explanation of the error and why it contradicts earlier information in the story.

Identify and quote the specific lines that:
1. Introduce the continuity error
2. Contain the earlier information that is contradicted by the error

If you do not find any continuity errors, state that no errors were found.

Based on your analysis, make a final decision on whether a continuity error exists in the story.

**Some tips and tricks for the task:**
- Pay attention to even little details in the story, the continuity errors often are not limited to the central plot point.
- If it helps, we recommend taking notes as you make your way through the story
- We recommend reading the story at least two times to assess the continuity error, to ensure the correctness of your answer.
- You might observe some logical error in the story, but make sure that it qualifies as a continuity error i.e. you should be able to find sentences in the story which have the error and the sentences with the original fact that was contradicted (see definitions below for a concrete example).

For more details on the definitions of continuity errors, contradictions, sentences with continuity errors, and sentences contradicted by continuity errors, please refer to the definitions below:

# Definitions

1. **Continuity Error**. A continuity error refers to a logical inconsistency in the story, where an event in the story contradicts some earlier established fact or rule about the story's characters, objects, plot, or the setting (like location or time period). E.g. If the story initially establishes a character to have blonde hair but later the same character is described with dark hair without any explanation, that is a continuity error.
2. **Contradiction**. A statement is said to contradict an established fact if both the statement and the fact cannot be true at the same time. E.g. A fact: "Lady Galadriel had golden hair" is contradicted by the statement: "Lady Galadriel gave a lock of her dark hair to Ghimli".

3. **Sentences with Continuity Error**. These refer to the sentence(s) in the story which introduces the continuity error, contradicting an earlier established fact. Consider the following story as an example:

> Lady Galadriel's golden hair shone so bright that it was believed to shine with the light of the Two Trees of Valinor. Ghimli was swept up with the hair of the elven maiden when he saw her for the first time in Lothlórien. When the time came for the farewell of the fellowship from Lothlórien, the lady asked Ghimli what gift he wanted from her, and the dwarf lord requested for a lock of her hair, the request which was famously denied to Fëanor. To everyone's surprise the lady gave Ghimli a lock of her dark hair. Ghimli could only cry with joy, calling lady Galadriel the fairest of all the maids on Middle earth. That lock of dark hairs, Ghimli would keep with him till the day he died.

In the story above, the sentences T*o everyone's surprise the lady gave Ghimli a lock of her dark hair* and *That lock of dark hairs, Ghimli would keep with him till the day he died* are the Sentences with Continuity Error, as they contradict the earlier established fact that Lady Galadriel had golden hair.

4. **Sentences Contradicted by Continuity Error**. These are the sentence(s) in the story that introduce the fact that is contradicted by the continuity error. E.g. in the Lady Galadriel story above, the sentence *Lady galadriel's golden hair shone so bright that it was believed to shine with the light of the Two Trees of Valinor* establishes that Lady Galadriel had golden hair, which is later contradicted by the continuity error.

# Examples

Below we provide some examples of stories with and without plot holes

## Example 1: Bamboo Cutter Moon Child Story

Long ago, a poor bamboo woodcutter and his wife, childless and sad, found a tiny, radiant girl inside a bamboo stalk. They took her in, named her Princess Moonlight, and their lives were filled with joy and prosperity as they discovered gold and precious stones in the bamboos. The girl grew quickly into a beautiful woman, bringing light and happiness to their home.

Many suitors from far and wide came to seek Princess Moonlight's hand in marriage, but she remained hidden. Five persistent knights, determined to win her, waited outside her home through all seasons, writing letters and poems, but received no response. They

implored the bamboocutter to speak on their behalf, and he urged the Princess to consider marriage for her future security.

Princess Moonlight agreed to meet them only if they could complete seemingly impossible tasks. The first knight was to bring Buddha's stone bowl from India, the second a jeweled branch from Mount Horai, the third the firerat's skin from China, the fourth the dragon's jewel, and the fifth the swallow's shell. The knights, though disheartened, set out on their quests.

The first knight, unable to travel to India, bought a bowl from a Kyoto temple, but it failed the Princess's test. The second knight fabricated a jeweled branch, but his deception was exposed by unpaid jewelers. The third knight obtained a fake firerat's skin, which burned in the fire. The fourth knight sent his servants on a futile search and later abandoned his quest. The fifth knight also failed to find the swallow's shell.

The Emperor, hearing of Princess Moonlight's beauty, sent a court lady to summon her, but she refused. The Emperor visited her himself and fell deeply in love, but she warned that she would disappear if forced to go to the palace. She revealed to her fosterparents and siblings that she was from the moon and would soon return, causing them great sorrow.

On the appointed night, a cloud descended, bringing moon beings to take Princess Moonlight back. Despite the bamboocutter's efforts to protect her, she was taken away, leaving behind a letter and the Elixir of Life for the Emperor. The Emperor, heartbroken, sent the Elixir to Mount Fuji, where it was burned. To this day, smoke is said to rise from the mountain's summit.

**Q. Did you find any continuity errors in the story?**
A. Yes

**Q. If you found an error, please provide an explanation of the error**
A. The couple was stated to be childless and there is no indication later in the story that they had more children. So the sentence that Princess Moonlight revealed to her  foster parents and siblings poses a continuity error.

**Q. If you found an error, please provide the lines of the story that contain the error. In case of multiple sentences, separate them by a semicolon ;**
A. She revealed to her fosterparents and siblings that she was from the moon and would soon return, causing them great sorrow.

**Q. If you found an error, please provide the list of sentences that are contradicted by the continuity error. In case of multiple sentences, separate them by a semicolon ;**
**A.** Long ago, a poor bamboo woodcutter and his wife, childless and sad, found a tiny, radiant girl inside a bamboo stalk.

Example 2: Why Dog And Cat Are Enemies Story

Once upon a time, there was a man and his wife who owned a golden ring that brought prosperity to its owner, though they were unaware of its power. They sold the ring for a small sum and soon fell into poverty, struggling to find their next meal. Their dog and cat also suffered from hunger. Determined to help their owners, the animals devised a plan to retrieve the ring. The dog suggested they obtain the ring from the chest where it was locked, using a mouse to gnaw through and retrieve it.

The cat agreed with the dog's plan and caught a mouse, threatening it to gnaw a hole in the chest and fetch the ring. The mouse complied, and the cat carried the ring in her mouth. Facing a broad river, the dog swam across with the cat on his back. The cat then quickly climbed over obstacles on their way home, while the dog had to go around them. The cat reached home first and delivered the ring to her master, who praised her and promised to care for her.

When the dog arrived, he was scolded and beaten for not helping to bring back the ring. The cat, basking in the warmth of the fireplace, remained silent. Angered by the unfair treatment and the cat's deceit, the dog chased her. Since that day, the enmity between cats and dogs has persisted.

**Q. Did you find any continuity errors in the story?**
A. No

**Q. If you found an error, please provide an explanation of the error**
A. NA

**Q. If you found an error, please provide the lines of the story that contain the error. In case of multiple sentences, separate them by a semicolon ;**
A. NA

**Q. If you found an error, please provide the list of sentences that are contradicted by the continuity error. In case of multiple sentences, separate them by a semicolon ;**

A. NA

## Example 3: Little Boy Blue Story

There once lived a poor widow who supported herself and her only son by gleaning in the fields. They lived in a small cottage at the foot of a beautiful valley by the river. Despite their poverty, the widow was content with her lot, for her home was pleasant, and her lovely boy was a constant delight to her. He had big blue eyes and fair golden curls and loved his mother dearly, always eager to help her with her work.

Years passed happily until the boy was eight years old, but then the widow fell sick, and their little store of money gradually disappeared. She worried about their future, but the boy, determined to help, decided to seek work from the Squire at the Hall. Initially reluctant, the widow finally agreed, making him a new suit from an old dress to ensure he looked presentable.

The Squire, in a kind mood, encountered the boy in his garden. The boy bravely asked for work to support his sick mother. Touched by his plea, the Squire's daughter, Madge, suggested he become their shepherd. The Squire agreed, promising a good wage and a silver horn to call the sheep and cows. Madge named him Little Boy Blue due to his blue attire.

Little Boy Blue returned home to share the good news. His mother wept with joy, knowing the Squire would be a kind master. The next morning, Little Boy Blue received a silver horn and golden cord and began his duties as a shepherd. He was diligent and vigilant, and his mother no longer needed to worry about food, as the Squire paid him well.

Little Boy Blue's mother began to recover, able to walk short distances with his help. However, one day, she slipped and broke her leg. Little Boy Blue found her in pain and managed to get her back to the cottage. He then rowed to the village to fetch the doctor, who treated her but warned she would be bedridden for many days.

The next morning, despite his exhaustion, Little Boy Blue went to work, leaving his mother with food and water. He struggled to stay awake while watching over the horses, but

eventually, he succumbed to sleep. The horses, left unattended, managed to break free from their enclosures and ran amok in the fields, trampling the Squire's crops. The Squire, upon discovering this, was furious and sought out Little Boy Blue.

Little Boy Blue was found asleep by a farmer's lad, Isaac, who informed the Squire. The Squire's daughter, Madge, intervened, comforting the boy and learning of his mother's accident. Moved by his story, the Squire and his daughter accompanied Little Boy Blue to his cottage and arranged for assistance for his mother.

The Squire's daughter sent a basket of dainties and her maid to nurse the widow. Little Boy Blue's mother recovered, and the Squire provided them with a new cottage near the great house. Little Boy Blue continued to faithfully manage the horses, growing up to have a farm of his own. His devotion to his mother had earned him the Squire's trust and friendship, proving that a loving heart and dedication can bring good fortune.

Q. Did you find any continuity errors in the story?
A. Yes

Q. If you found an error, please provide an explanation of the error
A. Little Blue Boy was hired to be a shepherd and call sheeps and cows. Him later managing horses without any explanation contradicts this established information.

Q. If you found an error, please provide the lines of the story that contain the error. In case of multiple sentences, separate them by a semicolon ;
A. He struggled to stay awake while watching over the horses, but eventually, he succumbed to sleep.; Little Boy Blue continued to faithfully manage the horses, growing up to have a farm of his own.

Q. If you found an error, please provide the list of sentences that are contradicted by the continuity error. In case of multiple sentences, separate them by a semicolon ;
A. The Squire agreed, promising a good wage and a silver horn to call the sheep and cows.; The next morning, Little Boy Blue received a silver horn and golden cord and began his duties as a shepherd.

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
