# OpenReview forum: "Finding Flawed Fictions: Evaluating Complex Reasoning in Language Models via Plot Hole Detection"
_colmweb.org/COLM/2025/Conference — COLM 2025_

### Official Review · Reviewer_tF48 · 2025-05-12

**Rating:** 6
**Confidence:** 4
**Ethics Flag:** 1

**Summary:**

The paper introduces FLAWEDFICTIONS, a benchmark for evaluating narrative-level reasoning in large language models (LLMs) through plot-hole (continuity-error) detection. Positive (with plotholes) examples are automatically generated by substituting an early story fact with a contradictory one; all data are subsequently human-verified. The authors benchmark a range of frontier LLMs and find sharp performance drops on long stories.

**Questions To Authors:**

Please address the concerns raised in the "Reasons to Reject" section, particularly points 1 and 2, or clarify if I have overlooked or misunderstood any aspect.

**Reasons To Accept:**

1. New task & dataset. Provides a partially synthetic, human-checked corpus focused on narrative consistency rather than surface comprehension.

2. Broad evaluation. Benchmarks both proprietary (GPT-4o, Claude 3.x) and open-weights models (Llama-3, DeepSeek-R1, Qwen-2.5), exposing weaknesses in long-form reasoning.

3. Initial analysis. Explores how LLM-generated summaries/adaptations introduce plot holes, offering practical diagnostic value.

**Reasons To Reject:**

1. The paper states that the task *typically goes beyond a literal understanding of what happened* (L25), framing it as an advance in narrative reasoning. However, based on my understanding, the dataset is constructed by replacing specific facts in the narrative (e.g., injury location), which resembles prior contradiction or factual error benchmarks. How does this differ fundamentally from existing setups like fact-swapping in news articles or Wikipedia entries? More concretely, in what way does the task actually require *complex and nuanced reasoning* (L26), beyond surface-level consistency checking?


2. Most of the related work cited focuses on pre-LLM narrative reasoning. Beyond MuSR, several relevant datasets and benchmarks have been published prior to the COLM'25 deadline. Please compare your setup against the following recent work and clarify what aspects (task formulation, scale, generation pipeline, or evaluation design) are novel:


[1] EventGround: Narrative Reasoning by Grounding to Eventuality-centric Knowledge Graphs. LREC-COLING'24. https://aclanthology.org/2024.lrec-main.587/

[2] Unveiling Narrative Reasoning Limits of Large Language Models with Trope in Movie Synopses. EMNLP'24. https://aclanthology.org/2024.findings-emnlp.872/

[3] NARRATIVE-OF-THOUGHT: Improving Temporal Reasoning of Large Language Models via Recounted Narratives. EMNLP'24. https://aclanthology.org/2024.findings-emnlp.963/


3. Table 1a shows that some state-of-the-art models already match or exceed human performance on your task. Could you comment on whether this leaves sufficient headroom for future modeling work?

---

> ### Author Response · Authors · 2025-06-02
> **Response**
>
> Thank you for the detailed review. We are glad that you appreciated the new dataset and task in our work as well as found the evaluation “*broad*” and the analysis “*offering practical diagnostic value*”. Below we try to address your concerns:
>
> ## 1. Positioning our work with respect to recent work
>
> Thank you for sharing the three papers in your review. In contrast to ours, **these works do not introduce a new dataset or benchmark, but rather focus on improving narrative understanding using existing benchmarks** (see detailed comparison below).  We believe that our work is novel with respect to all the aspects that you mention in comparison to these papers and provide a detailed comparison below:
>
> -  **Task Novelty**: We propose a novel narrative reasoning problem i.e. detecting plot holes in stories, not studied in any of these works, requiring distinct capabilities. The specified papers on the other hand focus on different problems like temporal reasoning, predicting movie tropes, and grounding story events in real world knowledge, which do not dictate how well can models reason for plot holes in stories.
>
> -  **Algorithm and Benchmark Novelty**: We propose a novel algorithm to automatically edit stories to contain plot holes in them: this partial-synthetic data approach avoids the risk of data contamination and aims to disentangle memorization from LLM reasoning capabilities. In contrast,**the three papers mentioned do not introduce any new benchmarks or methodologies to curate such datasets, and only consider existing datasets in their evaluations**. Jiayang et al. 2024 [1] consider StoryCloze (Mostafazadeh et al. 2016 [5]) and Multiple Choice Narrative Chain (Granroth-Wilding and Clark, 2016 [6]) datasets for their experiments. Su et al. 2024 [2] focuses on trope classification in movie summaries using the existing TiMoS dataset (Chang et al. 2021 [4]) and Zhang et al. 2024 [3] evaluates their prompting technique for the temporal graph generation task using datasets like ProScript (Sakaguchi et al. 2021 [7]).
>
> -  **Quality of Stories included in FlawedFictions**: Our benchmark includes natural human-authored stories, which are edited using our proposed algorithm FlawedFictionsMaker. These include stories with an average of 730 words for FlawedFictions and 2700 words for FlawedFictionsLong. In contrast, Jiayang et al. 2024 [1] and Zhang et al. 2024 [3] work with toy stories containing an enumeration of story events (e.g. Tom was tired, Tom wanted to have fun, Tom bought a movie ticket for Harry Potter) and they are significantly shorter. While Su et al. 2024 [2] focuses on longer stories, which are of similar lengths as our benchmark, these are the synopsis of movies, which potentially have different narrative structures compared to full stories.
>
> - **Novel Evaluation Design**  To evaluate how accurately a model is able to identify a continuity error in the story, we propose a localization task (with its corresponding evaluation metric CEEval-Full) that not only requires the model to correctly predict the story span containing the error but also the span entailing the fact that is contradicted. In contrast, Jiayang et al. 2024  [1] and Su et al. 2024 [2] focus on classification tasks for predicting a story's ending and movie tropes respectively, and Zhang et al. 2024 [3] focuses on a graph prediction problem using the graph edit distance as their evaluation metric.
>
> - **Focus on Assessing Quality of LLM Generated Narratives**: Besides evaluating the narrative reasoning capabilities of LLMs, we also evaluate the quality of LLM generated stories and summaries in terms of presence / absence of continuity errors in them. The three works focused on the reasoning aspect but not on narrative generation, which also distinguishes our work.
>
> To summarize, we propose a novel task of plot hole detection in our work, propose an automatic algorithm to curate a benchmark for the task, and in addition to evaluating LLMs’ capabilities to reason for plot holes, we also evaluate the quality of LLM generated stories in terms of their narrative consistency. We will make sure to update the related work section to include these works in the final version.
>
> ***
> *The response continues in the next comment*

---

> > ### Author Response · Authors · 2025-06-02
> > **Response Continued**
> >
> > ## 2. About the complex and nuanced nature of the task
> >
> > We would like to highlight that our algorithm doesn’t simply involve explicitly swapping the facts, but the generated counterfactuals which are used to construct the final stories can end up modifying more implicit implications of a fact in the story.  For example, modifying the story to take place in April instead of October could transform an otherwise drab landscape description into one full of blooming flowers reminiscent of spring (see  Lines 146-147; the example provided in Figure 1 is meant to simply demonstrate our FlawedFictionsMaker algorithm and the task intuitively). We provide more examples in Appendix section A.7, where the plot holes are more nuanced, and we summarize some of them here: in one of the stories (first example), two characters are shown to leave the house but are present in the house in a later scene. This requires the model to track the state of these characters to correctly reason for the inconsistency. In another story (third example), the plot hole lies in inconsistency in the characterization of one of the characters, who is initially portrayed as courteous and gentle, one who rejoices in success of others, but later the story shows conflict in the court due to his aloofness and self-centered nature. Further, in section 5, we discuss an entailment baseline that looks for contradictions between pairs of sentences in the story, but **this baseline achieves near random accuracy and nearly zero CEEval-Full score** (Table 1), which also highlights the contextual nature of the reasoning task.
> >
> > Hence, FlawedFictions includes performing nuanced contextual reasoning to correctly identify (or lack thereof) plot holes. We leave to future work to explicitly controlling for reasoning complexity during data generation.
> >
> > ***
> >
> > ## 3. Headroom for Future Modeling
> >
> > While few models can perform close to our annotators, the task itself is non trivial for humans due to their limited working memory (Lines 244-247), especially under experimental conditions where participants are required to remember all the introduced facts in a story to reason about contradictions. Hence, while some models can perform close to our annotators, model performance is still far from perfect, especially on the localization task (CEEval-Full metric), where the best model score is score 0.68, the maximum achievable score is 1, and a trivial baseline that always predicts no error obtains a score of 0.5. Note that because we algorithmically introduce our benchmark, we have a high degree of confidence about the localization of the plot holes.
> >
> >  Further, we notice that increased test time compute when using reasoning models like o1 and o3-mini leads to no improvement in performance. Hence, our work leaves room for improving reasoning models that can take advantage of additional test time compute for improving performance (Lines 289-298).
> >
> >  Finally, on the FlawedFictionsLong benchmark, which contains longer stories, **even the best models i.e. o1, only very slightly outperform a trivial baseline that always predicts no error found**. Hence, there is a significant room for improving performance of models on our task for longer stories.
> >
> > ***
> >
> > [1] Jiayang et al. EventGround: Narrative Reasoning by Grounding to Eventuality-centric Knowledge Graphs. LREC-COLING'24.
> >
> > [2] Su et al. Unveiling Narrative Reasoning Limits of Large Language Models with Trope in Movie Synopses. EMNLP'24.
> >
> > [3] Zhang et al. NARRATIVE-OF-THOUGHT: Improving Temporal Reasoning of Large Language Models via Recounted Narratives.
> >
> > [4] Chang et al. Situation and Behavior Understanding by Trope Detection on Films. WWW'21.
> >
> > [5] Mostafazadeh et al. A Corpus and Cloze Evaluation for Deeper Understanding of Commonsense Stories. NAACL'16.
> >
> > [6] Granroth-Wilding & Clark. What Happens Next? Event Prediction Using a Compositional Neural Network Model. AAAI'16.
> >
> > [7] Sakaguchi et al. proScript: Partially Ordered Scripts Generation. EMNLP Findings'21.

---

> > > ### Author Response · Authors · 2025-06-05
> > >
> > > Dear reviewer,
> > >
> > > Thank you so much for your original review. The discussion period ends soon: we would really appreciate if you could let us know your thoughts regarding our answers to your comments and questions.
> > >
> > > Thanks a lot,
> > > FlawedFictions Authors

---

> > ### Comment · Reviewer_tF48 · 2025-06-07
> >
> > The rebuttal satisfactorily addresses most of my concerns, particularly on task novelty; however, a concise breakdown of the specific reasoning skills uniquely required for plot-hole detection ("distinct capabilities", versus prior narrative tasks) would strengthen the argument and could be clarified in a future revision.

---

> > > ### Author Response · Authors · 2025-06-09
> > >
> > > Thank you for going through our response and adjusting your score. We will provide a more concise breakdown of reasoning capabilities required for plot hole detection and their comparison with other narrative reasoning papers in the future revision of the paper.

---

### Official Review · Reviewer_9VkJ · 2025-05-12

**Rating:** 5
**Confidence:** 4
**Ethics Flag:** 1

**Summary:**

The paper proposes a novel benchmark that evaluates the narrative understanding and reasoning capabilities of LLMs by detecting plot holes in stories. Plot holes refer to logical inconsistencies in a storyline that violate the internal logic or established rules of the story's world. The authors introduce an algorithm to automatically synthesize plot holes in human-written stories. This algorithm extracts relevant facts from the first act of a story, negates them in subsequent acts, and generates stories with potential plot holes. The benchmark consists of two tasks: a binary classification task to determine whether a story contains plot holes and a localization task to identify the specific text spans introducing the plot holes and the information they contradict. The paper evaluates the performance of several state-of-the-art LLMs on this benchmark, finding that LLMs struggle to accurately detect plot holes, particularly as story length increases. Furthermore, the authors demonstrate that LLM-generated stories and summaries exhibit significantly higher rates of plot holes compared to human-authored originals.

**Questions To Authors:**

N/A

**Reasons To Accept:**

- Innovative research direction
- Rigorous methodology for synthesizing plot holes
- Comprehensive evaluation

**Reasons To Reject:**

- Algorithm limitations: The algorithm relies heavily on LLMs and human filtering.
- Benchmark limitations: The benchmark primarily consists of short stories (~thousands of words). This may not fully reflect the challenges of plot hole detection in real-world long-form narratives. Furthermore, the benchmark's stories are sourced from specific domains such as fairy tales and myths, which may restrict the generalizability of the evaluation results. And the benchmark only covers human-written stories and neglects machine-generated stories, which I think are more meaningful since we do not expect LLMs to generate illogical stories.
- Evaluation metric limitations: The paper's evaluation metrics focus on the accuracy of model predictions but may not fully capture the richness and complexity of narrative reasoning.
- Lack of practical applications: The paper emphasizes the evaluation of LLMs' narrative reasoning capabilities through plot hole detection but provides limited exploration of practical applications. While it mentions the potential utility of plot hole detection in improving narrative consistency, it does not delve into specific methods or case studies demonstrating how to leverage this capability to enhance the quality of LLM-generated stories.

---

> ### Author Response · Authors · 2025-06-02
> **Response**
>
> Thank you for your review. We are glad you found our research direction “*innovative*” and found our plot hole generation algorithm and evaluations rigorous. Below we try to address your concerns:
>
> ***
>
> >  ***the benchmark only covers human-written stories and neglects machine-generated stories, which I think are more meaningful since we do not expect LLMs to generate illogical stories***
>
> Could you please provide some works supporting your claim that machine generated stories are more meaningful than human authored ones? There is evidence in prior work about LLMs generating narratives with logical inconsistencies [4,5]. Section 6 of our paper also provides evidence towards this phenomenon, where we found LLM generated stories and summaries exhibit a significant rate of detected plot holes.
>
> ***
> > ***The algorithm relies heavily on LLMs and human filtering***
>
> We would really appreciate if you could please elaborate your issue with using LLMs and human filtering for our data generation algorithm? We believe using LLMs with human filtering is actually a strength of our approach -- Using LLMs enables us to automatically generate high quality data for the plot hole detection task, while avoiding the pitfalls of dataset contamination i.e. the stories edited with our approach couldn't be present in LLMs' training data by design. Performing human verification on top of that helps ensure the legitimacy of a story and its corresponding plot hole edited with our algorithm, thus ensuring high quality evaluation data for the benchmark. This approach is also in line with a lot of recent work that uses LLMs to generate synthetic data for building benchmarks [1,2, 3].
>
> ***
>
> > ***The benchmark primarily consists of short stories***
>
> We believe that using stories with relatively short length is actually a strong point of our benchmark, since we are able to test limitations of LLMs, maintain high quality human filtering since the cognitive load increases significantly with length, and enable evaluation of models regardless of the context window they’re trained on. Our algorithm is length-agnostic and enables generating even longer and complex data as model capabilities improve. We'd like to highlight that it is not our intention to fully represent all human narratives, but rather give a general framework to generate narrative understanding evaluations that are more robust to data leakages.
>
> ***
>
> > ***the benchmark's stories are sourced from specific domains such as fairy tales and myths***
>
> We would like to clarify that  FlawedFictions does include other genres than fairytales and myths, like historical fiction , literary fiction, comedy etc. The detailed statistics of these genres which were tagged using GPT-4o-mini is provided below. While Folklore / Fairy Tale does dominate FlawedFictions with 43% stories followed by Mythology with 10% of the stories, there still are ~50% stories coming from other genres. Similar to the point we made about story lengths, our method is agnostic to the story genre and future work can build upon our work to generate stories with different distribution of these genres or even explore other domains like non-fiction.
>
> | Rank | Genre | Count | Percentage |
> |------|-------|-------|------------|
> | 1 | Folklore / Fairy Tale | 179 | 43.2% |
> | 2 | Mythology | 41 | 9.9% |
> | 3 | Literary Fiction | 36 | 8.7% |
> | 4 | Historical Fiction | 32 | 7.7% |
> | 5 | Comedy/Humor | 31 | 7.5% |
> | 6 | Fantasy | 21 | 5.1% |
> | 7 | Satire | 12 | 2.9% |
> | 8 | Romance | 9 | 2.2% |
> | 9 | Crime/Detective | 8 | 1.9% |
> | 10 | Supernatural | 7 | 1.7% |
> | 11 | Adventure | 6 | 1.4% |
> | 12 | Philosophical | 6 | 1.4% |
> | 13 | Drama | 6 | 1.4% |
> | 14 | War | 5 | 1.2% |
> | 15 | Social Commentary | 5 | 1.2% |
> | 16 | Psychological | 3 | 0.7% |
> | 17 | Fable | 2 | 0.5% |
> | 18 | Mystery/Thriller | 2 | 0.5% |
> | 19 | Biographical | 1 | 0.2% |
> | 20 | Western | 1 | 0.2% |
> | 21 | Coming-of-Age | 1 | 0.2% |
>
> ***
>
> > ***The paper's evaluation metrics focus on the accuracy of model predictions but may not fully capture the richness and complexity of narrative reasoning***
>
> We would like to clarify that we do not just evaluate accuracy metric but also evaluate if the model is able to correctly localize the sentences in the story that have the plot hole as well as the sentences that are contradicted by the plot hole (the metric is named CEEval-Full in the paper).  While we initially considered evaluating models’ explanations of the errors as well, those are much harder to evaluate due to their open-ended nature. However, through qualitative checks, we found a strong correspondence between a high CEEval-Full score and the model correctly explaining the plot hole in the story and vice versa.
>
> ***
>
> *The response continues in the next comment*

---

> > ### Author Response · Authors · 2025-06-02
> > **Response Continued**
> >
> > > ***While it mentions the potential utility of plot hole detection in improving narrative consistency, it does not delve into specific methods or case studies demonstrating how to leverage this capability to enhance the quality of LLM-generated stories***
> >
> > We elaborate in the Section 1 of the paper, Lines 44-55, that reasoning for plot holes implicitly tests for multiple sub-capabilities like state tracking, commonsense, pragmatics, and theory of mind reasoning. Further, we also discuss how a system capable of automatically detecting plot holes in stories can be useful to reviewing and editing both human authored and machine generated stories. While it would have been interesting to explore methods to improve narrative consistency of stories, that falls out of the scope of this work, as our focus here is on proposing an algorithm to synthesize plot holes in stories and curating a benchmark for plot hole detection. Future work can build on our work to improve narrative consistency and our paper can serve to provide ways to evaluate effectiveness of such approaches.
> >
> > ***
> >
> > [1] Zayne Rea Sprague, Xi Ye, Kaj Bostrom, Swarat Chaudhuri, Greg Durrett. MuSR: Testing the Limits of Chain-of-thought with Multistep Soft Reasoning. https://openreview.net/forum?id=jenyYQzue1
> >
> > [2] Melanie Sclar, Jane Yu, Maryam Fazel-Zarandi, Yulia Tsvetkov, Yonatan Bisk, Yejin Choi, Asli Celikyilmaz. Explore Theory of Mind: Program-guided adversarial data generation for theory of mind reasoning. https://arxiv.org/abs/2412.12175
> >
> > [3] Xiang Lisa Li, Farzaan Kaiyom, Evan Zheran Liu, Yifan Mai, Percy Liang, Tatsunori Hashimoto. AutoBencher: Towards Declarative Benchmark Construction. https://arxiv.org/abs/2407.08351
> >
> > [4] Mirowski et al. Co-Writing Screenplays and Theatre Scripts with Language Models: Evaluation by Industry Professionals. CHI'23.
> >
> > [5] Chakrabarty et al. Art or Artifice? Large Language Models and the False Promise of Creativity. CHI'24
> >
> > ***

---

> > > ### Author Response · Authors · 2025-06-05
> > >
> > > Dear reviewer,
> > >
> > > Thank you so much for your original review. The discussion period ends soon: we would really appreciate if you could let us know your thoughts regarding our answers to your comments and questions.
> > >
> > > Thanks a lot,
> > > FlawedFictions Authors

---

> > > > ### Comment · Reviewer_9VkJ · 2025-06-06
> > > >
> > > > I do not think I have any misunderstanding of this paper and tend to maintain my score.

---

> > > > > ### Author Response · Authors · 2025-06-08
> > > > >
> > > > > Thank you for your response! We believe the original review had a misinterpretation of our paper on several points such as why we use human authored stories for the benchmark instead of machine generated ones, using LLMs with human verification for benchmark curation, different genres covered in the benchmark (the stats above show genres beyond fairytales and myths), reason for working with short stories, and the evaluation metrics being limited to accuracy (which we clarify isn't true as we have localization task as well). We tried to address these points in our previous response, we hope they clarify our work better.

---

> > > > > > ### Comment · Reviewer_9VkJ · 2025-06-09
> > > > > >
> > > > > > I carefully read the response but it does not convince me. I decide to maintain my score.

---

### Official Review · Reviewer_KNPv · 2025-05-13

**Rating:** 7
**Confidence:** 4
**Ethics Flag:** 1

**Summary:**

This is a well-written paper that provides a sizable study into whether LLMs can reason over plot holes. This is done in three ways: first, through a method (FlawedFictionsMaker) that can create plot-holes; second, through two LLM-generated but human-verified datasets created using this first approach; and third, through a case study. The paper was mostly easy to read (great motivation and description, though maybe a little too much formality in, e.g., Definition 2.1, without "payoff" in the method / results. The results show that LLMs generally struggle with the tasks defined; although Claude 3.5 matches / nearly matches human performance, that human performance is lower than one would traditionally expect (76% and 68%). I agree with the authors that this is a hard task for humans, so this isn't a knock the way that, e.g., a lower IAA score would be.

**Questions To Authors:**

* No IRB is mentioned; please describe the level of IRB review.

* I don't fully understand the continuity error definition, and I don't think it's settled by the "extended" version with reader's beliefs (L116). Specifically, in Def 2.1, you have a continuity error defined as removing a proposition means that we can conclude the negation of that proposition. Principally, the negation is not the opposite, e.g., "not hot" is _not_ cold. Is there a closed world assumption being made here (if not in the prompt, then at least in the setup + definition)?

**Reasons To Accept:**

* The paper provides comprehensive results, both for the core tasks and for the reasonable ablations, such as scalable test-time inference.
* The paper considers a challenging and intriguing LM task / perspective.
* It provides **extensive** examples and description, along with error analysis.

**Reasons To Reject:**

Pending my questions, there are no strong reasons to reject. I would suggest either revising, or augmenting, some of the notation-heavy portions with callouts to a clear example. This formality, especially in section 3, can make the paper a bit harder to read, though the process is pretty straightforward.

---

> ### Author Response · Authors · 2025-06-02
> **Response**
>
> Thank you for taking the time to review our paper and providing detailed feedback. We are glad to know that you found our paper to provide “*comprehensive results*” and the plot hole detection task to be “*challenging and intriguing*”.  Below we try to answer your questions and concerns:
>
> ***
>
> > ***No IRB is mentioned; please describe the level of IRB review***
>
> We applied for two IRBs for this work, one for the dataset verification through annotators on prolific and one for the human performance evaluation through college undergraduates. In both cases we got an IRB exemption from the Human Subjects Division which determined our studies as human subjects research that qualifies for exempt status (Category 101). We will share the IRB IDs for both studies in the final version of the paper.
>
> ***
>
> > ***I don't fully understand the continuity error definition, and I don't think it's settled by the "extended" version with reader's beliefs.***
>
>
> Thank you for this important question. Let us provide a concrete example to clarify the definition and address the negation vs. opposition concern.
>
> Consider the following story:
>
> - Jack is drinking hot tea in a cup
> - Jack gives his tea to Jill
> - Jill drinks the cold tea from Jack’s cup
>
> From this we extract propositions: $\phi_1​ =$ "Jack drinks tea", $\phi_2 =$"tea is hot", $\phi_3 =$ "Jack gives tea to Jill", $\phi_4 =$ "Jill drinks tea", and $\phi_5 =$ "tea is cold".
>
> You are absolutely correct that removing  $\phi_5$​ from our proposition set does not allow us to conclude $\neg \phi_5$​ ("tea is not cold") from the remaining propositions alone, since "hot" does not logically entail "not cold" without additional assumptions.
>
> This is precisely why we need the extended definition with reader beliefs. A reader's commonsense knowledge includes the belief that "if tea is hot, then tea is not cold" ($\phi_2 \to \neg \phi_5$​). When we include this reader belief in our reasoning, we can derive the contradiction:
>
> $\\{ \phi_2 \to \neg \phi_5, \phi_2\\} \vdash \neg \phi_5$​ (by modus ponens)
>
> This shows that $\phi_5$ is logically inconsistent with the rest of the story when combined with reasonable reader beliefs, making it a continuity error according to our extended definition.
>
> The key insight is that continuity errors in fiction cannot be detected purely from explicit textual propositions—they require the integration of implicit world knowledge that readers bring to their understanding of the story. Thanks again for asking this question. We will include similar examples in section 2, so that the definitions are easier to understand.

---

> > ### Comment · Reviewer_KNPv · 2025-06-10
> >
> > Thank you for your response. I'm glad that you had this as IRB exempt.
> >
> > Thank you also for working through the example. It does help clarify, and I'd suggest that you include it in the paper. Broadly, I still think the formalisms in the paper can get in the way of understanding, so I'd also suggest trying to ground them in examples.
> >
> > FWIW, I do not agree with a number of the reasons to reject from other reviewers.

---

### Official Review · Reviewer_smF7 · 2025-05-13

**Rating:** 7
**Confidence:** 4
**Ethics Flag:** 1

**Summary:**

This paper introduces a novel algorithm designed to intentionally insert "plot holes" into human-authored stories. The primary purpose of this algorithm is to generate a collection of stories containing these inconsistencies, thereby creating a benchmark for evaluating the capabilities of LLMs. The authors' findings indicate that current LLMs struggle to identify plot holes within narratives. Additionally, the paper notes that LLMs themselves tend to generate stories or summaries that inadvertently include plot holes. Ultimately, the paper proposes that this newly developed benchmark can be a valuable tool for assessing the reasoning abilities and narrative comprehension of various LLMs.

**Questions To Authors:**

- It would be more helpful to provide a more detailed example of the pipeline? While Figure 1 offers a general overview, an illustration of the story generation process with a level of detail comparable to that found in section 3 would be greatly appreciated.

- I wonder if the authors considered a model (perhaps DeBERT) finetuned on part of the benchmark to assess the capability of such models on this task. I think that baseline could help contextualize the performance of LLMs with finetuned LMs.

- Are stories easily decomposable into three acts? The 3-act structure leads to a formulaic way of storytelling and I wonder the benchmark will be considered equally formulaic. I may have missed it in the appendix but could you provide some quantitative metrics on intermediate accuracy/performance of the LLMs for individual steps in the pipeline to construct the benchmark.

- Possible references to include:

Andrew Piper, Richard Jean So, and David Bamman. 2021. Narrative Theory for Computational Narrative Understanding. In Proceedings of the 2021 Conference on Empirical Methods in Natural Language Processing, pages 298–311, Online and Punta Cana, Dominican Republic. Association for Computational Linguistics.

**Reasons To Accept:**

Firstly, the proposed algorithm, while acknowledging its limitations in the types of plot holes it can generate, represents a practical initial step towards creating controllable story datasets with such flaws. Secondly, the paper includes a comprehensive evaluation of several contemporary LLMs. Furthermore, the resulting benchmark, if made publicly available, has the potential to be a useful resource for evaluating the reasoning abilities of LLMs. Finally, the paper's formalization of continuity errors could be a significant contribution that inspires further research in this area.

**Reasons To Reject:**

I noted that several significant aspects of the paper are located in the appendix. The placement of the error analysis, for instance, in the appendix makes it less accessible and disrupts the flow of reading.

Additionally, the paper's scope appears to be limited to a single type of plot hole. I also wonder if the authors explored the performance of a fine-tuned small language model, in addition to their experiments with large language models.

---

> ### Author Response · Authors · 2025-06-02
> **Response**
>
> Thank you for the detailed review. We are glad that you found our proposed algorithm novel, representing “*practical initial step towards creating controllable story datasets with such flaws*”. We are encouraged to know that you found our evaluation “*comprehensive*” and found our formalization of continuity errors to potentially be “*be a significant contribution that inspires further research in this area*”. We would like to assure you that we will be releasing our benchmark and code for public use with the final paper version.
>
> Please see the response below where we try to address your concerns:
>
> ***
>
> > ***I also wonder if the authors explored the performance of a fine-tuned small language model, in addition to their experiments with large language models.***
>
> We did try fine-tuning the Llama-3.1-8b Instruct model using LoRA on a separate dataset generated with FlawedFictionsMaker and found that fine-tuning does improve performance of the base model but overall performance remains low. The results are provided in the table below (along with performance of some other representative models):
>
> | Model | Accuracy | CEEval-Full |
> |-------|----------|-------------|
> | Llama-3.1-8b | 0.50 | 0.10 |
> | Llama-3.1-8b Finetuned | 0.53 | 0.36 |
> | GPT-4o | 0.64 | 0.58 |
> | o1 (medium) | 0.70 | 0.65 |
> | Claude-3.5-Sonnet | 0.76 | 0.67 |
>
> The training data used here consisted of ~1300 stories (both positive and negative examples) generated from the summaries of stories included in FairyTaleQA. Note that the training data wasn’t verified by human annotators unlike the data we use for the benchmark. We believe that with better filtering of the training data and using alternate training objectives like Reinforcement Learning with Verifiable Rewards (RLVR) might improve the models’ performance further, but we leave that exploration for future work.
>
>
> ***
>
> > ***the paper's scope appears to be limited to a single type of plot hole***
>
> While we agree that we focus on continuity errors in this work, as we explain in the paper (Lines 94-97), continuity errors can entail other forms of plot holes as well like out of character behavior and impossible events. Out of character behavior can be viewed as breaking the continuity of characters' established motivations and characteristics by their behavior later in the story. We have one such examples of FlawedFictionsMaker generating an out of character behavior in the appendix section A.7 (third example), where a character is initially portrayed as courteous and gentle, one who rejoices in success of others, but later the story shows conflict in the court due to his aloofness and self-centered nature. Similarly, impossible events represent events that contradict logic or scientific principles that are consistent with the story’s world. In appendix section A.7 in one of the stories (first example), two characters are shown to leave the house but are present in the house in a later scene,  representing a logically impossible event as a character cannot (normally) be in two places at the same time. However, we do acknowledge that these cases are relatively rare. Based on our estimates using GPT-4o ~22% of the continuity errors in the dataset can be interpreted as Out of Character Behavior and 3% as Impossible events. Future work can explore extensions to FlawedFictionsMaker for controlling the specific type and complexity of plot holes to be introduced.
>
> ***
>
> > ***I noted that several significant aspects of the paper are located in the appendix.***
>
> Due to the limited space we had to move some of the details to the Appendix. However, we tried to ensure that the main paper can be read in a self contained manner. Since CoLM provides an extra page for the final version, we will move results from section A.5 to the main paper, which contains the additional error analysis.
>
> ***
>
> *The response continues in the next comment*

---

> > ### Author Response · Authors · 2025-06-02
> > **Response to the Questions**
> >
> > Below we provide answers to your questions.
> >
> >
> > Q1. ***It would be more helpful to provide a more detailed example of the pipeline?***
> >
> > We will add another figure in the main paper demonstrating each step of the pipeline on one of the stories in our dataset.
> >
> > ***
> >
> > Q2. ***I wonder if the authors considered a model (perhaps DeBERT) finetuned on part of the benchmark.***
> >
> > Please see the point 2 in the response above.
> >
> > ***
> >
> > Q3. ***Are stories easily decomposable into three acts?***
> >
> > Not all stories are easily decomposable into a three act structure. For our benchmark, we primarily worked with fairytale stories, which do tend to be more formulaic in terms of their narrative structure. In that sense, yes stories in our benchmark are as formulaic as the base stories used to construct it. We would also like to highlight that a story having a three act structure is not necessary for our algorithm to work, and any decomposition of a story into smaller units should work (e.g. paragraph wise split). The purpose of using a three act decomposition is that it provides us with a neat first section of the story, where most of the exposition is delivered, giving us an ample amount of facts to be flipped in the following acts.
> >
> > About the accuracy of the LLMs on individual tasks we do not measure them because we do not have ground truths for these tasks. However, we perform human validation at the end of the pipeline so that if there was an error by the LLM in any of the individual steps that would reflect in the output and that particular story and its corresponding plot hole would be rejected by the human annotators.
> >
> > ***
> >
> > Q4. ***Possible references to include.***
> >
> > Thank you for sharing the reference, we will include it in the Related Work section of our paper.
> >
> > ***

---

### Author Response · Authors · 2025-06-02
**General Response**

We are grateful to all the reviewers for spending their valuable time on our paper and providing helpful feedback. We are encouraged to learn that they found our proposed task of plot hole detection innovative (Reviewers KNPv, 9VkJ, tF48) with Reviewer KNPv noting that "*the paper considers a challenging and intriguing LM task / perspective*",  and appreciated the rigorous nature of our proposed algorithm to edit existing stories with plot holes (Reviewers smF7, 9VkJ). All reviewers agreed about the comprehensiveness of the evaluation that we perform on our benchmark with Reviewers KNPv and tF48 also appreciating the ablations and analysis. We are also glad that the Reviewer smF7 found our formalization of continuity errors to potentially be a *“significant contribution that inspires further research in this area”*.

We address all reviewer comments individually. Since reviewers did not have questions/concerns in common, we highlight here some crucial clarifications for the proper understanding and framing of our work.

***9VkJ mentions as a benchmark limitation that “the benchmark only covers human-written stories and neglects machine-generated stories, which I think are more meaningful since we do not expect LLMs to generate illogical stories”***

We do not think that LLM generated stories being more “*meaningful*” than human authored ones is an accurate claim, and we believe our understanding to be a widespread intuition in the research community. Prior work [4,5] provides evidence that LLMs tend to generate narratives with logical inconsistencies, and experiments in Section 6 of our paper also contribute evidence to  this phenomenon by showing that LLM generated stories and summaries exhibit a significant rate of detected plot holes.

***

***9VkJ mentions as an algorithm limitation that “The algorithm relies heavily on LLMs and human filtering”.***

We disagree that this is a limitation to our algorithm, but rather a strong point. Using LLMs makes applying complex operations included in our algorithm to stories tractable, such as fact extraction, where we extract facts established in the first act of a story, and counterfactual story generation, which involves generating an alternate version of the story where a specified fact was not true. This helps in automatically generating data for evaluation and avoid pitfalls of dataset contamination. We run human verification on top of our algorithm to ensure that the stories and their respective plot holes that we include in our benchmark are legitimate and ensure high quality evaluation data.

We would be happy to discuss the reviewer's concerns further and would appreciate it if they provided some explanation on why they believe this to be a limitation besides this statement.


***

***tF48’s concern about differences with prior work on narrative reasoning with respect to*** *“task formulation, scale, generation pipeline, or evaluation design”*:

The three papers [1,2,3] cited by the reviewer are distinct from our work, **as these works do not introduce a new dataset or benchmark and rather focus on improving narrative reasoning for existing datasets**. We focus on a completely different reasoning task of plot hole detection requiring its own evaluation design while the others focus on knowledge graph event grounding, trope identification, and temporal reasoning. We also introduce an algorithm to automatically edit stories to contain plot holes in them and use that to curate our benchmark. No such generation pipeline has been used in these three papers.

***

***tF48’s concern about the nuanced nature of the task.***

In response to Reviewer tF48’s comment “*what way does the task actually require complex and nuanced reasoning (L26), beyond surface-level consistency checking*”, we highlight examples of stories in the paper that require tracking the state of entities in a story and reasoning about character motivations. For a more quantitative analysis representing the complex contextual nature of our benchmark, we re-state the results from the paper where a baseline that checks for contradiction between all pairs of sentences within a story results in random performance on the classification task and nearly zero score on the localization task included in our benchmark.

[1] Jiayang et al. EventGround: Narrative Reasoning by Grounding to Eventuality-centric Knowledge Graphs. LREC-COLING'24.

[2] Su et al. Unveiling Narrative Reasoning Limits of Large Language Models with Trope in Movie Synopses. EMNLP'24.

[3] Zhang et al. NARRATIVE-OF-THOUGHT: Improving Temporal Reasoning of Large Language Models via Recounted Narratives.

[4] Mirowski et al. Co-Writing Screenplays and Theatre Scripts with Language Models: Evaluation by Industry Professionals. CHI'23.

[5] Chakrabarty et al. Art or Artifice? Large Language Models and the False Promise of Creativity. CHI'24.

---

### Decision · Program_Chairs · 2025-07-08

**Decision:**

Accept

**Comment:**

This paper introduces the task of "plot hole detection" -- identifying whether a story contains a continuity error (where one fact is given that contradicts the logic of the rest of the story), and also identifying the specific place (e.g., sentnece) of that error.  It designs a new LLM-based pipeline for synthesizing stories containing plot holes from existing stories (including segmenting into three acts, extracting propositions, generating counterfactuals etc.), which critically involves human verification. It uses this dataset to benchmark the performance of a range of LLMs (where most struggle on the task), and also to provide an automatic means to assess the degree to which LLM stories themselves contain plot holes (as others -- both in research and in the general public -- have noted their tendency toward long-form logical inconsistency).  Reviewers in general found this work creative and thorough in its experiments and error analysis, with specific contributions in formalizing the concept of a "continuity error" and developing a dataset that could be useful for further analysis.  Weaknesses around contextualizing this work with respect to other narrative reasoning research, and understanding the headroom for improvement with respect to human performance were generally addressed in the author response.  Overall a strong and original paper that would merit interesting discussion.